# Graph Data Selection for Domain Adaptation: A Model-Free Approach

**Ting-Wei Li**
University of Illinois
Urbana-Champaign, IL USA
twli@illinois.edu

**Ruizhong Qiu**
University of Illinois
Urbana-Champaign, IL USA
rq5@illinois.edu

**Hanghang Tong**
University of Illinois
Urbana-Champaign, IL USA
htong@illinois.edu

## Abstract

Graph domain adaptation (GDA) is a fundamental task in graph machine learning, with techniques like shift-robust graph neural networks (GNNs) and specialized training procedures to tackle the distribution shift problem. Although these *model-centric* approaches show promising results, they often struggle with severe shifts and constrained computational resources. To address these challenges, we propose a novel *model-free* framework, GRADATE (*GRAph DATa sElector*), that selects the best training data from the source domain for the classification task on the target domain. GRADATE picks training samples without relying on any GNN model's predictions or training recipes, leveraging optimal transport theory to capture and adapt to distribution changes. GRADATE is *data-efficient*, scalable and meanwhile complements existing model-centric GDA approaches. Through comprehensive empirical studies on several real-world graph-level datasets and multiple covariate shift types, we demonstrate that GRADATE outperforms existing selection methods and enhances off-the-shelf GDA methods with much fewer training data.

## 1 Introduction

Graphs have emerged as a fundamental data structure for representing complex relationships across diverse domains, from modeling molecular interactions in biological networks [23, 6, 37, 85] to capturing user behaviors in recommendation systems [15, 19, 8, 9, 73]. In graph-level classification tasks [64, 66, 81, 20, 38], where the goal is to categorize graph structures, the distribution shift between source and target domains poses significant challenges. While numerous graph neural network (GNN) methods have been proposed for graph domain adaptation (GDA) [61, 12, 62, 55, 34], they predominantly relies on model architecture design and training strategies, which are inherently *model-dependent* and brittle. This model-centricity introduces practical challenges: (i) the need for extensive resources to train and validate different architectural variants and (ii) the ignorance of source data quality. To address these aforementioned issues, rather than relying on sophisticated GNN architectures or training procedures, we aim to answer a fundamental question:

*How to select the most relevant source domain data, based on available validation data, for better graph-level classification accuracy evaluated on the target domain ?*

In this paper, we propose a *model-free* method, GRADATE (*GRAph DATa sElector*), that selects a subset of important training data in the source domain independently of any specific GNN model design, making it both *data-efficient* and versatile. GRADATE reduces computational overhead and enables quick adaptation to unseen graph domains based on available validation data. Conceptually, GRADATE first leverages Fused Gromov-Wasserstein (FGW) distance [57] to compare graph samples. We provide a theoretical justification to demonstrate FGW's unique advantage over multi-layer GNNs for graph comparison. Then, FGW is used as a building block to measure the dataset-level distance between training and validation sets, which is termed as Graph Dataset Distance (GDD). Through

39th Conference on Neural Information Processing Systems (NeurIPS 2025).

theoretical analysis, we demonstrate that the domain generalization gap between source and target domain is upper-bounded by GDD, which naturally motivates us to minimize this GDD between training and validation sets by identifying the optimal subset of training graph data. At the core of GRADATE lies our novel optimization procedure, GREAT (*GDD sh**R**inkag**E** via sp**A**rse projec**T**ion*), that interleaves between (i) optimal transport-based distribution alignment with gradient updates on training sample weights and (ii) projection of training sample weights to sparse probability simplex. This dual-step process systematically increases the weights of beneficial training samples while eliminating the influence of detrimental samples that could harm generalization performance.

By extensive experiments on six real-world graph-level datasets and two types of covariate shifts, we first show that GRADATE significantly outperforms existing data-efficient selection methods. When coupled with vanilla GNNs that are only trained on its selected data, GRADATE even surpasses state-of-the-art GDA methods. Intriguingly, the practical implications of GRADATE extend beyond mere data selection. By operating independently of model architecture, GRADATE can seamlessly complement existing off-the-shelf GDA methods, further enhancing their performance through better data curation while significantly improving the *data-efficiency*.

In summary, our main contributions in this paper are as follows:

1. **Theoretical justification of FGW distance.** We show in Theorem 3.1 that the output distance of multi-layer GNNs is upper-bounded by Fused Gromov–Wasserstein distance [57] between graphs, which motivates us to use FGW as a building block to compare graphs.
2. **Novel graph dataset distance formulation.** Through Theorem 3.3, we prove that the graph domain generalization gap is upper-bounded by a notion of Graph Dataset Distance (GDD).
3. **A strong model-free graph selector.** We introduce GRADATE (*GRA**ph** DAT**a** s**E**lector*) as the first *model-free* method tailored for domain shift problem for graph-level classification tasks, complementing the predominant model-centric GDA methods (see Section 4.2).
4. **A data-efficient and powerful GDA method.** We show that trained with data selected by GRADATE, even vanilla GNN can beat sophisticated GDA baselines (see Section 4.3).
5. **A universal GDA model enhancer.** We demonstrate that the data selected by GRADATE can be combined with GDA methods to further enhance their performance and efficiency (see Section 4.4).

The rest of the paper is organized as follows. Section 2 introduces needed background knowledge. Section 3 details our definition of Graph Dataset Distance (GDD) and proposed GRADATE, followed by experimental results in Section 4. Section 5 presents related works. Finally, the conclusion is provided in Section 6.

## 2 Preliminaries

In this section, we briefly introduce optimal transport and graph optimal transport, which are fundamental background related to our proposed method. We also provide the formal problem formulation of graph domain adaptation (GDA) in Appendix K.

### 2.1 Optimal Transport

Optimal transport (OT) [27] defines a distance between probability distributions. It is defined as follows: given cost function $d(\cdot, \cdot) : \mathcal{X} \times \mathcal{X} \to \mathbb{R}^{\geq 0}$ and probability distributions $\mathbf{p}, \mathbf{q} \in \mathcal{P}(\mathcal{X})$, where $\mathcal{X}$ is a metric space, $\mathrm{OT}(\mathbf{p}, \mathbf{q}, d) \triangleq \min_{\pi \in \Pi(\mathbf{p}, \mathbf{q})} \int_{\mathcal{X} \times \mathcal{X}} d(x, x') \pi(x, x') \, \mathrm{d}x \mathrm{d}x'$, where $\Pi(\mathbf{p}, \mathbf{q})$ is the set of couplings with marginals $\mathbf{p}$ and $\mathbf{q}$. For supervised learning scenarios, we consider the empirical measures: $\mathbf{p} = \frac{1}{n} \sum_{i=1}^{n} \delta_{x_i}$ and $\mathbf{q} = \frac{1}{m} \sum_{j=1}^{m} \delta_{x'_j}$, where $\delta$ is the Dirac delta function. With the pairwise cost matrix $\mathbf{M} = [d(x_i, x'_j)]_{ij}$, we can re-formulate the OT problem as $\mathrm{OT}(\mathbf{p}, \mathbf{q}, d) = \mathrm{OT}(\mathbf{p}, \mathbf{q}, \mathbf{M}) \triangleq \min_{\pi \in \Pi(\mathbf{p}, \mathbf{q})} \sum_{i=1}^{n} \sum_{j=1}^{m} \mathbf{M}_{ij} \pi_{ij}$.

*Optimal Transport Dataset Distance (OTDD).* Alvarez-Melis and Fusi [2] construct a distance metric between *datasets*, where each dataset is represented as a collection of feature-label pairs $z = (x, y) \in \mathcal{Z}(= \mathcal{X} \times \mathcal{Y})$. The authors propose *label-specific distributions*, which can be seen as distributions over features $X$ of data samples with a specific label $y$, i.e. $\alpha_y(X) \triangleq P(X|Y = y)$. The metric on the space of feature-label pairs can thus be defined as a combination of *feature distance* and *label distance*: $d_{\mathcal{Z}}((x, y), (x', y')) \triangleq (d_{\mathcal{X}}(x, x')^r + c \cdot d(\alpha_y, \alpha_{y'})^r)^{1/r}$, where $d_{\mathcal{X}}$ is a metric on $\mathcal{X}$,

$d(\alpha_y, \alpha_{y'})$ is the *label distance* between distributions of features associated with labels $y$ and $y'$, $r \geq 1$ is the order of the distances and $c \geq 0$ is a pre-defined weight parameter. Consider two datasets $D_s = \{z_i^s : (x_i^s, y_i^s)\}_{i \in [n_1]}$, $D_t = \{z_i^t : (x_i^t, y_i^t)\}_{i \in [n_2]}$ and their corresponding uniform empirical distributions $\mathbf{p}, \mathbf{q} \in \mathcal{Z}$, where $\mathbf{p} = \sum_{i=1}^{n_1} \frac{1}{n_1} \delta_{z_i^s}$ and $\mathbf{q} = \sum_{i=1}^{n_2} \frac{1}{n_2} \delta_{z_i^t}$, OTDD between $D_s$ and $D_t$ is computed as: $\text{OTDD}(D_s, D_t) = \text{OT}(\mathbf{p}, \mathbf{q}, d_{\mathcal{Z}}) = \text{OT}(\mathbf{p}, \mathbf{q}, \mathbf{M}) = \min_{\pi \in \Pi(\mathbf{p},\mathbf{q})} \mathbb{E}_{(z^s,z^t) \sim \pi}[d_{\mathcal{Z}}(z^s, z^t)]$, where $\Pi(\mathbf{p}, \mathbf{q})$ is the set of valid couplings and $\mathbf{M} = [d_{\mathcal{Z}}(z_i^s, z_j^t)]_{ij}$ is the pairwise cost matrix.

## 2.2 Graph Optimal Transport

The Fused Gromov-Wasserstein (FGW) distance [57] integrates the Wasserstein distances [52] and the Gromov-Wasserstein distances [54, 42]. Formally, a graph $\mathcal{G}$ with $n$ nodes can be represented as a distribution over vectors in a $d$-dimensional space, where $d$ is the dimension of the node feature. The *features* are represented as $\mathbf{X} \in \mathbb{R}^{n \times d}$ and the *structure* can be summarized in an adjacency matrix $\mathbf{A} \in \mathbb{R}^{n \times n}$. We further augment $\mathcal{G}$ a probability distribution $\mathbf{p} \in \mathbb{R}^n$ over nodes in the graph, where $\sum_{i=1}^{n} \mathbf{p}_i = 1$ and $\mathbf{p}_i \geq 0, \forall i \in [n]$. To compute FGW distance between two attributed graphs ($\mathcal{G}_1 = \{\mathbf{A}_1, \mathbf{X}_1, \mathbf{p}_1\}$ and $\mathcal{G}_2 = \{\mathbf{A}_2, \mathbf{X}_2, \mathbf{p}_2\}$), we use pairwise feature distance as inter-graph distance matrix and adjacency matrices as intra-graph similarity matrices. The FGW distance is defined as the solution of the following optimization problem:
$\text{FGW}_\alpha(\mathcal{G}_1, \mathcal{G}_2) \triangleq \text{FGW}_\alpha([\|\mathbf{X}_1[i] - \mathbf{X}_2[j]\|_r]_{ij}, \mathbf{A}_1, \mathbf{A}_2, \mathbf{p}_1, \mathbf{p}_2, \alpha)$

$$= \left( \min_{\boldsymbol{\pi} \in \Pi(\mathbf{p}_1, \mathbf{p}_2)} \sum_{i,j,k,l} (1-\alpha)\|\mathbf{X}_1[i] - \mathbf{X}_2[j]\|_2^r + \alpha|\mathbf{A}_1(i,k) - \mathbf{A}_2(j,l)|^r \boldsymbol{\pi}(i,j)\boldsymbol{\pi}(k,l) \right)^{\frac{1}{r}},$$

where $\Pi(\mathbf{p}_1, \mathbf{p}_2) \triangleq \{\boldsymbol{\pi} | \boldsymbol{\pi}\mathbf{1}_{n_2} = \mathbf{p}_1, \boldsymbol{\pi}^T\mathbf{1}_{n_1} = \mathbf{p}_2, \boldsymbol{\pi} \geq 0\}$ is the collection of feasible couplings between $\mathbf{p}_1$ and $\mathbf{p}_2$, $\alpha \in [0, 1]$ acts as a trade-off parameter, and $r \geq 1$ is the order of the distances.

## 3 Methodology

In this section, we propose our framework, *GRAph DATa sElector*, abbreviated as GRADATE. In Section 3.1, we first introduce Theorem 3.1 to motivate our use of LinearFGW [46] for graph distance computation. After that, we define the Graph Dataset Distance (GDD), which measures the discrepancy between graph sets. In Section 3.2, we provide Theorem 3.3 to bound the domain generalization gap using GDD and then propose a GDD minimization problem that is solved by a novel optimization algorithm, GREAT (*GDD shRinkagE via spArse projecTion*). Finally, we introduce GRADATE that combines GDD computation and GREAT to select the most important subset of the training data to solve graph domain adaptation problem (definition is detailed in Appendix K).

### 3.1 Graph Dataset Distance (GDD): A Novel Notion to Compare Graph Datasets

#### 3.1.1 FGW Distance For Graph Comparison

*Challenges in Graph Distance Computation.* To find the optimal samples that can achieve better performance on the target domain, we first need an efficient way that can accurately capture the discrepancy among graphs. To achieve this, most methods rely on Graph Neural Networks (GNNs) [62, 12, 55, 72] to obtain meaningful representations of these structured objects. However, these approaches face critical limitations: (i) high computational complexity to train on full training set and (ii) sensitivity to GNN hyper-parameters. To address these drawbacks, we propose to use FGW distance [57] for replacing GNNs. As demonstrated in the following Theorem 3.1, FGW offers provable advantages that make it more suitable than GNNs to compare attributed graphs.

**Theorem 3.1.** *Given two graphs $\mathcal{G}_1 = (\mathbf{A}_1, \mathbf{X}_1)$ and $\mathcal{G}_2 = (\mathbf{A}_2, \mathbf{X}_2)$, for a $k$-layer graph neural network (GNN) $f$ with ReLU activations, under regularity assumptions in Appendix F.1.1, we have*

$$d_{\mathrm{W}}(f(\mathcal{G}_1), f(\mathcal{G}_2)) \leq C \cdot \text{FGW}_\beta(\mathcal{G}_1, \mathcal{G}_2), \tag{1}$$

*where $d_{\mathrm{W}}$ denotes the $r$-Wasserstein distance, $C$ and $\beta$ are constants depending on GNN $f$, regularity constants and $k$.*

*Proof.* The proof is in Appendix F.1. □

*Implication of Theorem 3.1.* We have the following two main insights: (i) since the theorem holds for any possible $k$-layer GNNs, with ReLU activations, FGW is provably able to capture the differences between attributed graphs in a way that upper-bounds the discrepancy between learned GNN representations; (ii) to the best of our knowledge, this is the first time that FGW is proved to be the distance metric that enjoys this theoretical guarantee, and hence we adopt it as the major basis for our graph data selection method.

*Practical Consideration.* We utilize LinearFGW [46] as an efficient approximation of FGW distance. Formally, LinearFGW defines a distance metric $d_{\text{LinearFGW}}(\cdot, \cdot)$ where $d_{\text{LinearFGW}}(\mathcal{G}_i, \mathcal{G}_j)$ is the LinearFGW distance between any pair of graphs $\mathcal{G}_i, \mathcal{G}_j$. LinearFGW offers an approximation to FGW with linear time complexity with respect to the number of training graphs. While we omit the details of LinearFGW here for brevity, they can be found in Appendix A and Algorithm 1.

### 3.1.2 Graph-Label Distance

With FGW as a theoretically grounded metric for *model-free* comparison between individual attributed graphs, we further extend it to compare sets of labeled graphs across domains, which aids our domain adaptation process. Inspired by OTDD [2] (detailed in Section 2.1), we extend the original definition of *label distance* to incorporate distributions in graph subsets $S, S'$, namely $\alpha_y^S$ and $\alpha_{y'}^{S'}$. Given a set of labeled graphs $\mathcal{D} = \{\mathcal{G}_i, y_i\}_{i=1}^N$ and label set $\mathcal{Y}$, we formulate the *graph-label distance* between label $y \in \mathcal{Y}$ in graph subset $S = \{\mathcal{G}_i^S, y_i^S\}_{i \in |S|} \subseteq \mathcal{D}$ and label $y' \in \mathcal{Y}$ in graph subset $S' = \{\mathcal{G}_j^{S'}, y_j^{S'}\}_{j \in |S'|} \subseteq \mathcal{D}$ as follows:

$$d(\alpha_y^S, \alpha_{y'}^{S'}) = \text{OT}(\mathbf{p}_y^S, \mathbf{q}_{y'}^{S'}, d_{\text{LinearFGW}}), \tag{2}$$

where $\mathbf{p}_y^S = \frac{1}{|i:y_i^S=y|} \sum_{i:y_i^S=y} \delta_{\mathcal{G}_i^S}$, $\mathbf{q}_{y'}^{S'} = \frac{1}{|j:y_j^{S'}=y|} \sum_{j:y_j^{S'}=y} \delta_{\mathcal{G}_j^{S'}}$ are label-specific uniform empirical measures with the distance metric measured by LinearFGW. Intuitively, we collect graphs in subset $S$ with label $y$ and graphs in subset $S'$ with label $y'$ as distributions. Then, we compute the optimal transport distance between these distributions and define the distance as *graph label distance*.

### 3.1.3 Graph Dataset Distance (GDD)

Building upon the aforementioned graph-label distance, we then propose the notion of Graph Dataset Distance (GDD), which compares two graph subsets at a dataset-level. Specifically, based on Equation (2), we can define a distance metric $d_{g\mathcal{Z}}^c$ between graph subsets $S, S'$:

$$d_{g\mathcal{Z}}^c((\mathcal{G}_i^S, y_i^S), (\mathcal{G}_j^{S'}, y_j^{S'})) = d_{\text{LinearFGW}}(\mathcal{G}_i^S, \mathcal{G}_j^{S'}) + c \cdot d(\alpha_y^S, \alpha_{y'}^{S'}), \tag{3}$$

which is a combination of LinearFGW distance and graph-label distance with a weight parameter $c \geq 0$ balancing the importance of two terms. GDD can thus be defined as:

$$\text{GDD}(\mathcal{D}^S, \mathcal{D}^{S'}) = \text{OT}(\mathbf{p}^S, \mathbf{q}^{S'}, d_{g\mathcal{Z}}^c), \tag{4}$$

where $\mathbf{p}^S = \frac{1}{|S|} \sum_{i \in [|S|]} \delta_{(\mathcal{G}_i^S, y_i^S)}$ and $\mathbf{q}^{S'} = \frac{1}{|S'|} \sum_{j \in [|S'|]} \delta_{(\mathcal{G}_j^{S'}, y^{S'})}$ are uniform empirical measures. We summarize the computation of GDD in Appendix B and Algorithm 2.

*Remark* 3.2. If we set $c = 0$, GDD will omit the label information and only consider the distributional differences of graph data themselves, which matches the setting of unsupervised GDA problem.

## 3.2 GRADATE: A Model-Free Graph Data Selector

### 3.2.1 GDD Bounds Domain Generaization Gap

We give the following Theorem 3.3 to elucidate the utility of GDD and its relation to model performance discrepancy between graph domains. In short, we seek to utilize empirical observations in the source domain to minimize the expected risk calculated on the target domain $P_t$, namely, $\mathbb{E}_{(\mathcal{G}, y) \sim P_t}[\mathcal{L}(f(\mathcal{G}), y)]$, which promotes the model performance (i.e., lower expected risk) on the target domain.

**Theorem 3.3** (Graph Domain Generalization Gap). *Define the cost function among graph-label pairs as $d_{g\mathcal{Z}}^c$ with some positive $c$ (via Equation (3)). Let $\mathbf{w}$ denote the source distribution weight. Suppose that $d_{g\mathcal{Z}}^c$ is $C$-Lipschitz. Then for any model $f$ trained on a training set, we have*

$$\mathbb{E}_{(\mathcal{G},y)\sim\mathbf{q}^{\mathrm{val}}}[\mathcal{L}(f(\mathcal{G}),y)] \leq \mathbb{E}_{(\mathcal{G},y)\sim\mathbf{p}^{\mathrm{train}}(\mathbf{w})}[\mathcal{L}(f(\mathcal{G}),y)] + C \cdot \mathrm{GDD}(\mathcal{D}_{\mathbf{w}}^{\mathrm{train}}, \mathcal{D}^{\mathrm{val}}),$$

*where* $\mathrm{GDD}(\mathcal{D}_{\mathbf{w}}^{\mathrm{train}}, \mathcal{D}^{\mathrm{val}}) = \mathrm{OT}(\mathbf{p}^{\mathrm{train}}(\mathbf{w}), \mathbf{q}^{\mathrm{val}}, d_{g\mathcal{Z}}^c)$ *is the graph dataset distance between the weighted training dataset (defined by* $\mathbf{w}$*) and target dataset.*

*Proof.* The proof can be found in Appendix F.2. □

*Implication of Theorem 3.3.* If we can lower the GDD between training and validation data, the discrepancy in the model performance with respect to training and validation sets may also decrease. Specifically, under the scenario where distribution shift occurs, some source data might be irrelevant or even harmful when learning a GNN model that needs to generalize well on the target domain. This motivates us to present our main framework, GRADATE (*GRAph DATa sElector*), which selects the best training data from the source domain for graph domain adaptation.

### 3.2.2 GDD Minimization Problem

Guided by the implication of Theorem 3.3, we formulate the GDD minimization problem as follows.

**Definition 3.4** (Graph Dataset Distance Minimization). Given training and validation sets $\mathcal{D}^{\mathrm{train}}$ and $\mathcal{D}^{\mathrm{val}}$, we aim to find the best distribution weight $\mathbf{w}^*$ that minimizes GDD betwen the training and validation sets under the sparsity constraint. Namely,

$$\mathbf{w}^* = \min_{\mathbf{w}} \mathrm{OT}(\mathbf{p}^{\mathrm{train}}(\mathbf{w}), \mathbf{q}^{\mathrm{val}}, d_{g\mathcal{Z}}^c), \qquad \text{s.t.} \sum_i \mathbf{w}[i] = 1, \mathbf{w} \geq 0, \|\mathbf{w}\|_0 \leq \lfloor n \cdot \tau \rfloor, \quad (5)$$

where $\mathbf{p}^{\mathrm{train}}(\mathbf{w}) = \sum_{i\in[n]} \mathbf{w}_i \delta_{(\mathcal{G}_i^{\mathrm{train}}, y_i^{\mathrm{train}})}$ and $\mathbf{q}^{\mathrm{val}} = \frac{1}{m} \sum_{j\in[m]} \delta_{(\mathcal{G}_j^{\mathrm{val}}, y_j^{\mathrm{val}})}$.

*Optimization Procedure (*GREAT *algorithm).* To solve the above GDD minimization problem, we propose GREAT (*GDD shRinkagE via spArse projecTion*) to optimize the weight $\mathbf{w}$ over the entire training set. Starting from a uniform training weight vector $\mathbf{w}$, GREAT iteratively refines the importance of training samples through two key steps. In each iteration, it first computes the Graph Dataset Distance (GDD) between the reweighted training distribution $\mathbf{p}^{\mathrm{train}}(\mathbf{w})$ and the validation distribution $\mathbf{q}^{\mathrm{val}}$, using a pairwise cost matrix $\tilde{\mathbf{D}} \in \mathbb{R}^{n\times m}$ that encodes distances between individual training and validation samples. The gradient of GDD with respect to $\mathbf{w}$, denoted $\mathbf{g}_{\mathbf{w}}$, is then used to update the weights[1]. Following this, the weight vector $\mathbf{w}$ is sparsified by retaining only the top-$k$ largest entries and re-normalized to remain on the probability simplex. After $T$ such iterations, the non-zero indices in the final $\mathbf{w}$ define the selected training subset $\mathbf{S}$. The detailed algorithm procedure is presented in Appendix C and Algorithm 3.

### 3.2.3 GRADATE

Combining GDD computation and optimization module GREAT, we summarize the main procedure of GRADATE in Algorithm 4 (details can be found in Appendix D). In short, given training and validation data, GRADATE iteratively calculates GDD based on current training weight and searches for a better one through GREAT. The final output of GRADATE corresponds to the selected subset of training data that is best suitable for adaptation to the target domain. We further provide the time complexity analysis of GRADATE as follows.

*Time Complexity of* GRADATE. Let $N$ be the number of training graphs, $M$ be the number of validation graphs, $n$ be the number of nodes in each graph (WLOG, we assume all graph share the same size), $L$ be the largest class size, $\tau$ is the approximation error introduced by approximate OT solvers [2], $K$ be the number of iterations for solving LinearFGW, and $T$ the number of update steps used in GREAT. The runtime complexity can be summarized in the following proposition. [3]

---

[1] Note that we leverage the conclusion introduced in [26] to compute this gradient (stated as Theorem F.3).

[2] This is due to the entropic regularization in Sinkhorn iterations for empirical OT calculation.

[3] For empirical runtime behavior, we refer readers to Appendix M.

**Proposition 3.5** (Time Complexity Analysis [2, 26, 1]). *The off-line procedure of* GRADATE *(i.e. can be computed before accessing the test set) has the time complexity* $\mathcal{O}(NMKn^3 + NML^3 \log L)$ *and the on-line procedure of* GRADATE *has the time complexity* $\mathcal{O}(TNM \log(\max(N, M))\tau^{-3})$.

## 4 Experiments

We conduct extensive experiments to evaluate the effectiveness of GRADATE across six real-world graph classification settings under two different types of distribution shift. Our experiments are designed to answer the following research questions:

- (**RQ1**): How does GRADATE compare to existing data selection methods?
- (**RQ2**): How does GRADATE + vanilla GNNs compare to model-centric GDA methods?
- (**RQ3**): To what extent can GRADATE enhance the effectiveness of model-centric GDA methods?

We will answer these research questions in Section 4.2, 4.3 and 4.4, correspondingly.

### 4.1 General Setup

In this section, we state the details of datasets and settings of GRADATE and baseline methods.

*Datasets and Graph Domains.* We consider graph classification tasks conducted on six real-world graph-level datasets, including IMDB-BINARY [69], IMDB-MULTI [69], MSRC_21 [45], `ogbg-molbace` [22], `ogbg-molbbbp` [22] and `ogbg-molhiv` [22]. The former three datasets are from the TUDataset [44]; while the latter three datasets are from the OGB benchmark [22]. We define the graph domains by *graph density* and *graph size*, which are the types of covariate shift that are widely studied in the literature [72, 41, 56, 5, 71, 10, 84]. Specifically, graphs are sorted by corresponding properties in an *ascending* order and split into train/val/test sets with ratios $60\%/20\%/20\%$. For brevity, we provide results on *graph density* shift in the main content. Additional experiments on *graph size* shift can be found in Appendix G. We also include the empirical cumulative distribution function (ECDF) plots of all settings in Appendix P to demonstrate the shift level.

*Details of* GRADATE *and Baselines.* To compute LinearFGW within GRADATE, we follow the default parameter settings in its github repository.[4] The trade-off parameter $\alpha$ is computed in $\{0.5, 0.9\}$[5] and the order $r$ is set to 2. The update step is fixed to $T = 10$ and the learning rate equal to $\eta = 10^{-4}$ across different settings. A popular model-free data valuation method is LAVA [26]. We apply LAVA for graph data selection and make the following modifications. We first leverage LinearFGW to form the pairwise distance matrix and compute GDD. LAVA then picks the smallest $k$ entries of the calibrated gradients as output. For the computation of GDD, we consider label signal $c \in \{0, 5\}$. We also incorporate KIDD-LR [66] as a model-centric but data-efficient baseline, which is a state-of-the-art graph dataset distillation method.

### 4.2 GRADATE as a Model-Free Graph Selector

To answer (**RQ1**), we compare GRADATE with other data-efficient methods, including (1) model-free techniques: random selection and LAVA [26] and (2) model-centric techniques: KIDD-LR [66].

*Experiment Setup.* To test the effectiveness of these selection methods, we fix the backbone GNN models in use to train the data selected by each method. Two popular GNN models are chosen, including GCN [30] and GIN [65] with default hyper-parameters following Zeng et al. [81] and the corresponding original papers. We also consider results on GAT [58] and GraphSAGE [18]. Please see Appendix G.1, G.2 and G.3 for more details. Here we consider selection ratio $\tau \in [0.1, 0.2, 0.5]$. More details on the architectures and training protocol can be found in Appendix I.

*Results.* As shown in Table 1, across all datasets, GRADATE outperforms the baseline methods under different selection ratios. It is also worth noting that even with few selected data, GRADATE

---

[4]https://github.com/haidnguyen0909/LinearFGW

[5]Since the datasets do not contain node features, we consider a larger $\alpha$ to place a greater emphasis on the structural properties.

| Dataset | Selection Method | GCN | | | | GIN | | | |
|---|---|---|---|---|---|---|---|---|---|
| | | $\tau=10\%$ | $\tau=20\%$ | $\tau=50\%$ | Full | $\tau=10\%$ | $\tau=20\%$ | $\tau=50\%$ | Full |
| IMDB-BINARY | Random | 0.737± 0.056 | 0.660± 0.012 | 0.868± 0.009 | 0.822±0.012 | 0.600± 0.019 | 0.710± 0.049 | 0.770± 0.053 | 0.783±0.031 |
| | KiDD-LR | 0.697± 0.041 | 0.787± 0.034 | 0.810± 0.022 | | 0.682± 0.013 | 0.772± 0.029 | 0.795± 0.014 | |
| | LAVA | 0.620± 0.000 | 0.620± 0.000 | 0.620± 0.000 | | 0.777± 0.019 | 0.795± 0.007 | 0.800± 0.007 | |
| | GraDate | **0.805**± 0.000 | **0.855**± 0.024 | **0.890**± 0.015 | | **0.800**± 0.008 | **0.832**± 0.002 | **0.900**± 0.013 | |
| IMDB-MULTI | Random | 0.139± 0.032 | 0.092± 0.032 | 0.080± 0.000 | 0.102±0.017 | 0.102± 0.015 | 0.180± 0.005 | 0.156± 0.057 | 0.143±0.056 |
| | KiDD-LR | 0.156± 0.022 | 0.154± 0.046 | 0.171± 0.052 | | 0.058± 0.044 | 0.093± 0.010 | 0.077± 0.025 | |
| | LAVA | 0.183± 0.000 | 0.183± 0.000 | 0.183± 0.000 | | **0.190**± 0.009 | 0.177± 0.019 | 0.193± 0.025 | |
| | GraDate | **0.588**± 0.286 | **0.349**± 0.323 | **0.611**± 0.242 | | 0.183± 0.073 | **0.266**± 0.133 | **0.361**± 0.162 | |
| MSRC_21 | Random | 0.576± 0.029 | 0.702± 0.045 | 0.830± 0.004 | 0.860±0.007 | 0.427± 0.035 | 0.801± 0.046 | 0.857± 0.011 | 0.883±0.015 |
| | KiDD-LR | 0.702± 0.007 | 0.766± 0.015 | 0.848± 0.004 | | 0.763± 0.025 | 0.792± 0.017 | 0.863± 0.015 | |
| | LAVA | 0.623± 0.007 | **0.819**± 0.012 | 0.895± 0.012 | | 0.667± 0.012 | 0.851± 0.007 | 0.933± 0.004 | |
| | GraDate | **0.719**± 0.007 | 0.797± 0.008 | **0.906**± 0.004 | | **0.787**± 0.046 | **0.860**± 0.007 | **0.942**± 0.008 | |
| ogbg-molbace | Random | 0.551± 0.100 | 0.375± 0.012 | 0.581± 0.039 | 0.617±0.073 | 0.637± 0.012 | 0.621± 0.027 | 0.537± 0.085 | 0.560±0.063 |
| | KiDD-LR | 0.592± 0.054 | 0.484± 0.020 | 0.592± 0.008 | | 0.613± 0.090 | 0.456± 0.041 | 0.589± 0.035 | |
| | LAVA | 0.627± 0.033 | **0.637**± 0.030 | **0.637**± 0.014 | | 0.602± 0.028 | 0.633± 0.048 | 0.672± 0.028 | |
| | GraDate | **0.655**± 0.046 | 0.578± 0.035 | 0.614± 0.042 | | **0.642**± 0.083 | **0.660**± 0.026 | **0.684**± 0.026 | |
| ogbg-molbbbp | Random | 0.567± 0.037 | 0.488± 0.088 | 0.478± 0.021 | 0.478±0.069 | 0.534± 0.084 | 0.648± 0.045 | 0.623± 0.019 | 0.671±0.034 |
| | KiDD-LR | 0.428± 0.025 | 0.477± 0.080 | 0.457± 0.013 | | 0.424± 0.005 | 0.450± 0.070 | 0.464± 0.052 | |
| | LAVA | 0.596± 0.058 | 0.566± 0.021 | 0.547± 0.044 | | 0.619± 0.044 | 0.642± 0.120 | **0.747**± 0.024 | |
| | GraDate | **0.604**± 0.065 | **0.601**± 0.047 | **0.557**± 0.001 | | **0.657**± 0.039 | **0.677**± 0.072 | 0.715± 0.015 | |
| ogbg-molhiv | Random | 0.603± 0.005 | **0.615**± 0.004 | 0.621± 0.001 | 0.625±0.001 | 0.608± 0.015 | 0.609± 0.030 | 0.593± 0.012 | 0.596±0.015 |
| | KiDD-LR | 0.590± 0.005 | 0.608± 0.001 | 0.595± 0.011 | | 0.597± 0.042 | 0.595± 0.039 | 0.608± 0.020 | |
| | LAVA | 0.531± 0.035 | 0.594± 0.013 | 0.601± 0.013 | | 0.614± 0.002 | 0.638± 0.020 | 0.641± 0.010 | |
| | GraDate | **0.607**± 0.018 | 0.599± 0.012 | **0.622**± 0.004 | | **0.640**± 0.013 | **0.651**± 0.022 | **0.658**± 0.018 | |

Table 1: Performance comparison across data selection methods for *graph density* shift. We use **bold**/underline to indicate the 1st/2nd best results. In most settings, GRADATE achieves the best performance across datasets.

can already achieve or excess GNN performance trained with full data, showing the importance of data quality in the source domain. The main reason is that because under distribution shift, a certain number of harmful graphs exist in training set. These samples can mislead the model and degrade generalization performance.

## 4.3 GRADATE as a GDA Method

We answer (**RQ2**) by directly compare the combination of GRADATE and vanilla GNNs (including non-domain-adapted GCN, GIN, GAT and GraphSAGE) with state-of-the-art GDA models. We fix the sparsity ratio $\tau$ to $20\%$ across all selection methods.

*Experiment Setup.* The four GDA methods we consider include AdaGCN [12], GRADE [61], ASN [83] and UDAGCN [62]. We conduct GDA experiments based on the codebase of OpenGDA [53]. We include more details of model-specific parameter settings in Appendix J. We set the training set as the source domain and the validation set as the target domain. For GDD computation, we set label signal $c = 0$ to match the requirement of unsupervised GDA methods. Results on *graph size* shift can be found in Appendix G.4.

*Results.* Results are in Table 2. For model-centric GDA methods trained with full training data, the severe domain shift prohibits these methods from learning rich knowledge to perform well on the test data in the target domain. Instead, GRADATE finds the most useful data in the training set that results in simple GNN models with extraordinary classification accuracy while maintaining data efficiency. In addition, similar to the observation in Section 4.2, GRADATE selects non-trivial training data that outperforms other model-free data selection methods. In Appendix

## 4.4 GRADATE as a Model-Free GDA Enhancer

In order to answer (**RQ3**), we combine GRADATE with off-the-shelf GDA methods to study whether fewer but better training data can lead to even stronger adaptation performance.

*Experiment Setup.* Coupled with $10\%, 20\%, 50\%$ data selected by each model-free method (i.e., random, LAVA and GRADATE), four GDA baselines (considered in Section 4.3) are directly run on the shrunk training dataset with the same validation dataset under *graph density* shift. For results on *graph size* shift, we refer the readers to Appendix G.5.

| Type | Model | Data | Dataset | | | | | |
|------|-------|------|---------|---|---|---|---|---|
| | | | IMDB-BINARY | IMDB-MULTI | MSRC_21 | ogbg-molbace | ogbg-molbbbp | ogbg-molhiv |
| GDA | AdaGCN | Full | $0.808 \pm 0.015$ | $0.073 \pm 0.000$ | $0.319 \pm 0.032$ | $0.607 \pm 0.068$ | $\mathbf{0.778} \pm 0.002$ | $0.428 \pm 0.011$ |
| | GRADE | Full | $0.822 \pm 0.012$ | $0.123 \pm 0.061$ | $0.804 \pm 0.011$ | $\mathbf{0.683} \pm 0.016$ | $0.489 \pm 0.005$ | $0.564 \pm 0.005$ |
| | ASN | Full | $0.782 \pm 0.030$ | $0.119 \pm 0.047$ | $0.833 \pm 0.033$ | $0.580 \pm 0.065$ | $0.476 \pm 0.027$ | $0.516 \pm 0.021$ |
| | UDAGCN | Full | $0.807 \pm 0.013$ | $0.114 \pm 0.049$ | $0.351 \pm 0.019$ | $0.541 \pm 0.034$ | $0.522 \pm 0.015$ | $0.451 \pm 0.030$ |
| Vanilla | GCN | Random 20% | $0.660 \pm 0.012$ | $0.092 \pm 0.032$ | $0.702 \pm 0.045$ | $0.529 \pm 0.124$ | $0.528 \pm 0.030$ | $0.598 \pm 0.003$ |
| | | LAVA 20% | $0.620 \pm 0.000$ | $0.092 \pm 0.032$ | $0.819 \pm 0.045$ | $0.541 \pm 0.067$ | $0.503 \pm 0.043$ | $0.591 \pm 0.030$ |
| | | GRADATE 20% | $0.830 \pm 0.021$ | $0.349 \pm 0.323$ | $0.797 \pm 0.008$ | $0.585 \pm 0.074$ | $0.571 \pm 0.035$ | $0.583 \pm 0.006$ |
| | GIN | Random 20% | $0.710 \pm 0.049$ | $0.180 \pm 0.005$ | $0.801 \pm 0.046$ | $0.622 \pm 0.028$ | $0.480 \pm 0.041$ | $0.590 \pm 0.033$ |
| | | LAVA 20% | $0.778 \pm 0.045$ | $0.170 \pm 0.009$ | $\underline{0.851} \pm 0.012$ | $0.655 \pm 0.067$ | $0.644 \pm 0.021$ | $\underline{0.638} \pm 0.012$ |
| | | GRADATE 20% | $0.832 \pm 0.025$ | $0.266 \pm 0.133$ | $\mathbf{0.860} \pm 0.007$ | $\underline{0.662} \pm 0.006$ | $\underline{0.665} \pm 0.053$ | $\mathbf{0.644} \pm 0.017$ |
| | GAT | Random 20% | $0.662 \pm 0.029$ | $0.067 \pm 0.005$ | $0.713 \pm 0.008$ | $0.472 \pm 0.034$ | $0.486 \pm 0.041$ | $0.593 \pm 0.012$ |
| | | LAVA 20% | $0.835 \pm 0.002$ | $\underline{0.790} \pm 0.002$ | $0.842 \pm 0.026$ | $0.515 \pm 0.019$ | $0.511 \pm 0.069$ | $0.602 \pm 0.017$ |
| | | GRADATE 20% | $\mathbf{0.858} \pm 0.005$ | $\mathbf{0.800} \pm 0.133$ | $\underline{0.857} \pm 0.008$ | $0.518 \pm 0.026$ | $0.538 \pm 0.098$ | $0.598 \pm 0.004$ |
| | GraphSAGE | Random 20% | $0.738 \pm 0.059$ | $0.132 \pm 0.036$ | $0.731 \pm 0.027$ | $0.459 \pm 0.057$ | $0.472 \pm 0.016$ | $0.602 \pm 0.006$ |
| | | LAVA 20% | $0.835 \pm 0.005$ | $0.570 \pm 0.292$ | $0.827 \pm 0.015$ | $0.514 \pm 0.132$ | $0.491 \pm 0.095$ | $0.537 \pm 0.067$ |
| | | GRADATE 20% | $\underline{0.855} \pm 0.005$ | $0.580 \pm 0.281$ | $0.842 \pm 0.007$ | $0.536 \pm 0.062$ | $0.533 \pm 0.037$ | $0.541 \pm 0.014$ |

Table 2: Performance comparison across GDA and vanilla methods for *graph density* shift. We use **bold**/underline to indicate the 1st/2nd best results. GRADATE can consistently achieve top-2 performance across all datasets.

*Results.* As shown in Table 3, for most of the settings, GRADATE selects data that is the most beneficial to adapting to the target set. Notably, across many settings, only $10\%$ or $20\%$ GRADATE-selected data can outperform naively applying GDA methods on the full training data. This suggests that GRADATE can indeed improve *data-efficiency* by promoting the quality of training data. Furthermore, by effective data selection performed by GRADATE, the difficulty of addressing the domain shift can be lowered significantly and thus result in better adaptation performance.

### 4.5 Further Discussion

*LAVA vs GRADATE.* The modified version of LAVA utilizes LinearFGW to compare graphs and selects the training data with the smallest gradient value w.r.t. GDD. In contrast, GRADATE aims at finding optimal training data that directly minimizes GDD, which has a complete different motivation and enjoys a theoretical justification. Empirically, we also observe the superiority of GRADATE in most cases. Occasionally, LAVA achieve marginally better results than GRADATE, which may be attributed to the approximation error of LinearFGW and thus over-optimization on GDD.

*Random vs GRADATE.* From Table 2, we found GRADATE occasionally underperforms random selection with GraphSAGE, possibly because the neighbor sampling strategy introduces noise into global representations, weakening the supervision signal even for well-chosen training graphs.

*Selection Ratio vs GNN Performance.* From Tables 1 & 3, we find that a larger selection ratio may not always guarantee a better performance for selection methods including LAVA and GRADATE. This is because, under severe distribution shift between domains, a larger portion of training data may actually contain patterns that are irrelevant or even harmful to the target domain.

*Effect of label signal $c$.* While we treat $c$ as a tunable hyper-parameter that can be optimized for different settings (i.e. various combinations of dataset, shift types and selection ratio), we empirically find that searching within $\{0, 5\}$ can already lead to good performance throughout all experiments in this paper. We also provide additional experiments under a label-free setting (i.e. $c$ is forced to be equal to 0) in Appendix G.6 .

## 5 Related Work

*Data Selection.* Recent advancements in data selection have focused on optimizing data utilization, mainly on text and vision data to facilitate efficient training for large language/image models [29, 28, 43, 3, 63, 13, 70, 35]. For general model-free data selection, LAVA [26] offers a learning-agnostic data valuation method by seeking the data point that contributes the most to the distance between training and validation datasets. However, the paper studies predominantly on raw image datasets such

| Dataset | GDA Method → | AdaGCN | | | | GRADE | | | |
|---|---|---|---|---|---|---|---|---|---|
| | Selection Method ↓ | $\tau=10\%$ | $\tau=20\%$ | $\tau=50\%$ | Full | $\tau=10\%$ | $\tau=20\%$ | $\tau=50\%$ | Full |
| IMDB-BINARY | Random | 0.763±0.040 | 0.773±0.019 | 0.798±0.002 | 0.808±0.015 | 0.683±0.010 | 0.792±0.002 | 0.780±0.015 | 0.822±0.012 |
| | LAVA | 0.623±0.005 | 0.617±0.005 | 0.617±0.005 | | 0.620±0.073 | 0.627±0.009 | 0.680±0.047 | |
| | GRADATE | **0.810±0.032** | **0.817±0.024** | **0.822±0.017** | | **0.782±0.009** | **0.832±0.013** | **0.848±0.009** | |
| IMDB-MULTI | Random | 0.100±0.000 | 0.168±0.072 | 0.116±0.048 | 0.073±0.000 | 0.106±0.055 | 0.112±0.050 | 0.149±0.049 | 0.123±0.061 |
| | LAVA | 0.191±0.007 | 0.183±0.000 | 0.184±0.002 | | **0.183±0.000** | 0.189±0.008 | **0.186±0.003** | |
| | GRADATE | **0.333±0.229** | **0.373±0.285** | **0.391±0.294** | | 0.131±0.074 | **0.386±0.286** | 0.173±0.100 | |
| MSRC_21 | Random | 0.208±0.027 | 0.374±0.011 | 0.307±0.087 | 0.319±0.032 | 0.512±0.041 | 0.626±0.055 | 0.708±0.023 | 0.804±0.011 |
| | LAVA | 0.398±0.004 | **0.456±0.012** | 0.480±0.061 | | 0.608±0.018 | 0.743±0.021 | 0.860±0.014 | |
| | GRADATE | **0.415±0.112** | 0.406±0.043 | **0.532±0.039** | | **0.664±0.021** | **0.778±0.015** | **0.865±0.027** | |
| ogbg-molbace | Random | 0.436±0.021 | 0.485±0.038 | 0.565±0.085 | 0.607±0.068 | 0.538±0.023 | 0.554±0.025 | 0.611±0.015 | 0.683±0.016 |
| | LAVA | 0.574±0.017 | 0.589±0.074 | **0.607±0.071** | | 0.557±0.055 | **0.653±0.054** | 0.625±0.015 | |
| | GRADATE | **0.598±0.066** | **0.614±0.043** | 0.572±0.047 | | **0.599±0.044** | 0.636±0.035 | **0.634±0.006** | |
| ogbg-molbbbp | Random | 0.494±0.014 | 0.469±0.031 | 0.527±0.035 | 0.778±0.002 | 0.511±0.032 | 0.433±0.001 | 0.495±0.041 | 0.489±0.005 |
| | LAVA | 0.583±0.075 | 0.556±0.015 | **0.561±0.040** | | 0.549±0.013 | **0.579±0.041** | **0.543±0.013** | |
| | GRADATE | **0.593±0.038** | **0.596±0.022** | 0.546±0.026 | | **0.582±0.077** | 0.503±0.012 | 0.490±0.006 | |
| ogbg-molhiv | Random | 0.407±0.022 | 0.429±0.032 | 0.417±0.013 | 0.428±0.011 | 0.581±0.008 | 0.544±0.001 | 0.581±0.009 | 0.564±0.005 |
| | LAVA | 0.453±0.016 | 0.428±0.013 | 0.440±0.003 | | 0.566±0.011 | 0.571±0.005 | 0.572±0.019 | |
| | GRADATE | **0.463±0.041** | **0.473±0.021** | **0.447±0.038** | | **0.584±0.012** | **0.589±0.003** | **0.586±0.003** | |

| Dataset | GDA Method → | ASN | | | | UDAGCN | | | |
|---|---|---|---|---|---|---|---|---|---|
| | Selection Method ↓ | $\tau=10\%$ | $\tau=20\%$ | $\tau=50\%$ | Full | $\tau=10\%$ | $\tau=20\%$ | $\tau=50\%$ | Full |
| IMDB-BINARY | Random | 0.660±0.043 | 0.707±0.017 | 0.678±0.031 | 0.782±0.030 | 0.620±0.041 | 0.763±0.008 | 0.823±0.005 | 0.807±0.013 |
| | LAVA | 0.733±0.081 | 0.620±0.000 | 0.620±0.000 | | 0.620±0.000 | 0.643±0.033 | 0.620±0.000 | |
| | GRADATE | **0.748±0.037** | **0.818±0.016** | **0.855±0.011** | | **0.770±0.023** | **0.847±0.012** | **0.852±0.005** | |
| IMDB-MULTI | Random | 0.126±0.013 | 0.101±0.058 | 0.156±0.039 | 0.119±0.047 | 0.150±0.024 | 0.101±0.045 | 0.076±0.003 | 0.114±0.049 |
| | LAVA | 0.183±0.000 | 0.183±0.000 | 0.190±0.009 | | **0.183±0.000** | 0.183±0.000 | 0.182±0.002 | |
| | GRADATE | **0.292±0.352** | **0.588±0.286** | **0.381±0.301** | | 0.093±0.066 | **0.554±0.263** | **0.339±0.337** | |
| MSRC_21 | Random | 0.421±0.026 | 0.673±0.011 | 0.661±0.032 | 0.833±0.033 | 0.287±0.018 | 0.178±0.039 | 0.287±0.075 | 0.351±0.019 |
| | LAVA | 0.635±0.015 | 0.746±0.019 | 0.868±0.014 | | **0.453±0.035** | 0.447±0.052 | 0.623±0.059 | |
| | GRADATE | **0.687±0.048** | **0.804±0.021** | **0.904±0.012** | | 0.444±0.048 | **0.453±0.011** | **0.664±0.029** | |
| ogbg-molbace | Random | 0.539±0.074 | **0.637±0.009** | 0.507±0.061 | 0.580±0.065 | 0.478±0.037 | **0.581±0.018** | 0.513±0.028 | 0.541±0.034 |
| | LAVA | 0.578±0.036 | 0.603±0.009 | 0.646±0.050 | | 0.562±0.039 | 0.578±0.015 | 0.513±0.077 | |
| | GRADATE | **0.636±0.022** | 0.596±0.053 | **0.651±0.036** | | 0.533±0.041 | 0.565±0.039 | **0.531±0.051** | |
| ogbg-molbbbp | Random | 0.504±0.015 | 0.533±0.025 | 0.497±0.032 | 0.476±0.027 | 0.538±0.026 | 0.529±0.040 | 0.530±0.051 | 0.522±0.015 |
| | LAVA | 0.567±0.040 | **0.616±0.072** | **0.573±0.035** | | 0.579±0.031 | 0.547±0.021 | 0.558±0.021 | |
| | GRADATE | **0.573±0.088** | 0.596±0.100 | 0.535±0.027 | | **0.591±0.040** | **0.575±0.030** | **0.570±0.009** | |
| ogbg-molhiv | Random | 0.436±0.038 | 0.483±0.044 | 0.455±0.059 | 0.516±0.021 | 0.453±0.015 | 0.406±0.015 | **0.464±0.024** | 0.451±0.030 |
| | LAVA | 0.511±0.018 | **0.540±0.010** | 0.482±0.023 | | 0.458±0.029 | 0.427±0.007 | 0.445±0.018 | |
| | GRADATE | **0.527±0.041** | 0.491±0.080 | **0.491±0.050** | | 0.453±0.011 | **0.445±0.018** | 0.444±0.020 | |

Table 3: Performance comparison across combinations of GDA methods and data selection methods for *graph density* shift. We use **bold**/underline to indicate the 1st/2nd best results. GRADATE achieves the best performance in most settings.

as CIFAR-10 [31] and MNIST [32], where they already have high-quality pixel-value representations for computation. Unlike text or images, graphs lack a natural and uniform representation, making the development of model-free data selection more intricate. Tailored for graph-level tasks, graph dataset distillation is also a related topic. For example, Jin et al. [25] and Xu et al. [66] both propose to formulate a bi-level optimization problem to train a graph-level classifier. Jain et al. [24], on the other hand, utilizes Tree Mover Distance [11] to conduct graph-level sub-sampling with theoretical guarantees. However, these non-model-free methods might not be able to combat severe downstream distribution changes.

*Graph Domain Adaptation (GDA).* For grpah classification, GDA focuses on transferring knowledge from a source domain with labeled graph to a target domain. Model-centric GDA methods relying on GNNs have been pivotal in this area. For instance, Wu et al. [62] introduce UDAGCN, which integrates domain adaptation with GNNs to align feature distributions between domains. AdaGCN [12] addresses cross-network node classification leveraging adversarial domain adaptation to transfer label information between domains. Wu et al. [61] explore cross-network transfer learning through Weisfeiler-Lehman graph isomorphism test and introduce the GRADE algorithm that minimizes distribution shift to perform adaptation. ASN [83] explicitly separates domain-private and domain-shared information while capturing network consistency. More recently, Liu et al. [34] argue that excessive message passing exacerbates domain bias and propose A2GNN as a refined propagation scheme that disentangles transferable and domain-specific information; Chen et al. [7]

highlight the critical role of graph smoothness, presenting TDSS that enforces cross-domain consistency through spectral alignment. Meanwhile, Liu et al. [36] introduce a pairwise alignment strategy that leverages node-level relational matching to enhance inter-domain correspondence. However, these approaches mostly focus on designing architectures or training procedures and often rely heavily on the assumption that provided data in the training set is already optimal for the task, which is often invalid in real-world scenarios.

## 6 Conclusion

We introduce GRADATE, a model-free framework for graph classification that addresses distribution shift by solving a Graph Dataset Distance (GDD) minimization problem. By selecting the most beneficial data from the source domain, it offers a novel approach to improving GNN performance without relying on specific model predictions or training procedures. We also establish theoretical analysis on Fused Gromov–Wasserstein distance as a meaningful upper bound on GNN representation differences, and further justifies GDD as an optimization target to improve generalization performance. Across multiple real-world datasets and shift types, GRADATE consistently outperforms existing selection methods and GDA methods with better data efficiency. For future directions, we consider graph continual learning and multi-source domain adaptation.

## Acknowledgement

This work is supported by NSF (2416070) and AFOSR (FA9550-24-1-0002). The content of the information in this document does not necessarily reflect the position or the policy of the Government, and no official endorsement should be inferred. The U.S. Government is authorized to reproduce and distribute reprints for Government purposes notwithstanding any copyright notation here on.

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

# Appendix

The content of appendix is organized as follows:

1. **Algorithms**:
   - Appendix A talks about the details of LinearFGW [46] that we omit in the main text. We summarize the overall procedure of LinearFGW in Algorithm 1.
   - Appendix B goes through the steps to compute Graph Dataset Distance (GDD). The entire procedure is included in Algorithm 2.
   - Appendix C summarizes the submodule GREAT used in our main algorithm (Algorithm 3).
   - Appendix D summarizes our main algorithm GRADATE (Algorithm 4).
2. **Proofs**:
   - Appendix F provides the proofs for all the theorems in the main text.
3. **Additional Experiments**:
   - Appendix G.1 compares GRADATE with other data selection methods under *graph size* shift with additional GNN backbones.
   - Appendix G.2 compares GRADATE with other data selection methods under *graph density* shift.
   - Appendix G.3 compares GRADATE with other data selection methods under *graph size* shift with additional GNN backbones.
   - Appendix G.4 compares the combination of GRADATE and vanilla GNNs with other GDA methods under *graph size* shift.
   - Appendix G.5 compares the combination of GDA methods and GRADATE against other data selection methods under *graph size* shift.
   - Appendix G.6 ablates on the validation-label-free setting.
   - Appendix G.7 includes results on additional graph backbones.
   - Appendix G.8 includes results on additional GDA methods.
4. **Discussions**:
   - Appendix E discusses FGW and Graph Dataset Distance (GDD) in relation to prior notions such as Tree-Mover Distance (TMD) [11] and Maximum Mean Discrepancy (MMD) [17].
   - Appendix N discusses potential limitations and future direction of our work.
5. **Reproducibility**:
   - Appendix H provides the dataset statistics and licenses used in this work.
   - Appendix I introduces the overall settings of GNN to use for the graph data selection evaluation, including the model we select and the training protocols.
   - Appendix J includes GDA method-specific parameter settings, where we follow the default settings of the OpenGDA package [53].
6. **Others**:
   - Appendix K includes problem definition of Graph Domain Adaptation (GDA).
   - Appendix L provides additional related work.
   - Appendix M contains the empirical runtime of GRADATE.
   - Appendix P includes the ECDF plots of graph properties across datasets.

## A   Details of LinearFGW (Algorithm 1)

Formally, consider a set of $N$ graphs $\mathcal{D} = \{\mathcal{G}_i\}_{i=1}^N$, where each $\mathcal{G} = (\mathbf{A}, \mathbf{X}) \in \mathcal{D}$ represents an attributed graph with adjacency matrix $\mathbf{A} \in \mathbb{R}^{n \times n}$ and node feature matrix $\mathbf{X} \in \mathbb{R}^{n \times d}$. Note that $n$ is the number of nodes of $\mathcal{G}$ and $d$ is the dimension of node features. LinearFGW first requires a reference graph $\overline{\mathcal{G}} = (\overline{\mathbf{A}}, \overline{\mathbf{X}})$ where $\overline{\mathbf{A}} \in \mathbb{R}^{\bar{n} \times \bar{n}}$, $\overline{\mathbf{X}} \in \mathbb{R}^{\bar{n} \times d}$, $\bar{n}$ is the number of nodes and $d$ is the dimension of node features. Typically, $\overline{\mathcal{G}}$ is obtained by solving an *FGW barycenter problem*, which aims to find a "center" graph that has the minimum sum of pairwise graph distances over the entire graph set $\mathcal{D}$.

Following the notation used in Section 2.2, we define the inter-graph distance matrix $\mathbf{M}_{(\mathcal{G}_1, \mathcal{G}_2)}$ between any pair of graphs (named as $\mathcal{G}_1 = \{\mathbf{A}_1, \mathbf{X}_1, \mathbf{p}_1\}$ and $\mathcal{G}_2 = \{\mathbf{A}_2, \mathbf{X}_2, \mathbf{p}_2\}$) to be the pairwise Euclidean distance of node features. Namely, $\mathbf{M}_{(\mathcal{G}_1, \mathcal{G}_2)} = [\|\mathbf{X}_1[i] - \mathbf{X}_2[j]\|]_{ij}$. In addition, the intra-graph similarity matrix is chosen to be defined as their corresponding adjacency matrices

(i.e., $\mathbf{C}_{\mathcal{G}_1} = \mathbf{A}_1$ and $\mathbf{C}_{\mathcal{G}_2} = \mathbf{A}_2$). Together with uniform distributions[6] $\mathbf{p}_{\mathcal{G}_1} = \frac{\mathbf{1}_{n_1}}{n_1}$ and $\mathbf{p}_{\mathcal{G}_2} = \frac{\mathbf{1}_{n_2}}{n_2}$ over the nodes of $\mathcal{G}_1$ and $\mathcal{G}_2$ (with sizes $n_1$ and $n_2$), correspondingly, the *FGW barycenter problem*[7] can be formulated as follows:

$$\overline{\mathcal{G}} = \arg\min_{\mathcal{G}} \sum_{i=1}^{N} \text{FGW}(\mathbf{M}_{(\mathcal{G}_i, \mathcal{G})}, \mathbf{C}_{\mathcal{G}_i}, \mathbf{C}_{\mathcal{G}}, \mathbf{p}_{\mathcal{G}_i}, \mathbf{p}_{\mathcal{G}}, \alpha), \tag{6}$$

where $\alpha \in [0, 1]$ is the pre-defined trade-off parameter.

After calculating the reference graph $\overline{\mathcal{G}}$, we then obtain $N$ optimal transport plans $\{\boldsymbol{\pi}_i\}_{i\in[n]}$ as the solutions by computing $\text{FGW}(\mathcal{G}, \overline{\mathcal{G}})$ for each $\mathcal{G} \in \mathcal{D}$ (via solving Equation (1)). Then, the barycentric projection [46] of each graph's node edge with respect to $\overline{\mathcal{G}}$ can be written as

$$\mathbf{T}_{\text{node}}(\boldsymbol{\pi}_i) = \bar{n} \cdot \boldsymbol{\pi}_i \mathbf{X}_i \in \mathbb{R}^{\bar{n} \times d}, \tag{7}$$

$$\mathbf{T}_{\text{edge}}(\boldsymbol{\pi}_i) = \bar{n}^2 \cdot \boldsymbol{\pi}_i \mathbf{C}_i \boldsymbol{\pi}_i^\top \in \mathbb{R}^{\bar{n} \times \bar{n}}. \tag{8}$$

Finally, we can define the LinearFGW distance based on these barycentric projections. Namely, for any pair of graphs $(\mathcal{G}_i, \mathcal{G}_j)$, we define a distance metric $d_{\text{LinearFGW}}(\cdot, \cdot)$ over the graph set $\mathcal{D}$:

$$\begin{aligned} d_{\text{LinearFGW}}(\mathcal{G}_i, \mathcal{G}_j) = \\ (1 - \alpha)\|\mathbf{T}_{\text{node}}(\boldsymbol{\pi}_i) - \mathbf{T}_{\text{node}}(\boldsymbol{\pi}_j)\|_F^2 \\ + \alpha\|\mathbf{T}_{\text{edge}}(\boldsymbol{\pi}_i) - \mathbf{T}_{\text{edge}}(\boldsymbol{\pi}_j)\|_F^2, \end{aligned} \tag{9}$$

for $i, j \in [N]$. Note that $\|\cdot\|_F$ represents the Frobenius norm.

---

**Algorithm 1** LinearFGW [46]

---

1: **Input:** $N$ graphs $\mathcal{D} = \{\mathcal{G}_i\}_{i=1}^N$, trade-off parameter $\alpha$.
2: Initialize pairwise distance matrix $\mathbf{D} \in \mathbb{R}^{N \times N}$
3: Solve the FGW barycenter problem in Equation (6) and obtain the reference graph $\overline{\mathcal{G}}$;
4: **for** graph $\mathcal{G}_i$ **in** $\mathcal{D}$ **do**
5:     Compute $\text{FGW}(\mathcal{G}_i, \overline{\mathcal{G}})$ via solving Equation (1) and obtain $\boldsymbol{\pi}_i$;
6:     Compute $\mathbf{T}_{\text{node}}(\boldsymbol{\pi}_i)$ and $\mathbf{T}_{\text{edge}}(\boldsymbol{\pi}_i)$ via Equation (7)(8);
7: **end for**
8: **for** $\mathcal{G}_i$ **in** $\mathcal{D}$ **do**
9:     **for** $\mathcal{G}_j$ **in** $\mathcal{D}$ **do**
10:         Compute $\mathbf{D}[i, j] = d_{\text{LinearFGW}}(\mathcal{G}_i, \mathcal{G}_j)$ via Equation (9);
11:     **end for**
12: **end for**
13: **return** LinearFGW pairwise distance matrix $\mathbf{D}$.

---

## B  Summarization of GDD (Algorithm 2)

---

[6]Since we have no prior over the node importance in either graphs, the probability simplex will typically be set as uniform.

[7]The optimization algorithm for solving this problem is omitted. Please refer to the original paper for more details.

---

**Algorithm 2** (Training-Validation) GDD Computation

---

1: **Input:** labeled training graphs $\mathcal{D}^{\text{train}} = \{\mathcal{G}_i^{\text{train}}, y_i^{\text{train}}\}_{i=1}^n$, labeled validation graphs $\mathcal{D}^{\text{val}} = \{\mathcal{G}_i^{\text{val}}, y_i^{\text{val}}\}_{i=1}^m$, trade-off parameter $\alpha$, label signal strength $c \geq 0$, a shared label set $\mathcal{Y}$.
2: Compute pairwise LinearFGW distance matrix $\mathbf{D} \in \mathbb{R}^{n \times m}$ via Algorithm 1 with the graph set $\mathcal{D} = \mathcal{D}^{\text{train}} \cup \mathcal{D}^{\text{val}}$ and parameter $\alpha$;
3: Initialize new pairwise distance matrix $\tilde{\mathbf{D}} = \mathbf{D}$;
4: Initialize uniform empirical measures:
   $\mathbf{p}^{\text{train}} = \frac{1}{n} \sum_{i \in [n]} \delta_{(\mathcal{G}_i^{\text{train}}, y_i^{\text{train}})}, \mathbf{q}^{\text{val}} = \frac{1}{m} \sum_{j \in [m]} \delta_{(\mathcal{G}_j^{\text{val}}, y_j^{\text{val}})}$;
5: **for** training label $\ell_t$ **in** $\mathcal{Y}$ **do**
6:     **for** validation label $\ell_v$ **in** $\mathcal{Y}$ **do**
7:         Collect training index set with label $\ell_t$:
   $\mathbf{I}_{\ell_t} = \{i | y_i^{\text{train}} = \ell_t\}$;
8:         Collect validation data set with label $\ell_v$:
   $\mathbf{I}_{\ell_v} = \{j | y_j^{\text{val}} = \ell_v\}$;
9:         Compute *graph-label distance* in Equation (2):
   $d(\ell_t, \ell_v) = \text{OT}(\mathbf{p}_{\ell_t}^{\text{train}}, \mathbf{q}_{\ell_v}^{\text{val}}, d_{\text{LinearFGW}})$;
10:        Update distance sub-matrix $\tilde{\mathbf{D}}[i \in \mathbf{I}_{\ell_t}, j \in \mathbf{I}_{\ell_v}] = \mathbf{D}[i \in \mathbf{I}_{\ell_t}, j \in \mathbf{I}_{\ell_v}] + c \cdot d(\ell_t, \ell_v)$;
11:     **end for**
12: **end for**
13: Compute $\text{OTDD}(\mathcal{D}^{\text{train}}, \mathcal{D}^{\text{val}}) = \text{OT}(\mathbf{p}^{\text{train}}, \mathbf{q}^{\text{val}}, \tilde{\mathbf{D}})$ via the equation in Section 2.1.
14: **return** $\text{GDD}(\mathcal{D}^{\text{train}}, \mathcal{D}^{\text{val}}) = \text{OTDD}(\mathcal{D}^{\text{train}}, \mathcal{D}^{\text{val}})$.

---

## C   Summarization of GREAT (Algorithm 3)

Starting from a uniform training weight $\mathbf{w}$, GREAT alternates between two subroutines: (i) computes GDD between the two sets using pairwise distances $\tilde{\mathbf{D}} \in \mathbb{R}^{n \times m}$ as the cost matrix (Line 4) and obtains the gradient $\mathbf{g_w} = \nabla_{\mathbf{w}} \text{GDD}(\mathbf{p}^{\text{train}}(\mathbf{w}), \mathbf{q}^{\text{val}}, \tilde{\mathbf{D}})$ for updating $\mathbf{w}$ (Line 5) and (ii) gradually sparsifies $\mathbf{w}$ by retaining only the top-$k$ entries followed by normalization to ensure $\mathbf{w}$ is on the probability simplex (Line 6-9). After $T$ iterations, we extract the non-zero entries from the resulting $\mathbf{w}$ and name this training index set as $\mathbf{S}$.

---

**Algorithm 3** GREAT

---

1: **Input:** pairwise LinearFGW distance matrix $\tilde{\mathbf{D}} \in \mathbb{R}^{n \times m}$, selection ratio $\tau$, update step $T$, learning rate $\eta$.
2: Initialize uniform training weights: $\mathbf{w} = \frac{\mathbf{1}_n}{n}$;
3: **for** $t = 1$ **to** $T - 1$ **do**
4:     Compute $\text{GDD}(\mathbf{p}^{\text{train}}(\mathbf{w}), \mathbf{q}^{\text{val}}, \tilde{\mathbf{D}})$ via Algorithm 2;
5:     Compute $\mathbf{g_w} = \nabla_{\mathbf{w}} \text{GDD}(\mathbf{p}^{\text{train}}(\mathbf{w}), \mathbf{q}^{\text{val}}, \tilde{\mathbf{D}})$ via Theorem F.3;
6:     Compute current sparsity level:
   $k = n \cdot \max(\tau, \frac{T-t+1}{T-1} + \frac{\tau t}{T-1})$;
7:     Update data weight: $\mathbf{w} = \max(\mathbf{w} - \eta \cdot \mathbf{g_w}, \mathbf{0})$;
8:     Sparsify data weight: $\mathbf{w} = \mathbf{w} \odot \text{Top-}k(\mathbf{w})$;
9:     Apply $\ell_1$-normalization: $\mathbf{w} = \mathbf{w}/\|\mathbf{w}\|_1$;
10: **end for**
11: **return** training data index set $\mathbf{S} = \text{nonzero}(\mathbf{w})$.

---

## D   Summarization of GRADATE (Algorithm 4)

## E   Discussions on FGW & GDD and Previous Measures

*Comparison between FGW [57] and TMD [11].* Specifically, FGW has the following advantages over Tree Mover Distance (TMD) [11]. Firstly, Linear optimal transport theory [59, 46] can be

**Algorithm 4** GRADATE

1: **Input:** labeled training graphs $\mathcal{D}^{\text{train}} = \{\mathcal{G}_i^{\text{train}}, y_i^{\text{train}}\}_{i=1}^n$, labeled validation graphs $\mathcal{D}^{\text{val}} = \{\mathcal{G}_i^{\text{val}}, y_i^{\text{val}}\}_{i=1}^m$, trade-off parameter $\alpha$, label signal strength $c \geq 0$, selection ratio $\tau$, update step $T$, learning rate $\eta$.
2: Compute pairwise LinearFGW distance matrix $\mathbf{D} \in \mathbb{R}^{n \times m}$ via Algorithm 1 with the graph set $\mathcal{D}^{\text{train}}, \mathcal{D}^{\text{val}}$ and parameter $\alpha$;
3: Compute the (label-informed) pairwise distance matrix $\tilde{\mathbf{D}}$ with label signal $c$ (line 3-12) in Algorithm 2;
4: Compute $\mathbf{S} = \text{GREAT}(\tilde{\mathbf{D}}, \tau, T, \eta)$;
5: **return** selected training data index set $\mathbf{S}$.

utilized to bring down the costs for pairwise FGW distance computation while TMD does not have similar technique. Secondly, a single-pair FGW computation (with time complexity $\mathcal{O}(|\mathcal{V}|^3)$) is cheaper than a single-pair TMD computation (with time complexity $\mathcal{O}(\mathcal{L}|\mathcal{V}|^4)$), where $\mathcal{V}$ is graph size and $\mathcal{L}$ is the depth of TMD. While cheaper, FGW can achieve similar theoretical results as TMD.

*Comparison between GDD and MMD [17].* GDD offers a more flexible and expressive notion of graph dataset similarity than Max Mean Discrepency (MMD) [17], which solely compares aggregated graph embeddings. To be more specific, GDD has the following three advantages over MMD. Firstly, unlike MMD, which often depends on model-specific representations (such as pre-trained encoder) or require training, GDD does not involve training and is model-free. This makes it broadly applicable across various graph-level datasets without the need for task-specific models. Secondly, GDD can optionally incorporate *auxiliary label information* (when available), enabling more fine-grained and task-relevant comparisons between data distributions in classification settings. Finally, GDD is based on Wasserstein distance. Although not explicitly stated in our paper, this results in interpretable correspondences between data points across datasets that can directly be used for data selection or data re-weighting algorithms for domain adaptation applications.

# F    Proofs of Theorems

## F.1    Proof of Theorem 3.1

In this section, we prove Theorem 3.1. We first focus on a simplified case with $k = 1$, which implies that the underlying GNN has only one layer. Then, based on this result, we use induction to generalize the conclusion to any positive $k$, which represents multi-layer GNNs.

### F.1.1    Assumptions

With a slight abuse of notation, let a graph $\mathcal{G}$ denote its node set as well. We assume that $f$ only uses one-hop information followed by a linear transformation. Specifically, for any graph $\mathcal{G} = (\mathbf{A}, \mathbf{X})$ and any node $u \in \mathcal{G}$, the output $f(\mathcal{G})_u$ depends only on the local neighborhood of $u$, defined as $\mathcal{N}_{\mathcal{G}}(u) := \{\mathbf{A}[u, v], \mathbf{X}[v]\}_{v \in \mathcal{G}}$. This localized aggregation is first computed by a convolution function $g$, and the result is then passed through a linear transformation with weights $\mathbf{W}$ and bias $\mathbf{b}$, giving:

$$f(\mathcal{G})_u = \mathbf{W} \cdot g(\mathcal{N}_{\mathcal{G}}(u)) + \mathbf{b} = \mathbf{W} \cdot g\left(\{\mathbf{A}[u, v], \mathbf{X}[v]\}_{v \in \mathcal{G}}\right) + \mathbf{b}. \tag{10}$$

We assume the convolution function $g$ is $C_W$-Lipschitz w.r.t. the following FGW distance $d_{W;\alpha}$: for any nodes $u_1 \in \mathcal{G}_1$ and $u_2 \in \mathcal{G}_2$,

$$d_{W;\alpha}(\mathcal{N}_{\mathcal{G}_1}(u_1), \mathcal{N}_{\mathcal{G}_2}(u_2)) \tag{11}$$

$$:= \left(\inf_{\pi \in \Pi(\mu_1, \mu_2)} \mathbb{E}_{(v_1, v_2) \sim \pi}[(1 - \alpha)\|\mathbf{X}_1[v_1] - \mathbf{X}_2[v_2]\|^r + \alpha|\mathbf{A}_1[u_1, v_1] - \mathbf{A}_2[u_2, v_2]|^r]\right)^{1/r}, \tag{12}$$

where we use $\mu_1 := \text{Unif}(\mathcal{G}_1)$ and $\mu_2 := \text{Unif}(\mathcal{G}_2)$ in this work.

### F.1.2   Proof for $k = 1$

*Proof.* Let $\mu_1 := \mathsf{Unif}(\mathcal{G}_1)$, $\mu_2 := \mathsf{Unif}(\mathcal{G}_2)$.

For any coupling $\pi \in \Pi(\mu_1, \mu_2)$, by Jensen's inequality w.r.t. the concave function $x \mapsto x^{1/r}$,

$$\mathop{\mathbb{E}}_{(u_1, u_2) \sim \pi} [\|f(\mathcal{G}_1)_{u_1} - f(\mathcal{G}_2)_{u_2}\|] \tag{13}$$

$$= \mathop{\mathbb{E}}_{(u_1, u_2) \sim \pi} \Big[ \Big\| \big[ \mathbf{W} \cdot g(\{(\mathbf{A}_1[u_1, v_1], \mathbf{X}_1[v_1])\}_{v_1 \in \mathcal{G}_1}) + \mathbf{b} \big] - \big[ \mathbf{W} \cdot g(\{(\mathbf{A}_2[u_2, v_2], \mathbf{X}_1[v_2])\}_{v_2 \in \mathcal{G}_2}) + \mathbf{b} \big] \Big\| \Big] \tag{14}$$

$$\leq \mathop{\mathbb{E}}_{(u_1, u_2) \sim \pi} \Big[ C_W \|\mathbf{W}\| \cdot d_{W;\alpha}(\{(\mathbf{A}_1[u_1, v_1], \mathbf{X}_1[v_1])\}_{v_1 \in \mathcal{G}_1}, \{(\mathbf{A}_2[u_2, v_2], \mathbf{X}_2[v_2])\}_{v_2 \in \mathcal{G}_2}) \Big] \tag{15}$$

$$= C_W \|\mathbf{W}\| \cdot \mathop{\mathbb{E}}_{(u_1, u_2) \sim \pi} \Big[ \Big( \inf_{\pi' \in \Pi(\mu_1, \mu_2)} \mathop{\mathbb{E}}_{(v_1, v_2) \sim \pi'} [(1 - \alpha) \|\mathbf{X}_1[v_1] - \mathbf{X}_2[v_2]\|^r + \tag{16}$$

$$\alpha |\mathbf{A}_1[u_1, v_1] - \mathbf{A}_2[u_2, v_2]|^r] \Big)^{1/r} \Big]$$

$$\leq C \cdot \mathop{\mathbb{E}}_{(u_1, u_2) \sim \pi} \Big[ \Big( \mathop{\mathbb{E}}_{(v_1, v_2) \sim \pi} [(1 - \alpha) \|\mathbf{X}_1[v_1] - \mathbf{X}_2[v_2]\|^r + \alpha |\mathbf{A}_1[u_1, v_1] - \mathbf{A}_2[u_2, v_2]|^r)] \Big)^{1/r} \Big] \tag{17}$$

$$\leq C \cdot \mathop{\mathbb{E}}_{(u_1, u_2) \sim \pi} \Big[ \mathop{\mathbb{E}}_{(v_1, v_2) \sim \pi} [(1 - \alpha) \|\mathbf{X}_1[v_1] - \mathbf{X}_2[v_2]\|^r + \alpha |\mathbf{A}_1[u_1, v_1] - \mathbf{A}_2[u_2, v_2]|^r)] \Big]^{1/r}, \tag{18}$$

where $C_1 = C_W \|\mathbf{W}\|$. We explain the inequalities as follows. The first inequality is from our smoothness assumption stated in the previous subsection. The second is by removing the infimum. The third is another use of Jensen's inequality.

Since the above inequality holds for any valid coupling $\pi$, we can take infimum on both side. Thus, it follows that $d_W(f(\mathcal{G}_1), f(\mathcal{G}_2))$ is at most

$$\inf_{\pi \in \Pi(\mu_1, \mu_2)} \mathop{\mathbb{E}}_{(u_1, u_2) \sim \pi} [\|f(\mathcal{G}_1)_{u_1} - f(\mathcal{G}_2)_{u_2}\|] \tag{19}$$

$$\leq \inf_{\pi \in \Pi(\mu_1, \mu_2)} C_1 \cdot \mathop{\mathbb{E}}_{(u_1, u_2) \sim \pi} \Big[ \mathop{\mathbb{E}}_{(v_1, v_2) \sim \pi} [(1 - \alpha) \|\mathbf{X}_1[v_1] - \mathbf{X}_2[v_2]\|^r + \tag{20}$$

$$\alpha |\mathbf{A}_1[u_1, v_1] - \mathbf{A}_2[u_2, v_2]|^r)] \Big]^{1/r}$$

$$= C_1 \cdot \inf_{\pi \in \Pi(\mu_1, \mu_2)} \mathop{\mathbb{E}}_{(u_1, u_2) \sim \pi} \Big[ \mathop{\mathbb{E}}_{(v_1, v_2) \sim \pi} [(1 - \alpha) \|\mathbf{X}_1[v_1] - \mathbf{X}_2[v_2]\|^r + \tag{21}$$

$$\alpha |\mathbf{A}_1[u_1, v_1] - \mathbf{A}_2[u_2, v_2]|^r)] \Big]^{1/r} \tag{22}$$

$$= C_1 \cdot \Big( \inf_{\pi \in \Pi(\mu_1, \mu_2)} \mathop{\mathbb{E}}_{(u_1, u_2) \sim \pi} \Big[ \mathop{\mathbb{E}}_{(v_1, v_2) \sim \pi} [(1 - \alpha) \|\mathbf{X}_1[v_1] - \mathbf{X}_2[v_2]\|^r + \tag{23}$$

$$\alpha |\mathbf{A}_1[u_1, v_1] - \mathbf{A}_2[u_2, v_2]|^r)] \Big] \Big)^{1/r} \tag{24}$$

$$= C_1 \cdot \mathrm{FGW}_\alpha(\mathcal{G}_1, \mathcal{G}_2), \tag{25}$$

which completes the proof the case of $k = 1$. Note that the inequality is from our smoothness assumption stated in the previous section and the last equality is due to the definition of FGW distance with trade-off parameter $\beta = \alpha$. □

### F.1.3   Proof for general $k > 1$

*Proof.* For general $k > 1$, we can iteratively apply similar logic as in the case of $k = 1$ to bound the output distance with multi-layer GNNs. Specifically, we can write a $k$-layer GNN $f$ as a composite function that concatenates multiple convolution layer (i.e. $f_1, \cdots, f_k$)[8] with ReLU activation functions (i.e. $\sigma_1, \cdots, \sigma_{k-1}$): $f = f_k \circ \sigma_{k-1} \circ f_{k-1} \circ \cdots \circ \sigma_1 \circ f_1$, where $\sigma_j = \mathrm{ReLU}(\cdot)$. For any $m \leq k$, define $h_m := f_m \circ \sigma_{m-1} \circ h_{m-1} = f_m \circ h'_{m-1}$, where $h'_{m-1} = \sigma_{m-1} \circ h_{m-1}$.

---

[8]We assume all these convolution functions $\{f_m\}_{1 \leq m \leq k}$ satisfy the assumption we made in Section F.1.1 with constant $C_W$.

Note that we have $f = h_k$. Then, for any coupling $\pi \in \Pi(\mu_1, \mu_2)$, we have:

$$\mathop{\mathbb{E}}_{(u_1,u_2)\sim\pi}\big[\|f(\mathcal{G}_1)_{u_1} - f(\mathcal{G}_2)_{u_2}\|\big] \tag{26}$$

$$= \mathop{\mathbb{E}}_{(u_1,u_2)\sim\pi}\big[\|h_k(\mathcal{G}_1)_{u_1} - h_k(\mathcal{G}_2)_{u_2}\|\big] \tag{27}$$

$$= \mathop{\mathbb{E}}_{(u_1,u_2)\sim\pi}\big[\|(f_k \circ h'_{k-1})(\mathcal{G}_1)_{u_1} - (f_k \circ h'_{k-1})(\mathcal{G}_2)_{u_2}\|\big] \tag{28}$$

$$\leq \mathop{\mathbb{E}}_{(u_1,u_2)\sim\pi}\big[\|C_1 \cdot d_{W;\alpha}\big(h'_{k-1}(\mathcal{G}_1)_{u_1}, h'_{k-1}(\mathcal{G}_2)_{u_2}\big)\|\big] \tag{29}$$

$$= C_1 \mathop{\mathbb{E}}_{(u_1,u_2)\sim\pi}\bigg[\bigg(\inf_{\pi'\in\Pi(\mu_1,\mu_2)} \mathop{\mathbb{E}}_{(v_1,v_2)\sim\pi'}(1-\alpha)\big[\|h'_{k-1}(\mathcal{G}_1)_{v_1} - h'_{k-1}(\mathcal{G}_2)_{v_2}\|^r\big] + \tag{30}$$

$$\alpha\big[|\mathbf{A}_1[u_1,v_1] - \mathbf{A}_2[u_2,v_2]|^r\big]\bigg)^{1/r}\bigg]$$

$$\leq C_1 \mathop{\mathbb{E}}_{(u_1,u_2)\sim\pi}\bigg[\mathop{\mathbb{E}}_{(v_1,v_2)\sim\pi}\bigg((1-\alpha)\big[\|h'_{k-1}(\mathcal{G}_1)_{v_1} - h'_{k-1}(\mathcal{G}_2)_{v_2}\|^r\big] + \tag{31}$$

$$\alpha\big[|\mathbf{A}_1[u_1,v_1] - \mathbf{A}_2[u_2,v_2]|^r\big]\bigg)^{1/r}\bigg] \tag{32}$$

$$\leq C_1\bigg((1-\alpha)\mathop{\mathbb{E}}_{(v_1,v_2)\sim\pi}\big[\|h'_{k-1}(\mathcal{G}_1)_{v_1} - h'_{k-1}(\mathcal{G}_2)_{v_2}\|^r\big] + \tag{33}$$

$$\alpha\mathop{\mathbb{E}}_{\substack{(u_1,u_2)\sim\pi\\(v_1,v_2)\sim\pi}}\big[|\mathbf{A}_1[u_1,v_1] - \mathbf{A}_2[u_2,v_2]|^r\big]\bigg)^{1/r} \tag{34}$$

$$\leq C_1\bigg((1-\alpha)\mathop{\mathbb{E}}_{(v_1,v_2)\sim\pi}\big[\|h_{k-1}(\mathcal{G}_1)_{v_1} - h_{k-1}(\mathcal{G}_2)_{v_2}\|^r\big] + \tag{35}$$

$$\alpha\mathop{\mathbb{E}}_{\substack{(u_1,u_2)\sim\pi\\(v_1,v_2)\sim\pi}}\big[|\mathbf{A}_1[u_1,v_1] - \mathbf{A}_2[u_2,v_2]|^r\big]\bigg)^{1/r}. \tag{36}$$

Note that the first inequality is from the smoothness assumption, the second is by removing infimum, the third is by Jensen's inequality and the fourth is because $\mathrm{ReLU}(\cdot)$ is a contraction function.

Here, we can iteratively apply the regularity assumption specified in Section F.1.1 to expand the term above: $\|h_{m-1}(\mathcal{G}_1)_{v_1} - h_{m-1}(\mathcal{G}_2)_{v_2}\|^r, \forall m \in \{k, \cdots, 1\}$ to have the following deduction.

$$\mathop{\mathbb{E}}_{(u_1,u_2)\sim\pi}\big[\|f(\mathcal{G}_1)_{u_1} - f(\mathcal{G}_2)_{u_2}\|\big] \tag{37}$$

$$\leq C_1\bigg(C_1^{k-1}(1-\alpha)^k \mathop{\mathbb{E}}_{(v_1,v_2)\sim\pi}\big[\|\mathbf{X}_1[v_1] - \mathbf{X}_2[v_2]\|^r\big] + \tag{38}$$

$$\alpha\sum_{m=0}^{k-1}[C_1(1-\alpha)]^m \mathop{\mathbb{E}}_{\substack{(u_1,u_2)\sim\pi\\(v_1,v_2)\sim\pi}}\big[|\mathbf{A}_1[u_1,v_1] - \mathbf{A}_2[u_2,v_2]|^r\big]\bigg)^{1/r}$$

$$= C \cdot \bigg((1-\beta)\mathop{\mathbb{E}}_{(v_1,v_2)\sim\pi}\big[\|\mathbf{X}_1[v_1] - \mathbf{X}_2[v_2]\|^r\big] + \beta\mathop{\mathbb{E}}_{\substack{(u_1,u_2)\sim\pi\\(v_1,v_2)\sim\pi}}\big[|\mathbf{A}_1[u_1,v_1] - \mathbf{A}_2[u_2,v_2]|^r\big]\bigg)^{1/r}, \tag{39}$$

where $C = C_1\frac{\alpha+(1-\alpha)^k(C_1)^{k-1}(1-C_1)}{1-C_1(1-\alpha)}$ and $\beta = \frac{\alpha(1-C_1^k(1-\alpha)^k)}{\alpha+(1-\alpha)^k(C_1)^{k-1}(1-C_1)}$.

Since the above equation holds for any coupling $\pi$, we can take infimum from both sides to get:

$$d_W(f(\mathcal{G}_1, f(\mathcal{G}_2)) \tag{40}$$

$$= \inf_{\pi \in \Pi(\mu_1, \mu_2)} \mathbb{E}_{(u_1, u_2) \sim \pi}[\|f(\mathcal{G}_1)_{u_1} - f(\mathcal{G}_2)_{u_2}\|] \tag{41}$$

$$\leq C \cdot \inf_{\pi \in \Pi(\mu_1, \mu_2)} \left( (1 - \beta) \mathbb{E}_{(v_1, v_2) \sim \pi}\left[\|\mathbf{X}_1[v_1] - \mathbf{X}_2[v_2]\|^r\right] + \tag{42}\right.$$

$$\left. \beta \mathbb{E}_{\substack{(u_1, u_2) \sim \pi \\ (v_1, v_2) \sim \pi}}\left[|\mathbf{A}_1[u_1, v_1] - \mathbf{A}_2[u_2, v_2]|^r\right] \right)^{1/r} \tag{43}$$

$$= C \cdot \text{FGW}_\beta(\mathcal{G}_1, \mathcal{G}_2), \tag{44}$$

which completes the proof. $\qquad\square$

*Remark* F.1. To justify the smoothness assumption on $g$, we note that it is an abstraction of GNN aggregation functions. For example, aggregation operations such as *mean, max and sum* all satisfy our assumption.

*Remark* F.2. Note that our technical assumption and the results of Theorem 3.1 are independent. Firstly, the assumption on the convolution function $g$ is about the smoothness property *between node representations within a single graph*; while the results of Theorem 3.1 is bounding the FGW distance between *sets of node representations between two graphs*.

### F.2 Proof of Theorem 3.3

For any coupling $\pi \in \Pi(\mathbf{p}^{\text{train}}(\mathbf{w}), \mathbf{q}^{\text{val}})$, by Jensen's inequality and the Lipschitzness assumption,

$$\left| \mathbb{E}_{(\mathcal{G}, y) \sim \mathbf{p}^{\text{train}}(\mathbf{w})}[\mathcal{L}(f(\mathcal{G}), y)] - \mathbb{E}_{(\mathcal{G}, y) \sim \mathbf{q}^{\text{val}}}[\mathcal{L}(f(\mathcal{G}), y)] \right| \tag{45}$$

$$= \left| \mathbb{E}_{(\mathcal{G}^{\text{train}}, y^{\text{train}}) \sim \mathbf{p}^{\text{train}}(\mathbf{w})}[\mathcal{L}(f(\mathcal{G}^{\text{train}}), y^{\text{train}})] - \mathbb{E}_{(\mathcal{G}^{\text{val}}, y^{\text{val}}) \sim \mathbf{q}^{\text{val}}}[\mathcal{L}(f(\mathcal{G}^{\text{val}}), y^{\text{val}})] \right| \tag{46}$$

$$= \left| \mathbb{E}_{((\mathcal{G}^{\text{train}}, y^{\text{train}}), (\mathcal{G}^{\text{val}}, y^{\text{val}})) \sim \pi}[\mathcal{L}(f(\mathcal{G}^{\text{train}}), y^{\text{train}})] - \mathbb{E}_{((\mathcal{G}^{\text{train}}, y^{\text{train}}), (\mathcal{G}^{\text{val}}, y^{\text{val}})) \sim \pi}[\mathcal{L}(f(\mathcal{G}^{\text{val}}), y^{\text{val}})] \right| \tag{47}$$

$$= \left| \mathbb{E}_{((\mathcal{G}^{\text{train}}, y^{\text{train}}), (\mathcal{G}^{\text{val}}, y^{\text{val}})) \sim \pi}[\mathcal{L}(f(\mathcal{G}^{\text{train}}), y^{\text{train}}) - \mathcal{L}(f(\mathcal{G}^{\text{val}}), y^{\text{val}})] \right| \tag{48}$$

$$\leq \mathbb{E}_{((\mathcal{G}^{\text{train}}, y^{\text{train}}), (\mathcal{G}^{\text{val}}, y^{\text{val}})) \sim \pi}[|\mathcal{L}(f(\mathcal{G}^{\text{train}}), y^{\text{train}}) - \mathcal{L}(f(\mathcal{G}^{\text{val}}), y^{\text{val}})|] \tag{49}$$

$$\leq \mathbb{E}_{((\mathcal{G}^{\text{train}}, y^{\text{train}}), (\mathcal{G}^{\text{val}}, y^{\text{val}})) \sim \pi}[C \cdot d_{g\mathcal{Z}}^c((\mathcal{G}^{\text{train}}, y^{\text{train}}), (\mathcal{G}^{\text{val}}, y^{\text{val}}))] \tag{50}$$

$$= C \cdot \mathbb{E}_{((\mathcal{G}^{\text{train}}, y^{\text{train}}), (\mathcal{G}^{\text{val}}, y^{\text{val}})) \sim \pi}[d_{g\mathcal{Z}}^c((\mathcal{G}^{\text{train}}, y^{\text{train}}), (\mathcal{G}^{\text{val}}, y^{\text{val}}))]. \tag{51}$$

Since this holds for any coupling $\pi \in \Pi(\mathbf{p}^{\text{train}}(\mathbf{w}), \mathbf{q}^{\text{val}})$, then we have

$$\left| \mathbb{E}_{(\mathcal{G}, y) \sim \mathbf{p}^{\text{train}}(\mathbf{w})}[\mathcal{L}(f(\mathcal{G}), y)] - \mathbb{E}_{(\mathcal{G}, y) \sim \mathbf{q}^{\text{val}}}[\mathcal{L}(f(\mathcal{G}), y)] \right| \tag{52}$$

$$\leq C \cdot \inf_{\pi \in \Pi(\mathbf{p}^{\text{train}}(\mathbf{w}), \mathbf{q}^{\text{val}})} \mathbb{E}_{((\mathcal{G}^{\text{train}}, y^{\text{train}}), (\mathcal{G}^{\text{val}}, y^{\text{val}})) \sim \pi}[d_{g\mathcal{Z}}^c((\mathcal{G}^{\text{train}}, y^{\text{train}}), (\mathcal{G}^{\text{val}}, y^{\text{val}}))] \tag{53}$$

$$= C \cdot \text{OT}(\mathbf{p}^{\text{train}}(\mathbf{w}), \mathbf{q}^{\text{val}}, d_{g\mathcal{Z}}^c) \tag{54}$$

$$= C \cdot \text{GDD}(\mathcal{D}_{\mathbf{w}}^{\text{train}}, \mathcal{D}^{\text{val}}). \tag{55}$$

It follows that

$$\mathbb{E}_{(\mathcal{G}, y) \sim \mathbf{q}^{\text{val}}}[\mathcal{L}(f(\mathcal{G}), y)] \leq \mathbb{E}_{(\mathcal{G}, y) \sim \mathbf{p}^{\text{train}}(\mathbf{w})}[\mathcal{L}(f(\mathcal{G}), y)] + C \cdot \text{GDD}(\mathcal{D}_{\mathbf{w}}^{\text{train}}, \mathcal{D}^{\text{val}}), \tag{56}$$

which completes the proof. Note that the first inequality follows from Jensen's inequality (w.r.t. the absolute function). The second and third inequalities are both due to the smoothness assumption stated in Theorem 3.3.

### F.3 Proof of Theorem F.3

**Theorem F.3** (Gradient of GDD w.r.t. Training Weights; 26). *Given a distance matrix* $\mathbf{D}$, *a validation empirical measure* $\mathbf{q}^{val}$ *and a training empirical measure* $\mathbf{p}^{train}(\mathbf{w})$ *based on the weight* $\mathbf{w}$. *Let* $\boldsymbol{\beta}(\pi^*)$ *be the dual variables with respect to* $\mathbf{p}^{train}(\mathbf{w})$ *for the* GDD *problem defined in Equation (5). The gradient of* $\mathrm{GDD}(\mathbf{p}^{train}(\mathbf{w}), \mathbf{q}^{val}, \mathbf{D})$ *with respect to* $\mathbf{w}$ *can be computed as:*

$$\nabla_{\mathbf{w}}\mathrm{GDD}(\mathbf{p}^{\mathrm{train}}(\boldsymbol{w}), \mathbf{q}^{\mathrm{val}}, \mathbf{D}) = \boldsymbol{\beta}^*(\pi^*),$$

*where* $\boldsymbol{\beta}^*(\pi^*)$ *is the optimal solution w.r.t.* $\mathbf{p}^{train}(\mathbf{w})$ *to the dual of the GDD problem.*

*Proof.* Omitted. Please see the Sensitivity Theorem stated by Bertsekas [4]. □

## G  Additional Experiments

### G.1  Comparing data selection methods for *graph size* shift on GCN & GIN

We conduct the same evaluation as Table 1 on *graph size* shift with GCN and GIN as backbone model in Table 4.

| Dataset | GNN Architecture → | GCN | | | | GIN | | | |
|---|---|---|---|---|---|---|---|---|---|
| | Selection Method ↓ | $\tau = 10\%$ | $\tau = 20\%$ | $\tau = 50\%$ | Full | $\tau = 10\%$ | $\tau = 20\%$ | $\tau = 50\%$ | Full |
| IMDB-BINARY | Random | $0.573_{\pm 0.041}$ | $0.612_{\pm 0.008}$ | $0.645_{\pm 0.051}$ | $0.630_{\pm 0.008}$ | $0.620_{\pm 0.007}$ | $0.582_{\pm 0.009}$ | $0.605_{\pm 0.019}$ | $0.602_{\pm 0.010}$ |
| | KiDD-LR | $0.592_{\pm 0.015}$ | $0.540_{\pm 0.014}$ | $0.652_{\pm 0.008}$ | | $0.553_{\pm 0.013}$ | $0.555_{\pm 0.012}$ | $0.577_{\pm 0.012}$ | |
| | LAVA | $\underline{0.824}_{\pm 0.008}$ | $\underline{0.823}_{\pm 0.019}$ | $\mathbf{0.837}_{\pm 0.006}$ | | $\underline{0.822}_{\pm 0.005}$ | $\mathbf{0.830}_{\pm 0.011}$ | $\mathbf{0.848}_{\pm 0.002}$ | |
| | GRADATE | $\mathbf{0.826}_{\pm 0.009}$ | $\mathbf{0.825}_{\pm 0.018}$ | $\underline{0.830}_{\pm 0.007}$ | | $\mathbf{0.823}_{\pm 0.002}$ | $\underline{0.820}_{\pm 0.008}$ | $\underline{0.832}_{\pm 0.008}$ | |
| IMDB-MULTI | Random | $\underline{0.374}_{\pm 0.031}$ | $0.354_{\pm 0.008}$ | $0.366_{\pm 0.008}$ | $0.386_{\pm 0.006}$ | $0.351_{\pm 0.008}$ | $0.372_{\pm 0.039}$ | $0.369_{\pm 0.019}$ | $0.368_{\pm 0.010}$ |
| | KiDD-LR | $0.329_{\pm 0.010}$ | $0.416_{\pm 0.064}$ | $0.432_{\pm 0.010}$ | | $\underline{0.346}_{\pm 0.048}$ | $0.371_{\pm 0.010}$ | $0.412_{\pm 0.018}$ | |
| | LAVA | $0.314_{\pm 0.006}$ | $\underline{0.426}_{\pm 0.003}$ | $\underline{0.600}_{\pm 0.005}$ | | $0.341_{\pm 0.049}$ | $\underline{0.388}_{\pm 0.018}$ | $\underline{0.563}_{\pm 0.007}$ | |
| | GRADATE | $\mathbf{0.353}_{\pm 0.000}$ | $\mathbf{0.524}_{\pm 0.016}$ | $\mathbf{0.602}_{\pm 0.004}$ | | $\mathbf{0.349}_{\pm 0.046}$ | $\mathbf{0.497}_{\pm 0.015}$ | $\mathbf{0.604}_{\pm 0.006}$ | |
| MSRC_21 | Random | $0.450_{\pm 0.008}$ | $0.497_{\pm 0.011}$ | $0.781_{\pm 0.019}$ | $0.816_{\pm 0.026}$ | $0.149_{\pm 0.007}$ | $0.418_{\pm 0.008}$ | $0.690_{\pm 0.015}$ | $0.749_{\pm 0.023}$ |
| | KiDD-LR | $\mathbf{0.725}_{\pm 0.017}$ | $0.819_{\pm 0.015}$ | $0.857_{\pm 0.008}$ | | $\mathbf{0.649}_{\pm 0.012}$ | $0.743_{\pm 0.008}$ | $0.781_{\pm 0.050}$ | |
| | LAVA | $0.617_{\pm 0.015}$ | $\underline{0.825}_{\pm 0.014}$ | $\underline{0.918}_{\pm 0.018}$ | | $0.617_{\pm 0.008}$ | $\underline{0.810}_{\pm 0.004}$ | $\underline{0.889}_{\pm 0.011}$ | |
| | GRADATE | $\underline{0.670}_{\pm 0.017}$ | $\mathbf{0.836}_{\pm 0.017}$ | $\mathbf{0.953}_{\pm 0.011}$ | | $\underline{0.629}_{\pm 0.011}$ | $\mathbf{0.813}_{\pm 0.008}$ | $\mathbf{0.901}_{\pm 0.008}$ | |
| ogbg-molbace | Random | $0.443_{\pm 0.014}$ | $0.504_{\pm 0.022}$ | $0.476_{\pm 0.011}$ | $0.434_{\pm 0.033}$ | $0.479_{\pm 0.070}$ | $0.471_{\pm 0.092}$ | $0.578_{\pm 0.030}$ | $0.548_{\pm 0.028}$ |
| | KiDD-LR | $0.446_{\pm 0.040}$ | $0.489_{\pm 0.049}$ | $0.483_{\pm 0.011}$ | | $0.547_{\pm 0.080}$ | $0.523_{\pm 0.060}$ | $0.571_{\pm 0.013}$ | |
| | LAVA | $\underline{0.563}_{\pm 0.045}$ | $\underline{0.574}_{\pm 0.067}$ | $\underline{0.535}_{\pm 0.044}$ | | $\underline{0.645}_{\pm 0.035}$ | $\mathbf{0.641}_{\pm 0.027}$ | $\mathbf{0.648}_{\pm 0.025}$ | |
| | GRADATE | $\mathbf{0.570}_{\pm 0.080}$ | $\mathbf{0.599}_{\pm 0.037}$ | $\mathbf{0.575}_{\pm 0.056}$ | | $\mathbf{0.646}_{\pm 0.033}$ | $\underline{0.618}_{\pm 0.061}$ | $\underline{0.630}_{\pm 0.020}$ | |
| ogbg-molbbbp | Random | $0.499_{\pm 0.041}$ | $0.635_{\pm 0.042}$ | $0.648_{\pm 0.031}$ | $0.618_{\pm 0.037}$ | $0.698_{\pm 0.010}$ | $0.633_{\pm 0.043}$ | $0.691_{\pm 0.040}$ | $0.779_{\pm 0.017}$ |
| | KiDD-LR | $0.639_{\pm 0.025}$ | $0.599_{\pm 0.013}$ | $0.611_{\pm 0.023}$ | | $0.546_{\pm 0.105}$ | $0.656_{\pm 0.038}$ | $0.609_{\pm 0.081}$ | |
| | LAVA | $\underline{0.667}_{\pm 0.015}$ | $\mathbf{0.675}_{\pm 0.013}$ | $\mathbf{0.691}_{\pm 0.017}$ | | $\underline{0.859}_{\pm 0.019}$ | $\underline{0.889}_{\pm 0.016}$ | $\underline{0.893}_{\pm 0.011}$ | |
| | GRADATE | $\mathbf{0.677}_{\pm 0.007}$ | $\underline{0.671}_{\pm 0.015}$ | $\underline{0.673}_{\pm 0.041}$ | | $\mathbf{0.866}_{\pm 0.016}$ | $\mathbf{0.890}_{\pm 0.011}$ | $\mathbf{0.895}_{\pm 0.012}$ | |
| ogbg-molhiv | Random | $0.576_{\pm 0.008}$ | $0.579_{\pm 0.004}$ | $0.594_{\pm 0.001}$ | $0.592_{\pm 0.000}$ | $0.613_{\pm 0.004}$ | $0.617_{\pm 0.045}$ | $0.624_{\pm 0.015}$ | $0.664_{\pm 0.027}$ |
| | KiDD-LR | $0.556_{\pm 0.001}$ | $0.551_{\pm 0.027}$ | $0.595_{\pm 0.003}$ | | $0.586_{\pm 0.055}$ | $0.586_{\pm 0.014}$ | $0.629_{\pm 0.019}$ | |
| | LAVA | $\mathbf{0.669}_{\pm 0.001}$ | $\mathbf{0.683}_{\pm 0.004}$ | $\mathbf{0.659}_{\pm 0.002}$ | | $\mathbf{0.769}_{\pm 0.014}$ | $0.737_{\pm 0.012}$ | $0.796_{\pm 0.025}$ | |
| | GRADATE | $\underline{0.640}_{\pm 0.002}$ | $\underline{0.638}_{\pm 0.006}$ | $\underline{0.629}_{\pm 0.000}$ | | $\underline{0.731}_{\pm 0.017}$ | $\mathbf{0.767}_{\pm 0.004}$ | $\mathbf{0.805}_{\pm 0.024}$ | |

Table 4: Performance comparison across data selection methods for *graph size* shift on GCN and GIN. We use **bold**/underline to indicate the 1st/2nd best results. GRADATE achieves top-2 performance across all datasets and is the best-performer in most settings.

### G.2  Comparing data selection methods for *graph density* shift on GAT & GraphSAGE

We conduct the same evaluation as Table 1 on *graph density* shift with GAT and GraphSAGE as backbone model in Table 5.

### G.3  Comparing data selection methods for *graph size* shift on GAT & GraphSAGE

We conduct the same evaluation as Table 1 on *graph size* shift with GAT and GraphSAGE as backbone model in Table 6.

### G.4  Comparing GDA and vanilla methods for *graph size* shift

We conduct the same evaluation as Table 2 on *graph size* shift in Table 7.

| Dataset | Selection Method ↓ | GAT | | | | GraphSAGE | | | |
|---|---|---|---|---|---|---|---|---|---|
| | GNN Architecture → | $\tau=10\%$ | $\tau=20\%$ | $\tau=50\%$ | Full | $\tau=10\%$ | $\tau=20\%$ | $\tau=50\%$ | Full |
| IMDB-BINARY | Random | $0.602_{\pm0.005}$ | $0.695_{\pm0.035}$ | $0.797_{\pm0.005}$ | $0.807_{\pm0.033}$ | $0.730_{\pm0.014}$ | $0.785_{\pm0.025}$ | $0.762_{\pm0.027}$ | $0.823_{\pm0.009}$ |
| | K1DD-LR | $0.683_{\pm0.041}$ | $0.803_{\pm0.005}$ | $0.817_{\pm0.024}$ | | $0.662_{\pm0.054}$ | $0.785_{\pm0.025}$ | $0.775_{\pm0.054}$ | |
| | LAVA | $\underline{0.818}_{\pm0.010}$ | $\underline{0.857}_{\pm0.009}$ | $\underline{0.885}_{\pm0.018}$ | | $\underline{0.827}_{\pm0.005}$ | $\underline{0.840}_{\pm0.021}$ | $\underline{0.883}_{\pm0.012}$ | |
| | GRADATE | $\mathbf{0.850}_{\pm0.023}$ | $\mathbf{0.865}_{\pm0.008}$ | $\mathbf{0.892}_{\pm0.012}$ | | $\mathbf{0.835}_{\pm0.015}$ | $\mathbf{0.852}_{\pm0.035}$ | $\mathbf{0.907}_{\pm0.005}$ | |
| IMDB-MULTI | Random | $0.087_{\pm0.014}$ | $0.071_{\pm0.006}$ | $0.076_{\pm0.003}$ | $0.080_{\pm0.000}$ | $0.090_{\pm0.005}$ | $0.203_{\pm0.061}$ | $0.126_{\pm0.064}$ | $0.097_{\pm0.024}$ |
| | K1DD-LR | $0.176_{\pm0.024}$ | $0.121_{\pm0.044}$ | $0.158_{\pm0.036}$ | | $0.154_{\pm0.028}$ | $0.124_{\pm0.068}$ | $0.054_{\pm0.011}$ | |
| | LAVA | $\underline{0.597}_{\pm0.273}$ | $\mathbf{0.599}_{\pm0.294}$ | $\underline{0.341}_{\pm0.049}$ | | $\mathbf{0.341}_{\pm0.049}$ | $\mathbf{0.307}_{\pm0.164}$ | $\underline{0.328}_{\pm0.317}$ | |
| | GRADATE | $\mathbf{0.790}_{\pm0.000}$ | $\underline{0.589}_{\pm0.287}$ | $\mathbf{0.776}_{\pm0.039}$ | | $\underline{0.306}_{\pm0.216}$ | $\underline{0.299}_{\pm0.282}$ | $\mathbf{0.363}_{\pm0.238}$ | |
| MSRC_21 | Random | $0.462_{\pm0.029}$ | $0.763_{\pm0.007}$ | $0.857_{\pm0.018}$ | $0.860_{\pm0.007}$ | $0.617_{\pm0.017}$ | $0.725_{\pm0.033}$ | $0.842_{\pm0.029}$ | $0.874_{\pm0.004}$ |
| | K1DD-LR | $0.661_{\pm0.030}$ | $0.778_{\pm0.015}$ | $0.860_{\pm0.025}$ | | $0.681_{\pm0.073}$ | $0.787_{\pm0.025}$ | $0.857_{\pm0.004}$ | |
| | LAVA | $\underline{0.699}_{\pm0.047}$ | $\underline{0.816}_{\pm0.037}$ | $\underline{0.912}_{\pm0.007}$ | | $\underline{0.766}_{\pm0.029}$ | $\underline{0.857}_{\pm0.015}$ | $\underline{0.918}_{\pm0.011}$ | |
| | GRADATE | $\mathbf{0.716}_{\pm0.017}$ | $\mathbf{0.822}_{\pm0.004}$ | $\mathbf{0.921}_{\pm0.007}$ | | $\mathbf{0.781}_{\pm0.026}$ | $\mathbf{0.877}_{\pm0.026}$ | $\mathbf{0.944}_{\pm0.011}$ | |
| ogbg-molbace | Random | $0.480_{\pm0.040}$ | $\mathbf{0.606}_{\pm0.085}$ | $0.637_{\pm0.075}$ | $0.583_{\pm0.042}$ | $0.459_{\pm0.149}$ | $0.478_{\pm0.097}$ | $0.503_{\pm0.034}$ | $0.622_{\pm0.119}$ |
| | K1DD-LR | $\underline{0.558}_{\pm0.012}$ | $0.443_{\pm0.029}$ | $0.628_{\pm0.023}$ | | $0.606_{\pm0.023}$ | $\underline{0.596}_{\pm0.079}$ | $\underline{0.607}_{\pm0.047}$ | |
| | LAVA | $\mathbf{0.564}_{\pm0.097}$ | $0.519_{\pm0.007}$ | $\underline{0.696}_{\pm0.031}$ | | $\underline{0.620}_{\pm0.075}$ | $\mathbf{0.649}_{\pm0.004}$ | $\mathbf{0.651}_{\pm0.059}$ | |
| | GRADATE | $0.501_{\pm0.017}$ | $\underline{0.541}_{\pm0.048}$ | $\mathbf{0.720}_{\pm0.004}$ | | $\mathbf{0.621}_{\pm0.067}$ | $0.587_{\pm0.078}$ | $0.568_{\pm0.126}$ | |
| ogbg-molbbbp | Random | $0.511_{\pm0.034}$ | $0.529_{\pm0.027}$ | $0.513_{\pm0.018}$ | $0.569_{\pm0.030}$ | $0.463_{\pm0.012}$ | $0.385_{\pm0.032}$ | $0.468_{\pm0.008}$ | $0.447_{\pm0.008}$ |
| | K1DD-LR | $0.444_{\pm0.050}$ | $0.405_{\pm0.021}$ | $0.434_{\pm0.025}$ | | $0.392_{\pm0.002}$ | $0.415_{\pm0.028}$ | $0.466_{\pm0.034}$ | |
| | LAVA | $\underline{0.584}_{\pm0.054}$ | $\underline{0.552}_{\pm0.018}$ | $\underline{0.603}_{\pm0.021}$ | | $\underline{0.526}_{\pm0.087}$ | $\mathbf{0.612}_{\pm0.005}$ | $\underline{0.495}_{\pm0.029}$ | |
| | GRADATE | $\mathbf{0.617}_{\pm0.038}$ | $\mathbf{0.578}_{\pm0.038}$ | $\mathbf{0.632}_{\pm0.036}$ | | $\mathbf{0.580}_{\pm0.067}$ | $\underline{0.558}_{\pm0.064}$ | $\mathbf{0.528}_{\pm0.027}$ | |
| ogbg-molhiv | Random | $0.601_{\pm0.017}$ | $0.591_{\pm0.011}$ | $0.581_{\pm0.016}$ | $0.571_{\pm0.030}$ | $0.577_{\pm0.016}$ | $0.591_{\pm0.007}$ | $0.594_{\pm0.005}$ | $0.588_{\pm0.003}$ |
| | K1DD-LR | $0.620_{\pm0.001}$ | $0.616_{\pm0.003}$ | $0.615_{\pm0.007}$ | | $\mathbf{0.607}_{\pm0.008}$ | $0.534_{\pm0.057}$ | $0.603_{\pm0.018}$ | |
| | LAVA | $\underline{0.621}_{\pm0.001}$ | $\mathbf{0.631}_{\pm0.003}$ | $\mathbf{0.624}_{\pm0.014}$ | | $0.575_{\pm0.012}$ | $\mathbf{0.607}_{\pm0.008}$ | $0.608_{\pm0.007}$ | |
| | GRADATE | $\mathbf{0.638}_{\pm0.001}$ | $\underline{0.620}_{\pm0.002}$ | $\underline{0.619}_{\pm0.004}$ | | $\underline{0.599}_{\pm0.021}$ | $\underline{0.598}_{\pm0.009}$ | $\mathbf{0.610}_{\pm0.006}$ | |

Table 5: Performance comparison across data selection methods for *graph density* shift on GAT and GraphSAGE. We use **bold**/underline to indicate the 1st/2nd best results. GRADATE is the best-performer in most settings.

| Dataset | Selection Method ↓ | GAT | | | | GraphSAGE | | | |
|---|---|---|---|---|---|---|---|---|---|
| | GNN Architecture → | $\tau=10\%$ | $\tau=20\%$ | $\tau=50\%$ | Full | $\tau=10\%$ | $\tau=20\%$ | $\tau=50\%$ | Full |
| IMDB-BINARY | Random | $0.678_{\pm0.082}$ | $0.558_{\pm0.022}$ | $0.660_{\pm0.085}$ | $0.595_{\pm0.007}$ | $0.555_{\pm0.011}$ | $0.563_{\pm0.012}$ | $0.562_{\pm0.012}$ | $0.567_{\pm0.018}$ |
| | K1DD-LR | $0.683_{\pm0.071}$ | $0.587_{\pm0.081}$ | $0.665_{\pm0.098}$ | | $0.663_{\pm0.035}$ | $0.558_{\pm0.013}$ | $0.595_{\pm0.007}$ | |
| | LAVA | $\underline{0.808}_{\pm0.014}$ | $\underline{0.830}_{\pm0.004}$ | $\underline{0.835}_{\pm0.000}$ | | $\underline{0.807}_{\pm0.026}$ | $\underline{0.808}_{\pm0.027}$ | $\underline{0.830}_{\pm0.004}$ | |
| | GRADATE | $\mathbf{0.835}_{\pm0.018}$ | $\mathbf{0.833}_{\pm0.005}$ | $\mathbf{0.837}_{\pm0.010}$ | | $\mathbf{0.808}_{\pm0.016}$ | $\mathbf{0.828}_{\pm0.024}$ | $\mathbf{0.838}_{\pm0.016}$ | |
| IMDB-MULTI | Random | $\mathbf{0.384}_{\pm0.014}$ | $0.408_{\pm0.004}$ | $0.384_{\pm0.034}$ | $0.374_{\pm0.028}$ | $0.336_{\pm0.010}$ | $0.357_{\pm0.005}$ | $0.381_{\pm0.026}$ | $0.391_{\pm0.026}$ |
| | K1DD-LR | $\underline{0.366}_{\pm0.020}$ | $\underline{0.434}_{\pm0.006}$ | $0.404_{\pm0.010}$ | | $0.339_{\pm0.030}$ | $\mathbf{0.418}_{\pm0.030}$ | $\underline{0.422}_{\pm0.011}$ | |
| | LAVA | $0.333_{\pm0.058}$ | $0.417_{\pm0.015}$ | $\underline{0.577}_{\pm0.014}$ | | $\underline{0.374}_{\pm0.021}$ | $0.389_{\pm0.039}$ | $0.392_{\pm0.036}$ | |
| | GRADATE | $0.342_{\pm0.050}$ | $\mathbf{0.537}_{\pm0.003}$ | $\mathbf{0.616}_{\pm0.002}$ | | $\mathbf{0.392}_{\pm0.032}$ | $0.360_{\pm0.025}$ | $\mathbf{0.517}_{\pm0.071}$ | |
| MSRC_21 | Random | $0.284_{\pm0.025}$ | $0.614_{\pm0.056}$ | $0.731_{\pm0.018}$ | $0.787_{\pm0.004}$ | $0.497_{\pm0.023}$ | $0.412_{\pm0.050}$ | $0.725_{\pm0.022}$ | $0.810_{\pm0.027}$ |
| | K1DD-LR | $\underline{0.626}_{\pm0.004}$ | $0.746_{\pm0.026}$ | $0.830_{\pm0.011}$ | | $\underline{0.722}_{\pm0.030}$ | $0.798_{\pm0.029}$ | $0.789_{\pm0.026}$ | |
| | LAVA | $0.620_{\pm0.015}$ | $\underline{0.798}_{\pm0.012}$ | $\underline{0.909}_{\pm0.018}$ | | $0.693_{\pm0.029}$ | $\underline{0.827}_{\pm0.015}$ | $\underline{0.918}_{\pm0.008}$ | |
| | GRADATE | $\mathbf{0.643}_{\pm0.039}$ | $\mathbf{0.860}_{\pm0.021}$ | $\mathbf{0.947}_{\pm0.014}$ | | $\mathbf{0.760}_{\pm0.023}$ | $\mathbf{0.842}_{\pm0.007}$ | $\mathbf{0.944}_{\pm0.004}$ | |
| ogbg-molbace | Random | $0.515_{\pm0.006}$ | $0.464_{\pm0.056}$ | $0.488_{\pm0.005}$ | $0.463_{\pm0.004}$ | $0.523_{\pm0.048}$ | $0.463_{\pm0.020}$ | $0.583_{\pm0.027}$ | $0.487_{\pm0.108}$ |
| | K1DD-LR | $0.480_{\pm0.011}$ | $0.452_{\pm0.016}$ | $0.467_{\pm0.022}$ | | $0.507_{\pm0.078}$ | $0.457_{\pm0.029}$ | $0.467_{\pm0.043}$ | |
| | LAVA | $\underline{0.524}_{\pm0.037}$ | $\underline{0.545}_{\pm0.058}$ | $\mathbf{0.613}_{\pm0.080}$ | | $\mathbf{0.650}_{\pm0.011}$ | $\underline{0.481}_{\pm0.007}$ | $\underline{0.550}_{\pm0.060}$ | |
| | GRADATE | $\mathbf{0.529}_{\pm0.029}$ | $\mathbf{0.570}_{\pm0.062}$ | $\underline{0.509}_{\pm0.006}$ | | $\underline{0.556}_{\pm0.040}$ | $\mathbf{0.561}_{\pm0.090}$ | $\mathbf{0.564}_{\pm0.095}$ | |
| ogbg-molbbbp | Random | $0.666_{\pm0.003}$ | $0.677_{\pm0.007}$ | $0.684_{\pm0.008}$ | $0.679_{\pm0.004}$ | $\underline{0.634}_{\pm0.018}$ | $0.648_{\pm0.028}$ | $0.641_{\pm0.025}$ | $0.680_{\pm0.010}$ |
| | K1DD-LR | $0.594_{\pm0.017}$ | $0.596_{\pm0.020}$ | $0.650_{\pm0.002}$ | | $0.518_{\pm0.046}$ | $0.594_{\pm0.004}$ | $0.602_{\pm0.061}$ | |
| | LAVA | $\underline{0.714}_{\pm0.011}$ | $\mathbf{0.731}_{\pm0.019}$ | $\underline{0.710}_{\pm0.036}$ | | $\mathbf{0.639}_{\pm0.022}$ | $0.602_{\pm0.030}$ | $\underline{0.645}_{\pm0.026}$ | |
| | GRADATE | $\mathbf{0.735}_{\pm0.021}$ | $\underline{0.699}_{\pm0.027}$ | $\mathbf{0.713}_{\pm0.005}$ | | $0.623_{\pm0.041}$ | $\mathbf{0.650}_{\pm0.019}$ | $\mathbf{0.652}_{\pm0.041}$ | |
| ogbg-molhiv | Random | $0.584_{\pm0.001}$ | $0.585_{\pm0.002}$ | $0.589_{\pm0.003}$ | $0.588_{\pm0.005}$ | $0.610_{\pm0.024}$ | $0.491_{\pm0.031}$ | $0.603_{\pm0.001}$ | $0.596_{\pm0.000}$ |
| | K1DD-LR | $0.586_{\pm0.001}$ | $0.584_{\pm0.001}$ | $0.584_{\pm0.004}$ | | $0.549_{\pm0.071}$ | $0.588_{\pm0.008}$ | $0.564_{\pm0.011}$ | |
| | LAVA | $\mathbf{0.704}_{\pm0.005}$ | $\mathbf{0.773}_{\pm0.007}$ | $\mathbf{0.759}_{\pm0.002}$ | | $\mathbf{0.721}_{\pm0.004}$ | $\mathbf{0.701}_{\pm0.008}$ | $\mathbf{0.686}_{\pm0.011}$ | |
| | GRADATE | $\underline{0.663}_{\pm0.005}$ | $\underline{0.655}_{\pm0.009}$ | $\underline{0.660}_{\pm0.008}$ | | $\underline{0.637}_{\pm0.002}$ | $\underline{0.646}_{\pm0.019}$ | $\underline{0.640}_{\pm0.007}$ | |

Table 6: Performance comparison across data selection methods for *graph size* shift on GAT and GraphSAGE. We use **bold**/underline to indicate the 1st/2nd best results. GRADATE achieves top-2 performance across most settings. The under-performance on `ogbg-molhiv` might due to the reason discussed in Section 4.5.

### G.5 Enhancing GDA methods for *graph size* shift

We conduct the same evaluation as Table 3 on *graph size* shift in Table 8.

### G.6 Validation-label-free setting

While we originally consider $c$ as tunable parameter, we acknowledge that the existence of validation labels will be implicitly required when $c \neq 0$, which might not be practical under some real-world scenarios. Here, we pick two GNN backbones (i.e. GCN [30] and GIN [65]) evaluate the effectiveness

| Type | Model | Data | IMDB-BINARY | IMDB-MULTI | MSRC_21 | ogbg-molbace | ogbg-molbbbp | ogbg-molhiv |
|------|-------|------|-------------|------------|---------|--------------|--------------|-------------|
| | | | | | | Dataset | | |
| GDA | AdaGCN | Full | $0.593_{\pm 0.012}$ | $0.362_{\pm 0.017}$ | $0.202_{\pm 0.075}$ | $0.513_{\pm 0.018}$ | $0.625_{\pm 0.137}$ | $0.412_{\pm 0.011}$ |
| | GRADE | Full | $0.648_{\pm 0.105}$ | $0.390_{\pm 0.019}$ | $0.696_{\pm 0.008}$ | $0.403_{\pm 0.018}$ | $0.669_{\pm 0.005}$ | $0.599_{\pm 0.005}$ |
| | ASN | Full | $0.633_{\pm 0.054}$ | $0.372_{\pm 0.009}$ | $0.734_{\pm 0.015}$ | $0.523_{\pm 0.091}$ | $0.616_{\pm 0.042}$ | $0.519_{\pm 0.077}$ |
| | UDAGCN | Full | $0.688_{\pm 0.049}$ | $0.392_{\pm 0.046}$ | $0.260_{\pm 0.049}$ | $0.448_{\pm 0.020}$ | $0.513_{\pm 0.024}$ | $0.439_{\pm 0.034}$ |
| Vanilla | GCN | Random 20% | $0.612_{\pm 0.008}$ | $0.354_{\pm 0.008}$ | $0.497_{\pm 0.011}$ | $0.504_{\pm 0.022}$ | $0.635_{\pm 0.042}$ | $0.579_{\pm 0.004}$ |
| | | LAVA 20% | $0.823_{\pm 0.019}$ | $0.426_{\pm 0.003}$ | $0.825_{\pm 0.014}$ | $0.574_{\pm 0.067}$ | $0.675_{\pm 0.013}$ | $0.683_{\pm 0.038}$ |
| | | GRADATE 20% | $0.825_{\pm 0.018}$ | $\underline{0.524}_{\pm 0.016}$ | $0.836_{\pm 0.017}$ | $0.599_{\pm 0.037}$ | $0.671_{\pm 0.015}$ | $0.638_{\pm 0.006}$ |
| | GIN | Random 20% | $0.582_{\pm 0.009}$ | $0.372_{\pm 0.039}$ | $0.418_{\pm 0.008}$ | $0.471_{\pm 0.092}$ | $0.633_{\pm 0.043}$ | $0.617_{\pm 0.045}$ |
| | | LAVA 20% | $0.830_{\pm 0.011}$ | $0.388_{\pm 0.018}$ | $0.810_{\pm 0.004}$ | $\mathbf{0.641}_{\pm 0.027}$ | $\underline{0.889}_{\pm 0.016}$ | $0.737_{\pm 0.012}$ |
| | | GRADATE 20% | $0.820_{\pm 0.008}$ | $0.497_{\pm 0.015}$ | $0.813_{\pm 0.008}$ | $\underline{0.618}_{\pm 0.061}$ | $\mathbf{0.890}_{\pm 0.011}$ | $\underline{0.767}_{\pm 0.004}$ |
| | GAT | Random 20% | $0.558_{\pm 0.022}$ | $0.408_{\pm 0.004}$ | $0.614_{\pm 0.056}$ | $0.464_{\pm 0.056}$ | $0.677_{\pm 0.007}$ | $0.585_{\pm 0.002}$ |
| | | LAVA 20% | $\underline{0.830}_{\pm 0.004}$ | $0.417_{\pm 0.015}$ | $0.798_{\pm 0.012}$ | $0.545_{\pm 0.058}$ | $0.731_{\pm 0.019}$ | $\mathbf{0.773}_{\pm 0.007}$ |
| | | GRADATE 20% | $\mathbf{0.833}_{\pm 0.005}$ | $\mathbf{0.537}_{\pm 0.003}$ | $\mathbf{0.860}_{\pm 0.021}$ | $0.570_{\pm 0.062}$ | $0.699_{\pm 0.027}$ | $0.655_{\pm 0.009}$ |
| | GraphSAGE | Random 20% | $0.563_{\pm 0.012}$ | $0.357_{\pm 0.005}$ | $0.412_{\pm 0.050}$ | $0.463_{\pm 0.020}$ | $0.648_{\pm 0.028}$ | $0.491_{\pm 0.031}$ |
| | | LAVA 20% | $0.808_{\pm 0.027}$ | $0.389_{\pm 0.039}$ | $0.827_{\pm 0.015}$ | $0.481_{\pm 0.007}$ | $0.602_{\pm 0.030}$ | $0.701_{\pm 0.008}$ |
| | | GRADATE 20% | $0.828_{\pm 0.024}$ | $0.360_{\pm 0.025}$ | $\underline{0.842}_{\pm 0.007}$ | $0.561_{\pm 0.090}$ | $0.650_{\pm 0.019}$ | $0.646_{\pm 0.019}$ |

Table 7: Performance comparison across GDA and vanilla methods for *graph size* shift. We use **bold**/underline to indicate the 1st/2nd best results. GRADATE can consistently achieve top-2 performance across all datasets and is the best performer in most settings.

of GRADATE with $c = 0$ (i.e. validation-label-free). The result is shown in Table 9. For simplicity, we only report relative performance improvement (%) of GRADATE over the strongest baseline under all settings. We can observe that the advantage of GRADATE remains comparable to what is reported in the main text.

## G.7 Additional backbones

In addition to typical GNN backbones, we also conduct experiments on a wider range of graph algorithms, including SGFormer [60] and APPNP [16]. In Table 10, we report relative performance improvement (%) of GRADATE over the strongest baseline under all settings. We can observe that GRADATE still outperforms other baselines mostly.

## G.8 Additional GDA methods

We add additional experiments based on two variants of A2GNN [34] with different losses and TDSS [7]. In Table 11, we report relative performance improvement (%) of GRADATE over the strongest baseline under all settings. For simplicity, we report results with selection ratio equals to 20%. It can be observed that GRADATE consistently provides the most significant enhancements for the three newly evaluated GDA methods compared to other selection baselines.

## H  Datasets

In Table 12, we provide details of datasets used in this work. For # NODES and # EDGES, we report the mean sizes across all graphs in the dataset.

## I  Backbone GNN Settings for Graph Selection Evaluation

*GNN Models.* We consider four widely used graph neural network architecture, GCN [30], GIN [65], GAT [58] and GraphSAGE [18]. The detailed model architectures are described as follows: (i) For GCN, we use three GCN layers with number of hidden dimensions equal to 32. ReLU is used between layers and a global mean pooling layer is set as the readout layer to generate graph-level embedding. A dropout layer with probability $p = 0.5$ is applied after the GCN layers. Finally, a linear layer with softmax is placed at the end for graph class prediction. (ii) For GIN, we use three-layer GIN with 32 hidden dimensions. We use ReLU between layers and global mean pooling for readout. A dropout layer with probability $0.5$ is placed after GIN layers and finally a linear layer with softmax for prediction. (iii) For GAT, we use two-layer GAT layers with four heads with global mean pooling

Table 8 (AdaGCN / GRADE):

| Dataset | GDA Method → | AdaGCN | | | | GRADE | | | |
|---|---|---|---|---|---|---|---|---|---|
| | Selection Method ↓ | $\tau=10\%$ | $\tau=20\%$ | $\tau=50\%$ | Full | $\tau=10\%$ | $\tau=20\%$ | $\tau=50\%$ | Full |
| IMDB-BINARY | Random | 0.582±0.091 | 0.520±0.103 | 0.455±0.120 | | 0.572±0.111 | 0.522±0.045 | 0.613±0.095 | |
| | LAVA | 0.818±0.012 | 0.815±0.005 | 0.810±0.013 | 0.593±0.012 | 0.813±0.007 | 0.814±0.007 | 0.816±0.005 | 0.648±0.105 |
| | GRADATE | **0.834**±0.014 | **0.830**±0.010 | **0.822**±0.022 | | 0.814±0.013 | **0.826**±0.007 | **0.827**±0.013 | |
| IMDB-MULTI | Random | 0.261±0.064 | 0.247±0.052 | 0.252±0.059 | | 0.312±0.034 | 0.280±0.000 | 0.282±0.030 | |
| | LAVA | 0.374±0.055 | 0.385±0.080 | 0.368±0.098 | 0.362±0.017 | 0.386±0.047 | 0.407±0.076 | 0.411±0.050 | 0.390±0.019 |
| | GRADATE | **0.386**±0.053 | **0.442**±0.085 | **0.509**±0.114 | | **0.401**±0.076 | **0.411**±0.076 | **0.503**±0.112 | |
| MSRC_21 | Random | 0.084±0.028 | 0.079±0.022 | 0.114±0.030 | | 0.137±0.012 | 0.379±0.053 | 0.667±0.055 | |
| | LAVA | 0.377±0.029 | **0.472**±0.030 | 0.540±0.103 | 0.202±0.075 | 0.532±0.029 | 0.728±0.038 | 0.854±0.035 | 0.696±0.008 |
| | GRADATE | **0.411**±0.075 | 0.465±0.042 | **0.593**±0.071 | | **0.553**±0.043 | **0.744**±0.044 | **0.867**±0.013 | |
| ogbg-molbace | Random | 0.498±0.091 | 0.477±0.038 | 0.498±0.063 | | 0.478±0.005 | 0.468±0.074 | 0.475±0.046 | |
| | LAVA | 0.510±0.027 | 0.523±0.066 | 0.529±0.056 | 0.513±0.018 | 0.509±0.055 | **0.559**±0.026 | **0.552**±0.033 | 0.403±0.018 |
| | GRADATE | **0.524**±0.057 | **0.560**±0.053 | **0.550**±0.030 | | **0.556**±0.069 | 0.508±0.025 | 0.512±0.022 | |
| ogbg-molbbbp | Random | 0.539±0.020 | 0.600±0.044 | 0.538±0.083 | | 0.632±0.006 | **0.648**±0.002 | **0.650**±0.002 | |
| | LAVA | 0.583±0.075 | 0.653±0.007 | 0.657±0.004 | 0.625±0.137 | 0.631±0.011 | 0.640±0.001 | 0.637±0.003 | 0.669±0.005 |
| | GRADATE | **0.662**±0.020 | **0.654**±0.004 | **0.664**±0.010 | | **0.636**±0.011 | 0.641±0.005 | 0.639±0.013 | |
| ogbg-molhiv | Random | 0.356±0.023 | 0.358±0.013 | 0.364±0.011 | | 0.588±0.014 | 0.615±0.012 | 0.592±0.010 | |
| | LAVA | 0.382±0.035 | **0.403**±0.033 | 0.384±0.041 | 0.412±0.011 | **0.673**±0.004 | **0.681**±0.002 | **0.668**±0.009 | 0.599±0.005 |
| | GRADATE | **0.393**±0.040 | 0.387±0.068 | **0.395**±0.040 | | 0.658±0.004 | 0.647±0.005 | 0.642±0.005 | |

Table 8 (ASN / UDAGCN):

| Dataset | GDA Method → | ASN | | | | UDAGCN | | | |
|---|---|---|---|---|---|---|---|---|---|
| | Selection Method ↓ | $\tau=10\%$ | $\tau=20\%$ | $\tau=50\%$ | Full | $\tau=10\%$ | $\tau=20\%$ | $\tau=50\%$ | Full |
| IMDB-BINARY | Random | 0.613±0.110 | 0.568±0.078 | 0.515±0.024 | | 0.507±0.077 | 0.467±0.029 | 0.605±0.067 | |
| | LAVA | 0.817±0.012 | 0.810±0.012 | **0.847**±0.017 | 0.633±0.054 | 0.817±0.012 | 0.811±0.005 | **0.837**±0.017 | 0.688±0.049 |
| | GRADATE | **0.825**±0.007 | **0.819**±0.024 | 0.834±0.012 | | **0.840**±0.008 | **0.831**±0.009 | 0.816±0.016 | |
| IMDB-MULTI | Random | 0.126±0.013 | 0.101±0.058 | 0.156±0.039 | | 0.340±0.061 | 0.306±0.019 | 0.307±0.033 | |
| | LAVA | 0.379±0.050 | 0.445±0.057 | **0.593**±0.004 | 0.372±0.009 | 0.348±0.051 | 0.387±0.093 | **0.519**±0.120 | 0.392±0.046 |
| | GRADATE | **0.425**±0.015 | **0.455**±0.097 | 0.577±0.006 | | **0.390**±0.055 | **0.444**±0.089 | 0.451±0.145 | |
| MSRC_21 | Random | 0.481±0.071 | 0.277±0.039 | 0.556±0.012 | | 0.151±0.072 | 0.204±0.065 | 0.209±0.062 | |
| | LAVA | 0.661±0.027 | 0.779±0.039 | 0.867±0.017 | 0.734±0.015 | 0.435±0.024 | **0.498**±0.090 | 0.563±0.099 | 0.260±0.049 |
| | GRADATE | **0.686**±0.022 | **0.796**±0.020 | **0.868**±0.034 | | **0.465**±0.051 | 0.470±0.097 | **0.616**±0.055 | |
| ogbg-molbace | Random | 0.465±0.048 | 0.440±0.049 | 0.466±0.060 | | 0.485±0.017 | 0.503±0.044 | 0.544±0.011 | |
| | LAVA | 0.496±0.077 | 0.560±0.032 | **0.596**±0.052 | 0.523±0.091 | 0.499±0.036 | 0.553±0.041 | 0.517±0.012 | 0.448±0.020 |
| | GRADATE | **0.565**±0.073 | **0.596**±0.053 | 0.546±0.023 | | **0.521**±0.002 | 0.519±0.022 | **0.555**±0.024 | |
| ogbg-molbbbp | Random | 0.537±0.091 | 0.530±0.076 | 0.545±0.062 | | 0.549±0.031 | 0.568±0.043 | 0.536±0.008 | |
| | LAVA | 0.606±0.024 | 0.635±0.008 | 0.646±0.000 | 0.616±0.042 | 0.655±0.005 | 0.649±0.003 | 0.673±0.011 | 0.513±0.024 |
| | GRADATE | **0.621**±0.016 | **0.640**±0.019 | **0.650**±0.017 | | **0.660**±0.008 | **0.674**±0.011 | **0.677**±0.027 | |
| ogbg-molhiv | Random | 0.385±0.023 | 0.459±0.086 | 0.397±0.070 | | 0.446±0.041 | 0.412±0.021 | 0.409±0.014 | |
| | LAVA | **0.449**±0.058 | **0.465**±0.088 | 0.399±0.074 | 0.519±0.077 | 0.426±0.021 | 0.431±0.042 | **0.433**±0.017 | 0.439±0.034 |
| | GRADATE | 0.435±0.044 | 0.423±0.096 | **0.474**±0.094 | | 0.433±0.020 | **0.434**±0.008 | 0.395±0.015 | |

Table 8: Performance comparison across combinations of GDA methods and data selection methods for *graph size* shift. We use **bold**/underline to indicate the 1st/2nd best results. GRADATE achieves the best performance in most settings.

| Dataset | GCN ($\tau=10\%$) | GCN ($\tau=20\%$) | GCN ($\tau=50\%$) | GIN ($\tau=10\%$) | GIN ($\tau=20\%$) | GIN ($\tau=50\%$) |
|---|---|---|---|---|---|---|
| IMDB-BINARY | +11.13 | +9.92 | +2.76 | +4.63 | +5.16 | +13.37 |
| IMDB-MULTI | +221.31 | +201.09 | +189.61 | +45.78 | +95.01 | +87.04 |
| MSRC_21 | +1.99 | +1.36 | +3.01 | +2.36 | +0.09 | +2.88 |
| ogbg-molbace | +1.32 | -4.39 | +13.17 | -16.32 | +2.84 | +7.48 |
| ogbg-molbbbp | +0.00 | -14.46 | +0.00 | -0.08 | +2.18 | +2.22 |
| ogbg-molhiv | +3.05 | +0.75 | +0.48 | -1.24 | +0.35 | +2.52 |

Table 9: Relative improvement (%) of GRADATE over the strongest baseline under different settings.

for readout. A dropout layer with probability $0.5$ is placed after GIN layers and finally a linear layer with softmax for prediction. (iv) For GraphSAGE, we use two GraphSAGE layers with mean aggregation operation. The hidden dimension is set to 32. A dropout layer with probability $p = 0.5$ is applied after the GCN layers. Finally, a linear layer with softmax is placed at the end for graph class prediction.

*Experiment Details.* We perform all our methods in Python and GNN models are built-in modules of PyTorch Geometric [14]. The learning rate is set to $10^{-2}$ with weight decay $5 \cdot 10^{-4}$. We

| Dataset | SGFormer ($\tau$=10%) | SGFormer ($\tau$=20%) | SGFormer ($\tau$=50%) | APPNP ($\tau$=10%) | APPNP ($\tau$=20%) | APPNP ($\tau$=50%) |
|---|---|---|---|---|---|---|
| IMDB-BINARY | +9.82 | +0.59 | -0.34 | +7.28 | +3.01 | -0.35 |
| IMDB-MULTI | +6.46 | +57.40 | +1.65 | -13.46 | -2.59 | +13.71 |
| MSRC_21 | +14.26 | +3.58 | +2.13 | +9.54 | +3.18 | +1.27 |
| ogbg-molbace | -9.66 | +0.05 | +1.90 | +4.03 | -2.31 | +2.84 |
| ogbg-molbbbp | +2.93 | +5.47 | -3.71 | +22.80 | +6.33 | -4.58 |
| ogbg-molhiv | +1.51 | -1.79 | +1.65 | -12.00 | +6.07 | +1.88 |

Table 10: Relative improvement (%) of GRADATE over the strongest baseline under different settings.

| Dataset | **A2GNN-ADV** ($\tau$=20%) | **A2GNN-MMD** ($\tau$=20%) | **TDSS** ($\tau$=20%) |
|---|---|---|---|
| IMDB-BINARY | +31.12 | +11.40 | +3.12 |
| IMDB-MULTI | +26.54 | +58.88 | -42.10 |
| MSRC_21 | +20.63 | -4.40 | +25.09 |
| ogbg-molbace | +1.46 | -2.57 | +2.88 |
| ogbg-molbbbp | +17.34 | +5.85 | -5.71 |
| ogbg-molhiv | +0.65 | +0.81 | +0.55 |

Table 11: Relative improvement (%) of GRADATE over three additionally added GDA methods under different datasets.

train 200 epochs for datasets IMDB-BINARY, IMDB-MULTI, MSRC_21 and 100 epochs for datasets ogbg-molbace, ogbg-molbbbp, ogbg-molhiv with early stopping, evaluating the test set on the model checkpoint that achieves the highest validation performance during training. For each combination of data and model, we report the mean and standard deviation of classification performance over 3-5 random trials. For TUDatasets, we use accuracy as the performance metric; for OGB datasets, we use AUCROC as the performance metric. The computation is performed on Linux with an NVIDIA Tesla V100-SXM2-32GB GPU. For graphs without node features, we also follow Zeng et al. [81] that generates degree-specific one-hot features for each node in the graphs.

## J   GDA Method-Specific Settings

We follow the default parameter settings in the code repository of OpenGDA [53]. We train 200 epochs for datasets IMDB-BINARY, IMDB-MULTI, MSRC_21 and 100 epochs for datasets ogbg-molbace, ogbg-molbbbp, ogbg-molhiv with early stopping, evaluating the test set on the model checkpoint that achieves the highest validation performance during training.

- AdaGCN [12]: We set the learning rate to $10^{-3}$ with regularization coefficient equal to $10^{-4}$. Dropout rate is 0.3 and $\lambda_b = 1, \lambda_{gp} = 5$.
- ASN [83]: We set the learning rate to $10^{-3}$ with regularization coefficient equal to $10^{-4}$. The dropout rate is 0.5. The difference loss coefficient, domain loss coefficient and the reconstruction loss coefficient is set to $10^{-6}, 0.1, 0.5$.
- GRADE [61]: We set the learning rate to $10^{-3}$ with regularization coefficient equal to $10^{-4}$. Dropout rate is set to 0.1.
- UDAGCN [62]: We set the learning rate to $10^{-3}$ with regularization coefficient equal to $10^{-4}$. The domain loss weight equals to 1.

## K   Additional Preliminary: Graph Domain Adaptation (GDA)

Consider a source domain $\mathcal{D}_s = (\mathcal{G}_i^s, y_i^s)_{i=1}^{n_s}$ and a target domain $\mathcal{D}_t = (\mathcal{G}_i^t, y_i^t)_{i=1}^{n_t}$, where each $\mathcal{G} = (\mathbf{A}, \mathbf{X})$ represents an attributed graph with the adjacency matrix $\mathbf{A} \in \mathbb{R}^{n \times n}$ and the node feature matrix $\mathbf{X} \in \mathbb{R}^{n \times d}$, where $n$ is the number of nodes and $d$ is the dimension of node features. With a

| DATASET | # GRAPHS | # NODES | # EDGES | #FEATURES | # CLASS | DATA SOURCE | LICENSE |
|---------|----------|---------|---------|-----------|---------|-------------|---------|
| IMDB-BINARY | 1000 | 19.77 | 96.53 | None | 2 | PyG [14] | MIT License |
| IMDB-MULTI | 1500 | 12.74 | 53.88 | None | 3 | PyG [14] | MIT License |
| MSRC_21 | 563 | 77.52 | 198.32 | None | 20 | PyG [14] | MIT License |
| ogbg-molbace | 1513 | 34.08 | 36.85 | 9 | 2 | OGB [21] | MIT License |
| ogbg-molbbbp | 2039 | 24.06 | 25.95 | 9 | 2 | OGB [21] | MIT License |
| ogbg-molhiv | 41127 | 25.51 | 27.46 | 9 | 2 | OGB [21] | MIT License |

Table 12: Dataset Statistics and Licenses.

shared label set $\mathcal{Y}$, the graphs in the source domain are labeled with $y_i^s \in \mathcal{Y}$. The two domains are drawn from shifted joint distributions of graph and label space, i.e., $P_s(\mathcal{G}, y) \neq P_t(\mathcal{G}, y)$. The goal of GDA is to learn a classifier $f : \mathcal{G} \to \mathcal{Y}$ with the source domain data that minimizes the expected risk on the target domain: $\mathbb{E}_{(\mathcal{G},y) \sim P_t}[\mathcal{L}(f(\mathcal{G}), y)]$, where $\mathcal{L}$ is a task-specific loss function.

## L  Additional Related Work

In the era of big data and AI [40, 39, 48–51, 47, 68, 74–76, 67, 33], efficient data utilization has become paramount for scalable machine learning. Beyond the data selection and domain adaptation methods discussed in the main text, our work also connects to several related research directions that provide complementary perspectives on graph learning and distribution matching. Specifically, there is a rich body of work at the intersection of optimal transport theory and graph data [24, 77–80, 82], which are closely related to our main methodology.

## M  Empirical Runtime of GRADATE

In Table 13 and Table 14, we provide the empirical runtime on datasets (IMDB-BINARY, IMDB-MULTI and MSRC_21) and datasets (ogbg-molbace, ogbg-molbbbp and ogbg-molhiv), respectively. We observe that the on-line runtime is insignificant compared to typical GNN training time. And the off-line computation is only run once, which can be pre-computed. Furthermore, we can achieve much better accuracy compared to LAVA with nearly no additional runtime.

| | Procedure / Dataset | IMDB-BINARY | IMDB-MULTI | MSRC_21 |
|---|---|---|---|---|
| Off-line Computation | FGW Pairwise distance | 7.41 | 9.61 | 18.18 |
| (GDD Computation) | Label-informed pairwise distance | 0.04 | 0.06 | 0.24 |
| On-line Computation | GREAT (Algorithm 3) | 0.28 | 0.52 | 0.11 |
| | LAVA | 0.09 | 0.14 | 0.03 |
| GNN Training Time | GCN (w/ 10% data) | 13.45 | 16.36 | 9.59 |
| | GCN (w/ 20% data) | 17.64 | 21.40 | 13.85 |
| | GCN (w/ 50% data) | 29.92 | 45.82 | 19.57 |

Table 13: **Empirical run-time behavior for TUDatasets (in seconds)**. We can observe that the off-line procedures can be run comparable to a single GNN training time and the on-line procedure has a negligible runtime compared to GNN training. In addition, we can achieve significantly better performance compared to LAVA with slight additional on-line runtime.

## N  Limitations and Outlook

1. *How to eliminate the dependence on validation data?* Although it is common in machine learning research to assume we have some validation set that represents the data distribution on the target set (or "statistically closer" to the target set), it might not be always available under certain extreme scenarios. Thus, our interesting future direction is to extend our framework to no-validation-data or test-time adaptation settings.

2. *Can our proposed method scale to extremely large settings?* When we have millions of *large* graphs in both training and validation set, the efficiency of GRADATE might be a concern. However, most of the computationally intensive sub-procedure of our method can be done off-line (see complexity analysis in Section 3.2 and empirical runtime in Appendix M) and

| | Procedure / Dataset | ogbg-molbace | ogbg-molbbbp | ogbg-molhiv |
|---|---|---|---|---|
| Off-line Computation (GDD Computation) | FGW Pairwise distance | 15.44 | 19.67 | 283.99 |
| | Label-informed pairwise distance | 0.12 | 0.17 | 53.82 |
| On-line Computation | GREAT (Algorithm 3) | 0.43 | 0.84 | 295.42 |
| | LAVA | 0.06 | 0.13 | 39.45 |
| GNN Training Time | GCN (w/ 10% data) | 12.76 | 13.91 | 190.34 |
| | GCN (w/ 20% data) | 13.08 | 22.59 | 312.08 |
| | GCN (w/ 50% data) | 22.21 | 30.48 | 845.46 |

Table 14: **Empirical run-time behavior for OGB datasets (in seconds)**. We can observe that the off-line procedures can be run comparable to a single GNN training time and the on-line procedure has a negligible runtime compared to GNN training. In addition, we can achieve significantly better performance compared to LAVA with slight additional on-line runtime.

the online runtime is ignorable compared to typical GNN training on full dataset. One possible mitigation is to do data clustering using FGW distance before running our main algorithm GRADATE to avoid computational overhead.

3. *How to select the optimal amount of data?* We demonstrate that in Section 4.5, the relationship between selection ratio and GNN adaptation performance is not always trivial across different settings. To ease the comparison pipeline, we fix to some target selection ratios (i.e. $10\%, 20\%, 50\%$) for our main experiments, but we acknowledge that these ratios might not yield the best adaptation performance or serve as the best indicator to comparison across different methods. Thus, one potential extension of our method is to automate the process of selecting the optimal selection ratios when dealing different levels of domain shifts.

4. *How to extend* GRADATE *to node-level graph domain adaption setting?* One possible extension to solve node-level tasks is as follows: decomposing source/target graphs into set of ego-graphs where each of these ego-graphs represent the local topology of each node, and applying GRADATE directly to select the optimal nodes for adaptation. Future directions that require more investigation include (i) how to decide the size of local vicinity for each ego-graph, (ii) how to co-consider node/edge selection for optimal adaptation and (iii) how to mitigate the information loss when extracting ego-graphs.

## O   Impact Statement

This paper discusses the advancement of the field of Graph Machine Learning. While there are potential societal consequence of our work, none of which we feel must be hightlighted .

# P    ECDF Plots of Different Covariate Shift Settings

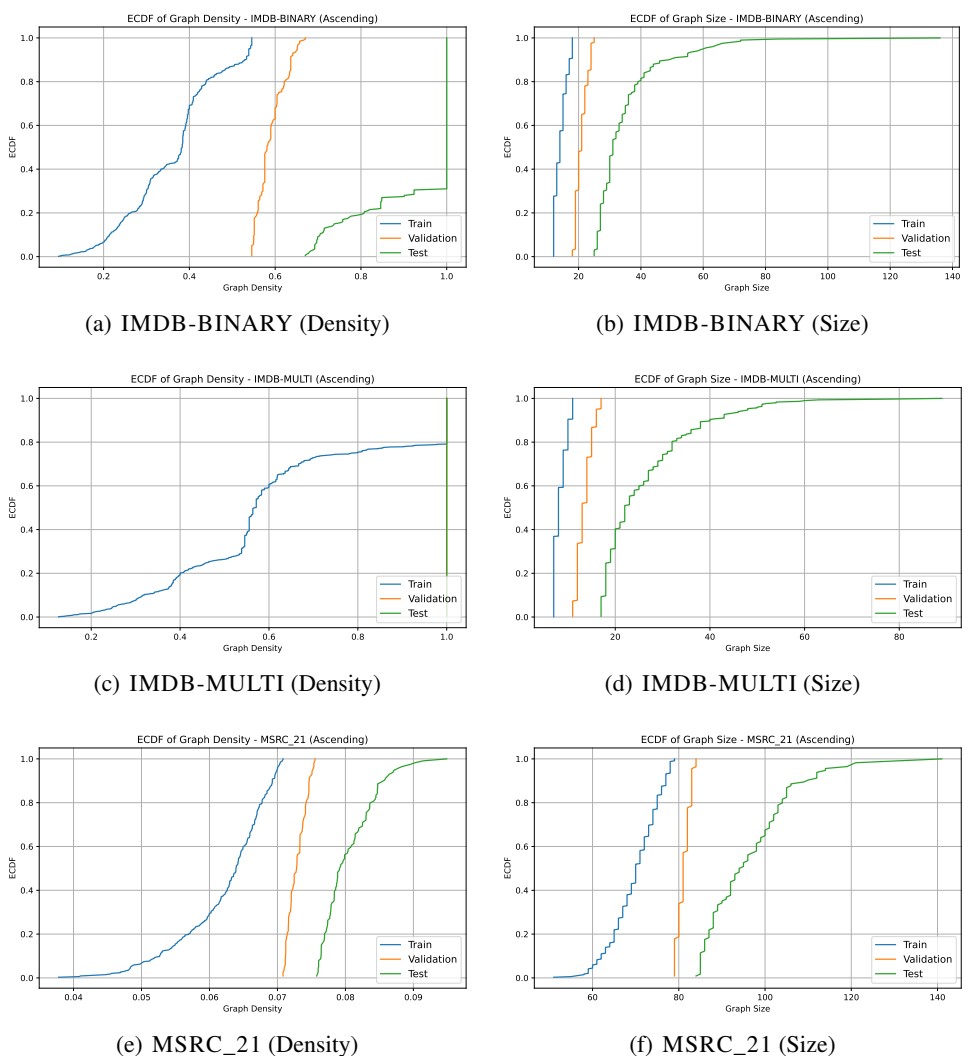

(a) IMDB-BINARY (Density)

(b) IMDB-BINARY (Size)

(c) IMDB-MULTI (Density)

(d) IMDB-MULTI (Size)

(e) MSRC_21 (Density)

(f) MSRC_21 (Size)

Figure 1: **ECDF plots of graph density and size for IMDB-BINARY, IMDB-MULTI, and MSRC_21 datasets**. The Blue, Orange, and Green curves represent the distributions of the training, validation, and test splits, respectively. Graphs are sorted in the ascending order by the specified shift (density or size).

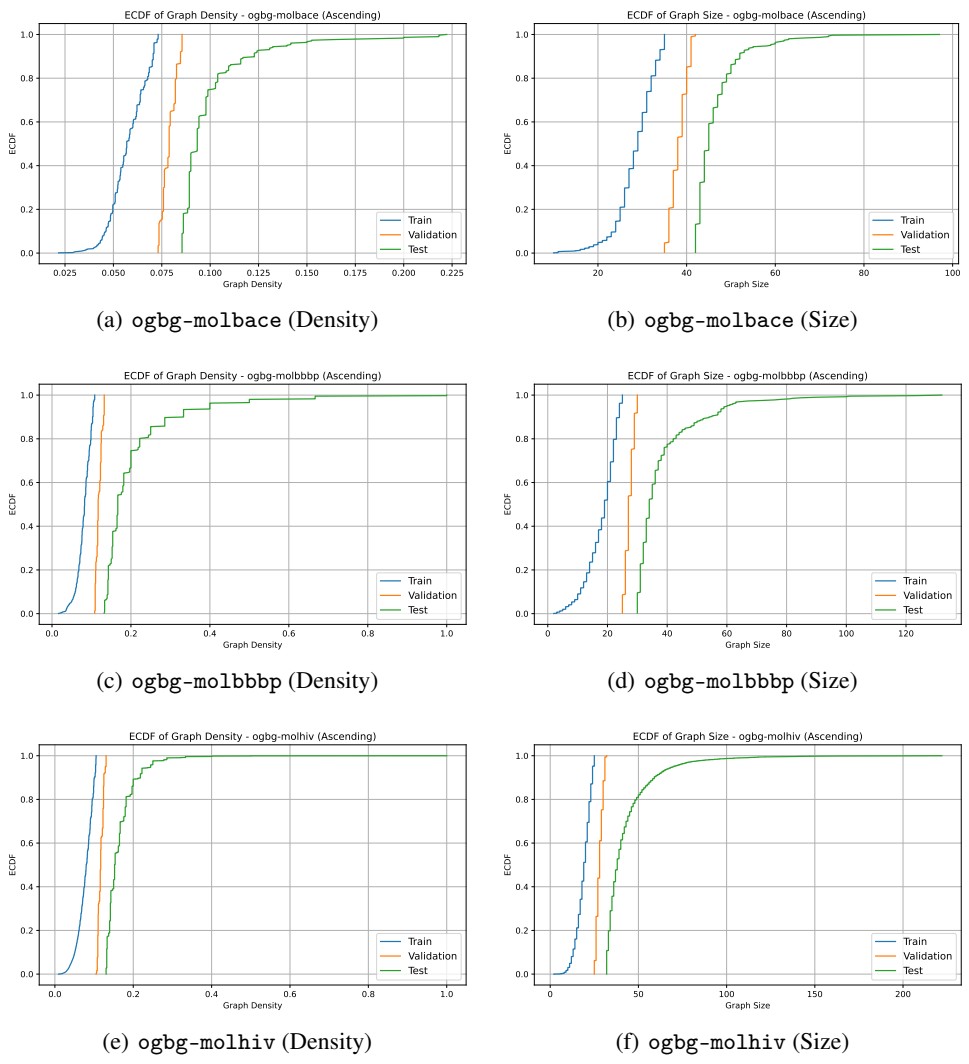

(a) `ogbg-molbace` (Density)

(b) `ogbg-molbace` (Size)

(c) `ogbg-molbbbp` (Density)

(d) `ogbg-molbbbp` (Size)

(e) `ogbg-molhiv` (Density)

(f) `ogbg-molhiv` (Size)

Figure 2: **ECDF plots of graph density and size for** `ogbg-molbbbp`**,** `ogbg-molbace`**, and** `ogbg-molhiv` **datasets**. The Blue, Orange, and Green curves represent the distributions of the training, validation, and test splits, respectively. Graphs are sorted in ascending order by the specified shift (density or size).

