# OpenReview forum: "Graph Data Selection for Domain Adaptation: A Model-Free Approach"
_NeurIPS.cc/2025/Conference — NeurIPS 2025 poster_

### Official Review · Reviewer_ThiT · 2025-06-23

**Clarity:** 3
**Significance:** 2
**Originality:** 3
**Rating:** 3
**Confidence:** 4

**Summary:**

This paper studied graph data selection problem under distribution shift setting. Specifically, the authors proposed a data-centric rather than model-centric framework. They designed Graph Dataset Distance inspired by FGW distance, and utilized optimal transport to find the most adaptive source data by minimizing GDD.

**Questions:**

1. GDA aims to make our model suitable to new data, but theoretically, different models prefer different part of data. So, can we really find a set of training graphs that benifits all models?
2. I understand that with the increase of seleciton ratio, model performance may decrease because of involving irrelevant training data. But I do not understand why the performance firstly decreases and then increases sometimes? For example, GCN+GRADATE on IMDB-MULTI in Table 1, where 20\% was the worst. If GRADATE started to involve irrelevant information at 20\%, 50\% should be lower, since the selection quality continuously descends.

**Ethical Concerns:**

["NO or VERY MINOR ethics concerns only"]

**Final Justification:**

I stick to my negative evaluation.

1. This paper studies graph domain adaption (GDA) problem, which is for a better performance on target domain. The proposed technique was for decreasing domain gap between source and target by reselecting source graphs. But this operation will also change the source performance, so it cannot theoretically guarantee how the target performance changes. Therefore, the designed method is not theoretically correct for GDA.

2. In the introduced GDD, the authors added label distance on FGW. But label distance does not work as shown in the tables in rebuttal, where only 50% cases shows the gains from label distance, while for the other 50% cases, label distance degrads the model. I do not think such random performances can support the usefulness of the proposed label distance, and then GDD.

Therefore, the two main contributions exist flaws.

For the first point, the authors finallly attempted to argue that this paper is for data selection, not for GDA. I am skeptical, because for me, the logic in abstract and introduction follows 1) what's GDA, 2) problems of model-centric methods for GDA, and 3) the authors proposed model-free method to tackle the challenges. \
Anthor point, please refer to line 72-73 in Section 2 Preliminaries, the authors wrote "We also provide the formal problem formulation of graph domain adaptation (GDA) in Appendix K." I do not understand if this paper is not for GDA but data selection, why the authors did not give the formal formulation for the latter?

**Limitations:**

yes.

**Paper Formatting Concerns:**

No.

**Quality:**

2

**Strengths And Weaknesses:**

Strenghts:
1. Clear motivation. The authors listed the drawbacks of model-dependent GDA, i.e., extensive resources and ignorance of source data quality. These drawbacks can actually motivate the model-free framework.
2. Theoretical support. The proposed framework is theorem driven. I roughly checked the correctness. It's generally ok, but I cannot guarantee the details.
3. Reasonable experimental settings. The authors did three types of numerical comparisons, verifying the improvement over data selection and model-centric GDA.

Weaknesses:
1. Theorem 3.3 proved minimizing GDD can decrease the discrepancy between training and validation performances. That's true, but cannot infer that validation loss also decreases. They minimized GDD by selecting training data according to vallidation data, so training loss accordingly changes, then validation loss changes. Therefore, minimizing GDD for a smaller gap cannot theoretically guarantee a better performance on validation.
2. Model complexity. The authors used optimal transportation theory, one of whose characteristics is high complexity with O(N^2) (or O(NM) in this paper).
3. Experiments. 1) All six datasets are small-scale ones. 2) Potential unfair comparison. Proposed GRADATE uses both training and validation labels to update the weight when $c\neq0$. The full-data training GNNs in table 1 and full-data training GDAs in table 3 only use training data. So, it's not clear the improvements come from whether the extra validation labels or not.
4. GDD incrementally adds label distance based on traditional FGW. The label distance is defined in the same way as FGW.

---

> ### Author Rebuttal · Authors · 2025-07-31
>
> We thank the reviewer for the insightful questions. We address each of those as follows.
>
> > **(W1) Theorem 3.3 proved minimizing GDD can decrease the discrepancy between training and validation performances. That's true, but cannot infer that validation loss also decreases. They minimized GDD by selecting training data according to validation data, so training loss accordingly changes, then validation loss changes. Therefore, minimizing GDD for a smaller gap cannot theoretically guarantee a better performance on validation.**
>
>
> This is a great observation! We acknowledge that the upper bound of validation loss might end up going up if the training loss is increased significantly during the selection process. However, we empirically observe that mostly, both the training and the validation loss decrease.
>
> To illustrate this, we train a GIN model with 50% GRADATE-selected data and compare it with a counterpart trained with full data. In Table 1, we report the **relative change of train/validation loss**. For example, Δ train loss = (train loss of 50%-data model - train loss of full model) / train loss of full model; and Δ validation loss is calculated similarly. Thus, **a more negative number means GRADATE lowers the train/validation loss more**. We can observe that under most cases, not only validation loss drops, the training loss also decreases if we train with 50% high-quality data selected by GRADATE.
>
>
> - **Table 1 (GRADATE data decreases train \& validation losses)**
>
>
> | Dataset | Δ train loss | Δ validation loss |
> |:-----------:|:-----------:|:-----------:|
> |  IMDB-BINARY  | -38.3%  | -29.2% |
> |  IMDB-MULTI   | -30.7%  | -0.45% |
> |  MSRC_21      | -60.4%  | +12.7% |
> |  ogbg-molbace | +17.1%  | -31.4% |
> |  ogbg-molbbbp | -79.3%  | -37.7% |
> |  ogbg-molhiv  | -64.1%  | -3.78% |
>
>
>
> > **(W2) Model complexity. The authors used optimal transportation theory, one of whose characteristics is high complexity with O(N^2) (or O(NM) in this paper).**
>
> Thanks for raising the concern. We would like to clarify that the `O(NM)` complexity is mainly due to the pairwise distance computation between each of `N` training graphs and each of `M` validation graphs, which has nothing to do with optimal-transport-based distance itself. With linear optimal transport theory and entropic regularization, the empirical runtime of GRADATE is very affordable compared to GNN training time, which can be observed in Section L (Table 10 in line 724). Additionally, runtime on the other three OGB datasets is reported above in the response to **Reviewer pN6d (Table 3 \& W4)**.
>
>
> > **(W3) Experiments. 1) All six datasets are small-scale ones. 2) Potential unfair comparison. Proposed GRADATE uses both training and validation labels to update the weight when `c ≠ 0`. The full-data training GNNs in table 1 and full-data training GDAs in table 3 only use training data. So, it's not clear the improvements come from whether the extra validation labels or not.**
>
> We address each concern as follows.
>
> - **Scale of the datasets**: We would like to argue that we have included significantly larger-scale analysis compared to several other recent works on graph domain adaptation, such as [1,2,3,4]. The largest dataset they use contain at most 2-3K samples; while the largest one (i.e. the *ogbg-molhiv* dataset) we experiment on contains ~42k samples.
>
>
> - **Potential unfair comparison**: As demonstrated in Remark 3.2 (line 165), we have the ability to set `c = 0` to avoid any usage of validation label. To fairly address this concern, we restrict GRADATE to this validation-label-free setting and re-evaluate its performance. **After restricting to this setting, the advantage of GRADATE remains similar to, or even better than, what is reported in paper.** Please refer to Table 1 \& 2 in the response to Reviewer pN6d (W3) for more details.
>
>
> > **(W4) GDD incrementally adds label distance based on traditional FGW. The label distance is defined in the same way as FGW.**
>
> We respectfully disagree with the assessment that GDD merely adds a label distance term on top of FGW in an incremental manner. The novelty of GDD lies not only in considering FGW and label distance, but in introducing **a self-contained framework grounded in theoretical insights that motivates a new data-centric method for graph domain adaptation.**  We summarize on the key components of GDD as follows:
>
> - **Theoretical motivation**: By Theorem 3.3, GDD upper-bounds the domain generalization gap, which leads to a principled data selection objective: minimizing GDD to align source and target distributions. This offers a theoretically grounded alternative to existing model-centric approaches.
>
> - **Justification of FGW**: Rather than using FGW as a black-box metric, we provide Theorem 3.1 as a novel theoretical foundation. Specifically, we prove that FGW distance captures discrepancies between attributed graphs in a way that upper-bounds the representation shift induced by any k-layer GNN. To the best of our knowledge, this is the first time to establish such a guarantee for FGW.
>
> - **Label distance formulation**: The label distance in GDD is not simply adopted from prior work. It is defined as the Earth Mover’s Distance (Wasserstein-1 distance) over graph-label distributions, using FGW as the cost metric. This constitutes the first formal framework for comparing distributions over labeled graphs.
>
> > **(Q1) GDA aims to make our model suitable to new data, but theoretically, different models prefer different part of data. So, can we really find a set of training graphs that benefits all models?**
>
>
> We agree with the reviewer that **no single set of data samples is optimal across all models**. However, in the presence of significant distributional shift between the source and target domains, certain harmful training samples are likely to negatively impact performance and should ideally be excluded by most algorithms. Nevertheless, in paper, we empirically show in Table 3 that **combined with the same set of GRADATE-selected data, most GDA methods can lead to improved performance**. In our response to Reviewer qLY5 (Q6), we further showcase the robustness of GRADATE-selected data by evaluating on three additional GDA methods and demonstrating their performances can also be boosted by the same set of training graphs.
>
> > **(Q2) I understand that with the increase of selection ratio, model performance may decrease because of involving irrelevant training data. But I do not understand why the performance firstly decreases and then increases sometimes? For example, GCN+GRADATE on IMDB-MULTI in Table 1, where 20% was the worst. If GRADATE started to involve irrelevant information at 20%, 50% should be lower, since the selection quality continuously descends.**
>
> Thanks for pointing out! This is indeed a very interesting observation that we didn't spot previously. One possible explanation is that GRADATE is not perfect in the sense that it gets distracted and selects some *harmful* data in the middle and then switches back to find higher-quality samples later on. We believe this phenomenon might worth further exploration beyond the current scope.
>
> In addition, we believe this is also related to the discussion in Section 4.5 (line 297-300) and one bullet point in Limitation \& Outlook section (line 738-744) where **the optimal amount of data for each dataset/shift type/shift level is different and needed to be configured automatically**. This question will be left as one of our future studies.
>
> [1] https://arxiv.org/abs/2306.04979
>
> [2] https://dl.acm.org/doi/abs/10.1145/3503161.3548012
>
> [3] https://arxiv.org/abs/2307.11341
>
> [4] https://arxiv.org/abs/2306.03256

---

> > ### Author Response · Authors · 2025-08-05
> >
> > Dear reviewer,
> >   Based on the latest email from committee, I would like to initiate a discussion. It will be great to let us know whether our rebuttal has addressed your concerns. Thanks!

---

> > ### Comment · Reviewer_ThiT · 2025-08-06
> >
> > Thanks for authors' responses. After reading, I still respectfully have the following concerns,
> >
> > 1. Only experimental results are not enough. The proposed GRADATE is fully supported by theorem 3.3. So, if theorem 3.3 cannot strictly benefit target domain, the performace of GRADATE is uncertain. Although the new table 1 partially showed the benefit, we cannot enumerate every dataset in the world. For me, this is a **theorem-driven method**, so the correctness of theorem is the base of model success.
> >
> > 2. In response to W3, the authors claimed that restricting $c=0$ can match or even outperform $c\neq 0$. 1) It's better to also report the percentage of $c\neq 0$ in table 1 & 2 for Reviewer pN6d, so I can directly compare the improvements brought by $c=0$ and $c\neq 0$. 2) Assumed this statement is true, can I understand the FGW is the dominated one, while  the newly proposed label distance in GDD did not contribute too much? Because the absence of label distance did not influence or even promote the performance.
> >
> > 3. I am not fully convinced by the response to Q2. Besides, I further observe that in table 1, GCN+GRADATE on IMDB-MULTI, ogbg-molbace and ogbg-molhiv decreased and then increased, but GIN+GRADATE increased monotonously. This may be an evidence for my Q1 that it's not optimal to use the same set of training graphs for all models. So can we really talk about GDA without the consideration of specific base model, the "model-free" setting in the title?

---

> > > ### Author Response · Authors · 2025-08-06
> > > **Clarification on the model-free setting**
> > >
> > > We thank the reviewer again for the constructive feedback.
> > >
> > > As the main concern from the reviewer, we would like to discuss our **model-free setting** as a whole before stepping into each question. In particular, we respectfully argue that **the reviewer might have some mis-understanding and over-expectation on GRADATE as well as the "model-free" setting**, which results in concern raised in Q1 and Q3. We have the following statements:
> > >
> > >
> > > 1. **Model-free is a property of GRADATE**: We re-state that the term "model-free" is used to describe our data selection process (i.e. GRADATE) since it does not depend on any GNN model training and learnable parameter. **Thus, our title is valid since this term correctly illustrates our main method's intrinsic property.**
> > >
> > >
> > >
> > > 2. **Over-expectation on the model-free GRADATE from the reviewer**:
> > >     - **(2.1) Validation loss**:  With Theorem 3.3, **we didn't claim that training with GRADATE-selected data can provably always lead to better validation loss.** In fact, only a model-centric method can do. (i.e. jointly optimize the right-hand side of Theorem 3.3, which equals to a model-specific training loss plus the GDD term) On the contrary, GRADATE can only explicitly optimize the GDD term due to its model-free nature. We thus showed in Table 1 during first-round rebuttal that *only optimizing GDD term can empirically decrease both training and validation losses, which should be taken as a plus instead of a weakness or concern of GRADATE.*
> > >     - **(2.2) GRADATE-selected data**: Similarly, **we didn't claim the optimality of GRADATE across all possible combinations of datasets and graph models**. When showcasing Table 1 in the paper, **we always compare GRADATE against other selection baselines while fixing the same GNN backbone**. We understand the reviewer is expecting the optimality of any GRADATE-selected subset and seeking a consistent trend along different selection ratios (ex: decrease-then-increase, always-increase, etc) across all backbone models and datasets. Instead of sticking to a single optimal subset, we would like the reviewer to notice the fact that across shift types, GNN backbones and datasets, **training with a certain small subset of GRADATE-selected samples can outperform training with the full dataset**, which should be well demonstrated in Table 1 and 3. While we leave the question of "what is the optimal amount of data to retain for each scenario" as an important future direction, the above phenomenon should already showcase that under distribution shift, model-free selection is possible and empirically useful.
> > >
> > > 3. **The main takeaway**:
> > >     - **(3.1)** While GRADATE is not theoretically guaranteed to achieve the above two strict requirements (i.e. 2.1 and 2.2) under a model-free setting, we show that GRADATE can benefit prevalent model-centric graph domain adaption methods across various shift types and diverse datasets while outperforming other selection methods.
> > >     - **(3.2)**  **Whether a model-specific/dependent data selection algorithm can outperform GRADATE or not** does not harm the practical usefulness of GRADATE and the search of such algorithm is actually out of scope of this work. These model-centric algorithms are under different assumption/paradigm with GRADATE and are not directly comparable against GRADATE.
> > >
> > >
> > > We hope these clarifications can make the reviewer better position GRADATE and its empirical strength. While we already answer some concerns presented in Q1 and Q3, we will follow-up for a more detailed answer for each of Q1-Q3 in later comments (just to avoid overwhelming the reviewer). Thanks for the understanding in prior!

---

> ### Author Response · Authors · 2025-08-07
>
> Based on the previous discussion, we would like to reply each of the reviewer's concern.
>
> > (Q1) Only experimental results are not enough. The proposed GRADATE is fully supported by theorem 3.3. So, if theorem 3.3 cannot strictly benefit target domain, the performace of GRADATE is uncertain. Although the new table 1 partially showed the benefit, we cannot enumerate every dataset in the world. For me, this is a theorem-driven method, so the correctness of theorem is the base of model success.
>
> Firstly, we respectfully disagree reviewer's statement on *"whether theorem 3.3 can or cannot strictly benefit target domain affects the correctness of theorem 3.3".* We would like to argue that **the correctness of Theorem 3.3 is always valid**. Recall that this theorem (named as **Graph Domain Generalization Gap**) mainly focuses on building up the connection of domain generalization gap and Graph Dataset Distance (GDD). The possibility that `val_loss (model trained with any selected subset of data) < val_loss (model trained with full data)` should not break the validity of the theorem at all.
>
> Together with **Discussion (2.1)** in our previous comment, we hope these statements could address the over-expectation and possible mis-understanding from the reviewer.
>
>
> > (Q2) In response to W3, the authors claimed that restricting `c=0` can match or even outperform `c ≠ 0`. 1) It's better to also report the percentage of `c ≠ 0` in table 1 & 2 for Reviewer pN6d, so I can directly compare the improvements brought by `c = 0` and `c ≠ 0`. 2) Assumed this statement is true, can I understand the FGW is the dominated one, while the newly proposed label distance in GDD did not contribute too much? Because the absence of label distance did not influence or even promote the performance.
>
> We provide the same table 1 & 2 for Reviewer pN6d with `c ≠ 0` here for a more direct comparison as the reviewer requested.
> In addition, we have the following three comments:
> - We apologize for the confusion on using the term "sometimes even outperform". What we mean here is empirically we observe adding label information does not improve the performance **everywhere**.
> - We attribute this phenomenon to the fact that GRADATE is approximated, which affects the accuracy of label distance. Specifically, the usage of entropic regularization in FGW computation and the linear optimal transport both introduce noise.
> - We thank the reviewer for scrutinizing the empirical usefulness of label signal. We will discuss this possible discrepancy in Section 4.5 as well.
>
>
> **Table 1 (GRADATE vs other data selection methods)**
>
> | Dataset   | GCN (10%)  | GCN (20%) | GCN (50%) | GIN (10%) | GIN (20%) | GIN (50%)
> |:-----------:|:-----------:|:-----------:|:-----------:|:-----------:|:-----------:|:-----------:|
> | IMDB-BINARY | +11.53%  | +8.64%    |  +2.53%   | +2.96%     |  +2.76%   | +9.12%     |
> | IMDB-MULTI | +240.12%   | +192.34%    |  +229.51%   | +3.82%    |  +47.97%   | +90.84%     |
> |  MSRC_21  | -0.03%  | +3.76%     |  +1.22%   | -6.55%     |  +4.24%   | +4.58%     |
> |  ogbg-molbace  | +3.82%  | -0.09%    |  +2.19%   | +0.07%     |  +6.28%   | +3.22%     |
> |  ogbg-molbbbp  | +6.58%  | +10.47%     |  +9.21%   | +6.13%    |  +4.47%  | -4.43%     |
> |  ogbg-molhiv  | +0.06%  | -2.60%     |  -0.29%   | +4.23%     |+2.19%   | +4.44%     |
>
> **Table 2 (GRADATE vs GDA baselines)**
>
> | Dataset   | Best GDA| A2GNN-ADV| A2GNN-MMD | TDSS | Best Random| Best LAVA | GRADATE
> |:-----------:|:-----------:|:-----------:|:-----------:|:-----------:|:-----------:|:-----------:|:-----------:|
> | IMDB-BINARY| 0.822 | 0.831 | *0.845* | 0.753 | 0.738 | 0.837 | **0.852**
> | IMDB-MULTI| 0.123 | 0.099 | 0.142 | 0.136 | 0.180 | *0.798* | **0.807**
> | MSRC_21| *0.833*| 0.824 | 0.818 | *0.833* | 0.801 | *0.833* | **0.877**
> | ogbg-molbace | **0.683** | 0.602 | 0.590 | 0.641 | 0.622 | 0.612 | *0.645*
> | ogbg-molbbbp | **0.778** | 0.528 | 0.564 | 0.560 | 0.528 | 0.641 | *0.644*
> | ogbg-molhiv | 0.567 | 0.622 | 0.620 | 0.466 | 0.602 | *0.624* | **0.633**
>
>
> > (Q3) I am not fully convinced by the response to Q2. Besides, I further observe that in table 1, GCN+GRADATE on IMDB-MULTI, ogbg-molbace and ogbg-molhiv decreased and then increased, but GIN+GRADATE increased monotonously. This may be an evidence for my Q1 that it's not optimal to use the same set of training graphs for all models. So can we really talk about GDA without the consideration of specific base model, the "model-free" setting in the title?
>
> We believe this concern should be addressed by **Discussion (1) and (2.2)** in our previous comment.

---

> > ### Comment · Reviewer_ThiT · 2025-08-07
> >
> > Thanks for further explanations from the authors. However, the new ones still did not convince me.
> >
> > 1. This paper studies GDA problem, and the core of GDA is for a better validation loss under shiftings. Please refer to italics in line 27-28, $\textit{How to ... , for better graph-level classification accuracy evaluated on the target domain?}$, where the authors also agreed this point. But, the authors newly acknowledged the proposed "GRADATE can only explicitly optimize the GDD (domain gap)" rather than better validation loss. I definitely agree that Theorem 3.3 is correct if we only consider decrease domain gap, but decreasing domain gap does not equal to a better validation loss here in my initial opinions, which the authors acknowledged is "a great observation".\
> > In a word, GRADATE is based on minimizing GDD, but GDD is for domain gap, not for better validation loss (because of changing source domain), thus not for GDA.
> >
> > 2. The proposed GDD=FGW+label distance (if $c\neq0$), where FGW was known before. Comparing the Table 1&2 for me and Table 1&2 for Reviewer pN6d, I found the probability that $c\neq0$ outperforms $c=0$ in all cases is 50% to 50%. So gains from the newly added "label distance" is very random. I concern if such random contribution is meaningful. The FGW is enough without label.
> >
> > I will keep my score.

---

> > > ### Author Response · Authors · 2025-08-08
> > >
> > > We first thank the reviewer again for deeply engaging the discussion with us.
> > >
> > > Regarding the two concerns, we would like to kindly argue the following points to clarify the positioning of this paper.
> > >
> > > > (Q1)
> > >   - **The correctness of Theorem 3.3 holds regardless of how we optimize the bound**:  The reviewer states the following: "*I definitely agree that Theorem 3.3 is correct if we only consider decrease domain gap*", which is unfortunately a misunderstanding. As titled, Theorem 3.3 showcases a bound on graph domain generalization gap but does not indicate anything about the optimization process (i.e. whether we optimize training loss or GDD term or none).
> > >
> > >   - **GRADATE only optimizes GDD but remains empirically strong**: Again, why GRADATE can only optimize GDD term is **due to our strict model-free constraint**. However, the success of GRADATE (main tables in the paper) and empirical decrease of train/validation loss (presented in Table 1 during first-round rebuttal) suggest that **under distribution shift, it is possible to get rid of harmful training data without learnable parameters, resulting a simpler training task (i.e. decreased training loss) and better generalization performance (i.e. empirical advantage of GRADATE)**.
> > >
> > >    - **GRADATE's effectiveness is beyond mere GDA**: Our core algorithm, GRADATE (Graph Data Selector), is itself a *data selection algorithm*. Direct GDA is one of its many applications. To be concrete, among our three main tables in the paper (i.e.  Table 1, 2, 3), **only Table 2 directly compares (GRADATE + vanilla GNNs) against (model-centric GDA baselines)**. While in Table 1 and 3, GRADATE's competitors are other selection methods, but not other GDA baselines. The two tables showcase the empirical strength of GRADATE from the aspect of **promoting source-only GNNs and enhancing existing GDA baselines, respectively, which spans beyond direct GDA.** (Please see line 52-65 for a summary of our main contribution)
> > >
> > >
> > > Combing the above bullet points, we kindly invite the reviewer to re-evaluate our work based on the understanding that
> > >
> > >   - Theorem 3.3 is proposed to connect GDD and graph domain generalization gap. Optimizing solely GDD should not be taken as a weakness but rather a constraint under a more challenging setting we study.
> > >   - Despite the strict model-free constraint during data selection, GRADATE can beat model-dependent GDA baselines.
> > >   - The effectiveness of GRADATE extends beyond serving as yet another GDA algorithm.
> > >
> > >
> > > > (Q2)
> > >   - **Label distance is conceptually novel**: Label distance and the resulting GDD is designed to formally define the distance between sets of labeled graphs, which has not been studied in previous literature. The contribution should be concrete in its own right. **With this, whether it always helps GDA (or other downstream tasks) universally should not diminish its novelty.**
> > >   - **Label signal as a hyper-parameter**: If the reviewer worries about the empirical advantage of incorporating label information, another thing we want to point out is that we always fix label signal `c=5` when `c≠0` throughout the paper and rebuttal mainly for simplicity. We believe a more principled approach to set `c` might be possible to fullfil the requirement of "`c≠0` always dominates `c=0`". Here, we should leave it as an interesting object requiring further study, not as a weakness of label distance itself.
> > >   - While we believe requiring incorporating labels to excel under all the settings might be too strict and `c` can be seen as a hyper-parameter, we admit this empirical phenomenon and we will be happy to discuss more in the Limitation section and Section 4.5.

---

> > > > ### Comment · Reviewer_ThiT · 2025-08-08
> > > >
> > > > Again, this paper studied GDA problem. Abstract section talked about what's GDA, the problems in model-centric methods for GDA, so the authors chose model-free setting. Therefore, all designed technique should serve GDA, because GDA is the target of this paper, while model-free data selection is just a tool to get the target. So-called "strict model-free constraint" is just the authors' choice to deal with GDA, but unfortunately, we both acknowleged this framework and its theorem do not directily tackle GDA. If the authors tell another story, like how to select data, rather than GDA, the logics of GDD and GRADATE are ok.
> > > >
> > > > If the "novel concept" cannot bring improvements, this concept is not practical, and no contribution to the community. I understand that $c$ is a hyper-parameter, and maybe some $c$ can always beat $c=0$, but before finding such $c$, the "novel concept", or the label distance, is hard to be viewed as an effective contribution.

---

> ### Author Response · Authors · 2025-08-08
>
> We thank the reviewer for the quick response. We address the concerns as follows:
>
> 1. While the reviewer agrees with us that **"This paper studied graph data selection problem under distribution shift setting"** (written in the reviewer's Summary section), we are regretful that the emphasis on GDA in our abstract still gives the reviewer an impression that we are exclusively addressing a GDA problem. We would like to point the reviewer to line 5-8 (we propose...) in abstract and also our central question (line 27-28) in the introduction section, namely,
>
> > **How to select** the most relevant source domain data, based on available validation data, for better graph-level classification accuracy evaluated on the target domain?
>
> *We would like to clarify that the answer to this question should not be another GDA algorithm, but should be a data selection method that works well in a domain shift setting.* GRADATE is a direct answer to this **how to select** question. **How to select is always the core storyline.** And this question should be well answered by our Table 1 and 3.
>
> Then what is the role of GDA? We would say GDA is one of the important applications to evaluate the downstream utility of GRADATE. Table 2 deals with the comparison against GDA.
>
> As a final remark, we are happy to slightly down-weight GDA-related sentences in abstract to avoid similar confusion from the reviewer. However, **it should be clear from Introduction (line 27-31, line 45-51) and all main tables in the Experiment that we are indeed studying a "data selection problem".**
>
> 2. For the label distance, we respectfully disagree that it is a must to find a single `c≠0` that swipes all `c=0` cases to justify label distance's practical usefulness. Such `c` might be dataset, shift type-dependent, which should not be a central question to answer in this work. Instead, we show that by only searching within two possible values of `c`, GRADATE's advantage over existing baseline is already evident.
>
> As the rebuttal period comes to an end, we sincerely thank the reviewer's deep engagement to let us clarify and strengthen our work. Thanks!

---

### Official Review · Reviewer_qLY5 · 2025-06-30

**Clarity:** 2
**Significance:** 2
**Originality:** 2
**Rating:** 4
**Confidence:** 4

**Summary:**

This paper studies model-free graph data selection under distribution shift. The authors claim that existing methods belong to model-centric approaches and they often struggle with severe shifts and constrained computational resources. To address these issues, the authors proposed to select the best training data from the source domain for the classification task on the target domain. Specifically, it utilizes optimal transport theory to pick training samples without relying on any GNN model predictions, which is data-efficient. Comprehensive experiment results on several graph level datasets demonstrate that the proposed model can outperform existing selection methods and enhance existing GDA methods.

**Questions:**

1.	The model is only evaluated on graph level task. It is unclear whether the proposed model could work in node level task.
2.	The authors claim that the proposed model is data efficient. However, based on the experimental results, sometimes using more data could achieve better performance.
3.	The authors claim that the proposed model is model-free data selection. However, lots of other GNN architectures are not evaluated including graph transformer [1] and PPNP [2].
4.	In Table 5, the GAT results for ogbg-molhiv are missing.
5.	For some datasets like IMDB-MULTI, using full data could result in worse performance. However, the authors did not give any explanations on this phenomenon.
6.	There are lots of other unsupervised graph domain adaptation baselines. It is unclear whether the proposed model also enhance their performance [3,4,5,6].
7.	It would be valuable to understand how GRADATE performs in extremely low-resource scenarios. For instance, what happens if only 1% of the selected data is used for training?

References:
[1] Chen J, Gao K, Li G, et al. NAGphormer: A Tokenized Graph Transformer for Node Classification in Large Graphs[C]//The Eleventh International Conference on Learning Representations. 2023.
[2] Gasteiger J, Bojchevski A, Günnemann S. Predict then Propagate: Graph Neural Networks meet Personalized PageRank[C]//International Conference on Learning Representations. 2018.
[3] Liu S, Zou D, Zhao H, et al. Pairwise alignment improves graph domain adaptation[J]. arXiv preprint arXiv:2403.01092, 2024.
[4] Liu M, Fang Z, Zhang Z, et al. Rethinking propagation for unsupervised graph domain adaptation[C]//Proceedings of the AAAI Conference on Artificial Intelligence. 2024, 38(12): 13963-13971.
[5] Chen W, Ye G, Wang Y, et al. Smoothness really matters: A simple yet effective approach for unsupervised graph domain adaptation[C]//Proceedings of the AAAI Conference on Artificial Intelligence. 2025, 39(15): 15875-15883.
[6] Fang R, Li B, Kang Z, et al. On the benefits of attribute-driven graph domain adaptation[J]. arXiv preprint arXiv:2502.06808, 2025.

**Ethical Concerns:**

["NO or VERY MINOR ethics concerns only"]

**Final Justification:**

Thanks for the rebuttal. I have adjusted my score.

**Limitations:**

yes

**Quality:**

2

**Strengths And Weaknesses:**

Pros:
1.	This paper proposes a model-free approach, which is new and different from existing model-centric methods.
2.	Theoretical justification is given to guide the design of the proposed components.
3.	The proposed model could enhance the performance of existing unsupervised graph domain adaptation methods.

Cons:
1.	The model is only evaluated on graph level task. It is unclear whether the proposed model could work in node level task.
2.	The authors claim that the proposed model is data efficient. However, based on the experimental results, sometimes using more data could achieve better performance.
3.	The authors claim that the proposed model is model-free data selection. However, lots of other GNN architectures are not evaluated including graph transformer [1] and PPNP [2].
[1] Chen J, Gao K, Li G, et al. NAGphormer: A Tokenized Graph Transformer for Node Classification in Large Graphs[C]//The Eleventh International Conference on Learning Representations. 2023.
[2] Gasteiger J, Bojchevski A, Günnemann S. Predict then Propagate: Graph Neural Networks meet Personalized PageRank[C]//International Conference on Learning Representations. 2018.

---

> ### Author Rebuttal · Authors · 2025-07-31
>
> Thank you for the thoughtful questions. We address each of these as follows.
>
> > **(W1)(Q1) The model is only evaluated on graph level task. It is unclear whether the proposed model could work in node level task.**
>
> We agree with the reviewer that node-level task is also an important aspect of graph domain adaptation. In this work, we focus specifically on the graph-level classification task, which poses unique challenges such as distribution shifts at the graph level. Thus, we believe that addressing graph-level adaptation task is valuable in its own right, as it is essential for real-world applications like molecule property prediction where the prediction target is naturally graph-level.
>
> Although this work explicitly deals with graph-level tasks, one natural extension to solve node-level task is as follows: decomposing both source and target graphs into set of ego-graphs where each of these ego-graphs represent the local topology of each node, and applying graph-level GRADATE to select a subset of nodes from source graph for optimal adaptation.
>
> We acknowledge that this direct extension is non-trivial and requires further investigation into several aspects: (i) how to determine the appropriate size of the local vicinity for each ego-graph, (ii) how to jointly consider node and edge selection for optimal adaptation, and (iii) how to mitigate potential information loss when extracting ego-graphs. We will add this discussion to the Limitations & Outlook section to inspire future research in this direction.
>
> > **(W2)(Q2) The authors claim that the proposed model is data efficient. However, based on the experimental results, sometimes using more data could achieve better performance.**
>
> - **Clarification**: We would like to clarify that *data efficiency* does not necessarily imply *less data always leads to better performance*. We fully acknowledge that, in some cases, training on 50% or even 100% of the data can result in stronger performance. However, our key observation is that GRADATE is able to identify a small subset of high-quality training data that is sufficient to match, or even surpass, the performance of full-data training in many settings. For example, by Table 1 \& 2 in the paper, we show that **GRADATE can often achieve comparable or superior results using as little as 20% of the training data.**
>
> - **Related discussion**: Another question to ask here is: *what is the optimal amount data to select for each setting?* We note this in Section M (line 738-744) as one limitation of GRADATE and will leave this as a direction for future work.
>
>
> > **(W3)(Q3) The authors claim that the proposed model is model-free data selection. However, lots of other GNN architectures are not evaluated including graph transformer and PPNP.**
>
> We conduct extra experiments on one graph transformer (i.e. SGFormer [1]) and APPNP [2]. In Table 1, we report **relative performance improvement (%)** of GRADATE over the strongest baseline under all settings (i.e. *GNN + X% selected data with Dataset Y*). We can observe that GRADATE outperforms other baselines mostly.
>
> - **Table 1 (GRADATE outperforms other data selection methods)**
>
> | Dataset   | SGFormer (10%) | SGFormer (20%) | SGFormer (50%) | APPNP (10%) | APPNP (20%) | APPNP (50%)
> |:-----------:|:-----------:|:-----------:|:-----------:|:-----------:|:-----------:|:-----------:|
> | IMDB-BINARY | +9.82%  | +0.59%   | -0.34% | +7.28%   | +3.01%     |  -0.35%   |
> | IMDB-MULTI | +6.46%   | +57.40%    |  +1.65%   | -13.46%    | -2.59%   | +13.71%     |
> |  MSRC_21  | +14.26%  | +3.58%     |  +2.13%   | +9.54%     |  +3.18%   | +1.27%     |
> |  ogbg-molbace  | -9.66%  | +0.05%    |  +1.90%   | +4.03%     |  -2.31%   | +2.84%     |
> |  ogbg-molbbbp  | +2.93%  | +5.47%     |  -3.71%   | +22.8%    |  +6.33%  | -4.58%     |
> |  ogbg-molhiv  | +1.51%  | -1.79%     |  +1.65%   | -12.00%     |+6.07%   | +1.88%     |
>
>
> > **(Q4) In Table 5, the GAT results for ogbg-molhiv are missing.**
>
> We apologize for the mistake. The number will be re-filled in the updated manuscript.
>
> > **(Q5) For some datasets like IMDB-MULTI, using full data could result in worse performance. However, the authors did not give any explanations on this phenomenon.**
>
> We assume the reviewer is referring to Table 1 and Table 3 in the paper.
>
> - For Table 1, the main reason is that because **under distribution shift, a certain number of harmful graphs exist in training set. These samples can mislead the model and degrade generalization performance.** GRADATE mitigates this issue by selecting a high-quality subset of training graphs that are better aligned with the target domain. As a result, models trained on GRADATE-selected data can outperform those trained on the full dataset.
> - For Table 3, the above reasoning also holds similarly. Even equipped with advanced model-centric GDA methods, severe shift level can still hinder their performances. By effective data selection performed by GRADATE, the difficulty of addressing the domain shift can be lowered significantly and thus result in better adaptation performance.
>
> For the updated manuscript, we will state this clearer in Section 4.2 and 4.4, respectively.
>
>
> > **(Q6) There are lots of other unsupervised graph domain adaptation baselines. It is unclear whether the proposed model also enhance their performance.**
>
> Thank you for pointing to these related works. Due to constraint on resource and open-source code availability, we add three additional experiments based on two variants of A2GNN [3] with different losses and TDSS [4]. In Table 2, we report **relative performance improvement (%)** of GRADATE over the strongest baseline under all settings (i.e. *GDA + X% selected data with Dataset Y*). For brevity, we report results with selection ratio equals to 20%. It can be observed that GRADATE consistently provides the most significant enhancements for the three newly evaluated GDA methods compared to other selection baselines.
>
> - Table 2 (GRADATE enhances additional GDA methods)
>
> | Dataset   |A2GNN-ADV (20%)| A2GNN-MMD (20%)| TDSS (20%)|
> |:-----------:|:-----------:|:-----------:|:-----------:|
> |IMDB-BINARY| +31.12% | +11.40%|+3.12% |
> |IMDB-MULTI| +26.54% |+58.88% |-42.1% |
> |MSRC_21| +20.63% |-4.40% |+25.09% |
> |ogbg-molbace| +1.46% | -2.57%|+2.88% |
> |ogbg-molbbbp|+17.34%| +5.85%|-5.71%|
> |ogbg-molhiv|+0.65% |+0.81% |+0.55%|
>
>
> > **(Q7) It would be valuable to understand how GRADATE performs in extremely low-resource scenarios. For instance, what happens if only 1% of the selected data is used for training?**
>
> To showcase the superiority of GRADATE in low-resource scenarios, we have conducted experiments with **1% training data** and three representative backbones, including GAT [5], SGFormer [1] and APPNP [2], on the same six datasets. The baselines are model-free selection methods: random selection and LAVA [6]. In Table 3, we report **relative performance improvement (%)** of GRADATE over the strongest baseline under all settings (i.e. *GNN + 1% selected data with Dataset Y*). We can see that under extreme low-resource scenario, GRADATE can achieve the best performance most of the time.
>
>
> - Table 3 (GRADATE outperforms other selection methods under lower-resource scenario)
>
> | Dataset   |GAT (1%)| SGFormer (1%)| APPNP (1%)|
> |:-----------:|:-----------:|:-----------:|:-----------:|
> |IMDB-BINARY| +10.74% | +3.24%|+6.92% |
> |IMDB-MULTI| +0.00% |+0.00% |+43.36% |
> |MSRC_21| -3.94% |-8.21% |+35.04% |
> |ogbg-molbace| +0.00% | +16.96%|+27.85% |
> |ogbg-molbbbp|+2.19%| -17.21%|+6.60%|
> |ogbg-molhiv|+0.01% |+8.93% |+2.01%|
>
>
>
> [1] https://arxiv.org/abs/2306.10759
>
> [2] https://arxiv.org/abs/1810.05997
>
> [3] https://arxiv.org/abs/2402.05660
>
> [4] https://arxiv.org/abs/2412.11654
>
> [5] https://arxiv.org/abs/1710.10903
>
> [6] https://arxiv.org/abs/2305.00054

---

> > ### Comment · Reviewer_qLY5 · 2025-08-05
> >
> > Thanks for the rebuttal. I have adjusted my score.

---

> > > ### Author Response · Authors · 2025-08-05
> > >
> > > We thank the reviewer for recognizing our work and providing constructive feedback. We also appreciate the reviewer for raising the score. Please let us know if there is any unresolved concern.

---

### Official Review · Reviewer_y91a · 2025-07-01

**Clarity:** 2
**Significance:** 2
**Originality:** 3
**Rating:** 4
**Confidence:** 1

**Summary:**

Graphs have become a fundamental data structure for representing complex relationships, with wide applications in areas such as molecular interaction modeling and recommendation systems. However, in graph-level classification tasks, distribution shift between the source and target domains poses a significant challenge.

Limitations of existing methods:
Although many Graph Neural Network (GNN)-based approaches have been proposed for Graph Domain Adaptation (GDA), most of them rely heavily on model architecture design and training strategies. This model-centric paradigm faces practical issues, such as the need for substantial computational resources to train and validate various architectural variants. Moreover, it often overlooks the quality of the source data. As a result, these methods tend to perform poorly under severe distribution shifts or limited computational budgets.

To address these limitations, this paper aims to answer a core question:
How can we select the most relevant training data from the source domain—based on available validation data—in order to improve graph-level classification accuracy on the target domain?

The paper proposes a novel model-free framework named GRADATE (GRAph DATa sElector), which aims to select the most relevant training data from the source domain to improve classification performance on the target domain.

**Questions:**

See weaknesses.

**Ethical Concerns:**

["NO or VERY MINOR ethics concerns only"]

**Limitations:**

yes

**Quality:**

3

**Strengths And Weaknesses:**

Strengths:
- The paper provides solid theoretical insights.
- The introduction is clear and easy to understand.
- The experimental evaluation is comprehensive.

Weaknesses:
- I have some concerns regarding Eq. 5. Is it valid to perform the minimization using validation data? Comparing the distance between training and validation data seems to implicitly treat the validation set as part of the training process. Wouldn't this contradict the standard setup of GNN-based methods, where validation data is only used for model selection, not training?

- In Section 4.2, the model trained on the selected 50% of data outperforms the one trained on the full dataset. Was validation data used as a selection criterion in this filtering process? If so, does this raise concerns about potential information leakage from the target domain?

---

> ### Author Rebuttal · Authors · 2025-07-31
>
> Thank you for recognizing our work! For the weakness and question you mentioned, we address each of those as follows.
>
>
> > **(W1) I have some concerns regarding Eq. 5. Is it valid to perform the minimization using validation data? Comparing the distance between training and validation data seems to implicitly treat the validation set as part of the training process. Wouldn't this contradict the standard setup of GNN-based methods, where validation data is only used for model selection, not training?**
>
>
> Thank you for pointing out. GRADATE will depend on validation **data** but not necessarily validation **label**. As demonstrated in Remark 3.2 (line 165), setting`c=0` will allow GRADATE to be independent of validation **label** during the selection process. Note that the *graph domain adaptation (GDA) baselines also have access to validation data during training*.
>
>
> > **(W2) In Section 4.2, the model trained on the selected 50% of data outperforms the one trained on the full dataset. Was validation data used as a selection criterion in this filtering process? If so, does this raise concerns about potential information leakage from the target domain?**
>
>
> Let us address each question as follows.
>
> - **Yes, the validation data is used during selection.** Similar to our response to (W1), the dependency on validation data matches the setting of graph domain adaptation baselines.
>
> - **No, there is no concern about information leakage.** While GRADATE makes use of validation *data* to select data from training set, the validation *label* is not used when we set `c=0`. Even with the constraint of `c=0`, GRADATE still yields the best performance across settings. (Please refer to Table 1 and 2 in our response to Reviewer pN6d)
>
> - The improved performance of GRADATE with less training data stems from its ability to **retain high-quality samples and eliminate distributionally mismatched ones.** In contrast, training on the full dataset under severe distribution shift can hurt performance due to the presence of misleading samples.

---

> > ### Author Response · Authors · 2025-08-05
> >
> > Dear reviewer,
> >   Based on the latest email from committee, I would like to initiate a discussion. It will be great to let us know whether our rebuttal has addressed your concerns. Thanks!

---

> > ### Comment · Reviewer_y91a · 2025-08-06
> > **Official Comment By Reviewer y91a**
> >
> > Thank you to the authors for their feedback. To be honest, I found the paper difficult to understand—perhaps due to the writing quality, the way the theorems are presented, or simply because I'm not very familiar with the area of graph domain adaptation. As such, I will keep my original score, but I kindly ask the committee to ignore my score, as I do not feel I fully understood the paper.

---

> ### Author Response · Authors · 2025-08-07
>
> We thank the reviewer for the feedback and apologize for any possible lack of clarity in our manuscript. It will be very helpful if the reviewer can point us to certain parts of the paper that confuse you and we are happy to provide a more detailed follow-up after that.
>
> To help the reviewer better understand our paper, we would like to summarize the main message we want to convey:
> We propose GRADATE to take a data-centric perspective to tackle the graph domain adaption problem. Specifically,
> - **GRADATE is theoretically-motivated**:
>   - We show Fused Gromov-Wasserstein (FGW) distance's expressiveness in theorem 3.1 to compare attributed graphs, which is an important building block of GRADATE to compute the proximity between domains.
>   - The newly introduced concept, Graph Dataset Distance (GDD), theoretically bounds the performance generalization gap between source and target domains. Motivated by theorem 3.3, GRADATE selects the best data from the training set to minimize this gap. As a result, we showcase better adaptation performance by merely training on these selected, target-aligned data.
>
> - **GRADATE has strong empirical performance**: Across various domain shift types, datasets and GNN backbones,
>   - Table 1 on page 7 (GRADATE outperforms other data selection methods)
>   - Table 2 on page 8 (GRADATE+vanilla GNNs outperform model-centric graph domain adaption methods)
>   - Table 3 on page 9 (GRADATE-selected data can even further enhance graph domain adaption methods)
>
> For the GDA setting,
>   - We provide a detailed problem definition for graph domain adaptation on page 25.
>   - [1] provides a very concise background (on page 2) on GDA methods that can benefit the understanding of our studied problem.
>
> [1] Graph Domain Adaptation: Challenges, Progress and Prospects

---

### Official Review · Reviewer_pN6d · 2025-07-03

**Clarity:** 3
**Significance:** 3
**Originality:** 3
**Rating:** 4
**Confidence:** 3

**Summary:**

The paper proposes GRADATE, a novel model-free framework for graph domain adaptation that selects useful training data using optimal transport theory. It offers theoretical guarantees for its distance metric, introduces an effective optimization algorithm, and demonstrates strong data efficiency. Extensive experiments on multiple real-world datasets show that GRADATE outperforms existing selection methods and can enhance standard GDA models with less data.

**Questions:**

none.

**Ethical Concerns:**

["NO or VERY MINOR ethics concerns only"]

**Final Justification:**

The authors have addressed my concern by providing additional experimental results. I am therefore setting my final score to 4, leading to acceptance.

**Limitations:**

yes.

**Paper Formatting Concerns:**

none.

**Quality:**

3

**Strengths And Weaknesses:**

Strengths:

1. GRADATE is explicitly model-agnostic, avoiding reliance on any specific GNN architecture.

2. The paper gives a clear theoretical justification for using Fused Gromov-Wasserstein distance. It also provides an explicit bound on the domain generalization gap via the proposed Graph Dataset Distance.

3. GRADATE shows it can achieve strong performance with significantly less training data.

4. The paper includes experiments on six real-world graph-level datasets under multiple types of covariate shift, outperforming state-of-the-art GDA baselines.



Weaknesses:

1. The literature review is somewhat incomplete. The paper does not sufficiently cover some recent graph domain adaptation works.

2. The method is only evaluated on graph-level tasks, but domain adaptation is also important for node-level prediction with established benchmarks. Discussion of whether and how GRADATE could be applied at the node level would be helpful.

3. The approach relies on labeled target-domain validation data, which is often unavailable in realistic test-time adaptation scenarios.

4. Despite approximations, the method’s time complexity remains high and could limit scalability.

---

> ### Author Rebuttal · Authors · 2025-07-31
>
> Thank you for recognizing our work! For the weakness you mentioned, we address each as follows.
>
>
> > **(W1) The literature review is somewhat incomplete. The paper does not sufficiently cover some recent graph domain adaptation works.**
>
>
> Thank you very much for pointing out. We have identified some recent works such as [1-5] and will include them in the Related Work section for a more detailed discussion. During rebuttal, we have conducted additional experiments based on recent graph domain adaptation (GDA) works [1-2]. Please refer to our response to (W3) and Table 2 below.
>
> > **(W2) The method is only evaluated on graph-level tasks, but domain adaptation is also important for node-level prediction with established benchmarks. Discussion of whether and how GRADATE could be applied at the node level would be helpful.**
>
> This is a very insightful question. Although GRADATE explicitly deals with graph-level tasks, one possible extension to solve node-level tasks is as follows: **decomposing source/target graphs into set of ego-graphs** where each of these ego-graphs represent the local topology of each node, and applying GRADATE directly to select the **optimal nodes** for adaptation.
>
> This is indeed a very interesting future direction that requires more investigation on (i) *how to decide the size of local vicinity for each ego-graph*, (ii) *how to co-consider node/edge selection for optimal adaptation* and (iii) *how to mitigate the information loss from extracting ego-graphs*. We will add this discussion to the Outlook section to inspire further studies.
>
> > **(W3) The approach relies on labeled target-domain validation data, which is often unavailable in realistic test-time adaptation scenarios.**
>
> We would like to clarify that:
> - **Validation label is not required to apply GRADATE**. As demonstrated in Remark 3.2 (line 165), setting label weight `c=0` means validation label is un-used during the selection process. While c was originally introduced as a hyper-parameter, we recognize that using `c ≠ 0` implicitly assumes access to labeled validation data, which is not always available.
>
> - **After restricting to validation-label-free setting, the advantage of GRADATE remains similar to, or even better than, what is reported in paper.** To fairly address this concern, we restrict GRADATE to the label-free setting and re-evaluate its performance as follows. Specifically,
>     - For Table 1, we report **relative performance improvement (%)** of GRADATE over the strongest baseline under all settings (i.e. *GNN + X% selected data with Dataset Y*)
>     - For Table 2, we include the (i) best GDA baseline out of the four considered in paper (termed as "Best GDA"), (ii) three new GDA methods (including A2GNN-ADV [1], A2GNN-MMD [1], and TDSS [2]), and (iii) data selection methods combined with the best GNN backbone (termed as "Best Random", "Best LAVA" and "GRADATE"). The numbers represent mean performances. Note that we use **bold** and *italic* to indicate the 1st/2nd best results.
>
> - **Table 1 (GRADATE outperforms other data selection methods)**
>
> | Dataset   | GCN (10%)  | GCN (20%) | GCN (50%) | GIN (10%) | GIN (20%) | GIN (50%) |
> |:-----------:|:-----------:|:-----------:|:-----------:|:-----------:|:-----------:|:-----------:|
> | IMDB-BINARY | +11.13%  | +9.92%    |  +2.76%   | +4.63%     |  +5.16%   | +13.37%     |
> | IMDB-MULTI | +221.31%   | +201.09%    |  +189.61%   | +45.78%    |  +95.01%   | +87.04%     |
> |  MSRC_21  | +1.99%  | +1.36%     |  +3.01%   | +2.36%     |  +0.09%   | +2.88%     |
> |  ogbg-molbace  | +1.32%  | -4.39%    |  +13.17%   | -16.32%     |  +2.84%   | +7.48%     |
> |  ogbg-molbbbp  | +0.00%  | -14.46%     |  +0.00%   | -0.08%    |  +2.18%  | +2.22%     |
> |  ogbg-molhiv  | +3.05%  | +0.75%     |  +0.48%   | -1.24%     |+0.35%   | +2.52%     |
>
> - **Table 2 (GRADATE outperforms GDA baselines and other selection methods)**
>
> | Dataset   | Best GDA| A2GNN-ADV| A2GNN-MMD | TDSS | Best Random| Best LAVA | GRADATE
> |:-----------:|:-----------:|:-----------:|:-----------:|:-----------:|:-----------:|:-----------:|:-----------:|
> | IMDB-BINARY| 0.822 | 0.831 | *0.845* | 0.753 | 0.738 | 0.835 | **0.858**
> | IMDB-MULTI| 0.123 | 0.099 | 0.142 | 0.136 | 0.180 | *0.790* | **0.800**
> | MSRC_21| 0.833 | 0.824 | 0.818 | 0.833 | 0.801 | *0.851* | **0.865**
> | ogbg-molbace | **0.683** | 0.602 | 0.590 | 0.641 | 0.622 | 0.637 | *0.651*
> | ogbg-molbbbp | **0.778** | 0.528 | 0.564 | 0.560 | 0.528 | 0.642 | *0.656*
> | ogbg-molhiv | 0.567 | 0.622 | 0.620 | 0.466 | 0.602 | *0.633* | **0.637**
>
>
> > **(W4) Despite approximations, the method’s time complexity remains high and could limit scalability.**
>
> Thank you for raising the concern. We have reported the runtime of GRADATE on three TUD datasets in Section L (Table 10 in line 724) to demonstrate the real-world efficiency of GRADATE. Here, we also provide the runtime on three other OGB datasets. Even for the largest dataset (i.e. *ogbg-molhiv* , which contains ~42K graph samples), the empirical runtime of GRADATE remains moderate relative to the GNN training time.
>
> - **Table 3 (Runtime (in second) of GRADATE)**
>
> | **Procedure / Dataset**            | **ogbg-molbace** | **ogbg-molbbbp** | **ogbg-molhiv** |
> |:---------------------------------:|:------------------:|:------------------:|:-----------------:|
> | **Off-line Computation**          |                  |                  |                 |
> | FGW Pairwise distance             | 15.44            | 19.67            | 283.99          |
> | Label-informed pairwise distance  | 0.12             | 0.17             | 53.82           |
> | **On-line Computation**           |                  |                  |                 |
> | GREAT algorithm (Algorithm 2)     | 0.43             | 0.84             | 295.42          |
> | LAVA algorithm                    | 0.06             | 0.13             | 39.45           |
> | **GNN Training Time**             |                  |                  |                 |
> | GCN (w/ 10% data)                 | 12.76            | 13.91            | 190.34          |
> | GCN (w/ 20% data)                 | 13.08            | 22.59            | 312.08          |
> | GCN (w/ 50% data)                 | 22.21            | 30.48            | 845.46         |
>
>
> [1] https://arxiv.org/abs/2402.05660
>
> [2] https://arxiv.org/abs/2412.11654
>
> [3] https://arxiv.org/abs/2403.01092
>
> [4] https://dl.acm.org/doi/abs/10.1145/3503161.3548012
>
> [5] https://arxiv.org/abs/2306.04979

---

> > ### Comment · Reviewer_pN6d · 2025-08-05
> >
> > Thanks to the authors for providing detailed experimental results. Please ensure that these experiments are included in the final manuscript.

---

> > > ### Author Response · Authors · 2025-08-05
> > >
> > > We thank the reviewer for recognizing our work and the constructive feedback. We will also make sure to include additional experiment results in the updated manuscript.

---

### Decision · Program_Chairs · 2025-09-17

**Decision:**

Accept (poster)

**Comment:**

This paper proposes GRADATE, a model-free approach for data selection to address domain adaptation in graph-level classification. Shifting the focus from model-centric to data-centric solutions, this work leverages optimal transport (OT) theory to select a subset of source data that is most aligned with the target domain, without relying on a specific GNN architecture for the selection process. The method is supported by theoretical motivation and extensive empirical results.

The reviewers recognized the novelty of the data-centric approach for graph domain adaptation (GDA). However, the idea of using OT-based approaches for graph data selection is not entirely new, given existing works [1, 2] that have proposed to use Tree Mover's Distance for size generalization and graph subsampling. Initial reviews also raised important concerns, primarily regarding the use of labeled validation data in the selection process, which could lead to an unfair comparison with standard GDA methods. However, the authors effectively addressed this in their rebuttal by providing new experiments in a completely label-free setting, demonstrating that the method's strong performance holds. While some debate remained regarding the precise theoretical guarantees and the distinction between a data selection method versus a GDA method, the majority of reviewers were satisfied with the authors' clarifications and the strength of the empirical evidence.

Overall, this paper introduces a valuable and well-motivated data-centric perspective to the problem of graph domain adaptation. Despite some lingering discussion on its theoretical framing, the method's simplicity and strong empirical results make it a solid contribution to the field. Therefore, I recommend acceptance.

That said, I strongly encourage the authors to incorporate the following points in the camera-ready version:

1. While the proposed method could be applicable to other problems, the current paper is centered on graph domain adaptation, as stated in the introduction and validated by the experiments. The present title is too broad and does not reflect the actual scope. In particular, the phrase “under distribution shifts” is ambiguous—it can refer to both out-of-distribution (OOD) generalization and domain adaptation, whereas this work focuses only on the latter. Moreover, the method specifically addresses covariate shifts, which are only one type of distribution shift. I therefore recommend specifying “domain adaptation” and “covariate distribution shifts” explicitly in the title.

2. The additional experiments provided in the rebuttal substantially strengthen the paper—for example, the evaluation under the validation-label-free setting, the results with graph transformers, and the time cost comparison. These results should be incorporated into the main paper.

3. As noted earlier, prior studies [1, 2] have already explored optimal transport (OT) approaches for graph data selection. These works should be properly discussed in both the method and related work sections.

[1] Georgiev et al., Beyond Erdos-Rényi: Generalization in Algorithmic Reasoning.

[2] Jain et al., Subsampling Graphs with GNN Performance Guarantees.